# Scalable and Interpretable Representation Alignment with Ordinal Similarity

Diogo Soares [1]  Pankhil Gawade [1 2]  Andrea Dittadi [1 2]  Ewa Szczurek [1 3]

## Abstract

Evaluating representation similarity is fundamental to representation learning. However, existing metrics suffer from significant limitations: they lack interpretability due to shifting baselines, lack robustness to outliers, and are computationally intractable for large datasets, forcing reliance on heuristic approximations. To address this, we develop an ordinal-similarity framework, instantiated by the Triplet (TSI) and Quadruplet (QSI) Similarity Indices, which measure alignment by quantifying the consistency of ordinal relationships. We theoretically demonstrate this formulation is inherently interpretable, robust to outliers, and computationally efficient. Finally, we establish a formal equivalence between TSI and local neighborhood alignment, measured by Mutual Nearest Neighbors. Empirically, we validate these properties and show that ordinal similarity offers a scalable approach to measuring alignment, enabling practitioners to better understand and design representations.

## 1. Introduction

The ability to quantify representational similarity is a foundational challenge in modern machine learning, essential for interpreting model behavior (Nguyen et al., 2021; Huh et al., 2024; Klabunde et al., 2025) and aligning artificial systems with human cognition (Muttenthaler et al., 2023; Sucholutsky et al., 2025). Similarity metrics are central to understanding how neural networks interpret semantic concepts, directly informing the design of transfer learning, knowledge distillation (Park et al., 2019; Gou et al., 2021), and generative modeling (Yao et al., 2025; Yu et al., 2025; Leng et al., 2025). Given their extensive influence on both

[1]Institute of AI for Health, Helmholtz Munich [2]School of Computation, Information and Technology, Technical University of Munich [3]Faculty of Mathematics, Informatics and Mechanics, University of Warsaw. Correspondence to: Ewa Szczurek <ewa.szczurek@helmholtz-munich.de>.

*Proceedings of the 43rd International Conference on Machine Learning*, Seoul, South Korea. PMLR 306, 2026. Copyright 2026 by the author(s).

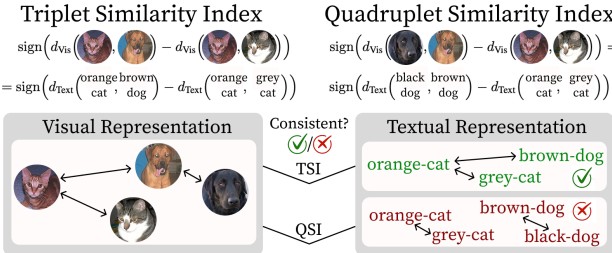

*Figure 1.* TSI and QSI measure alignment between two representation spaces (e.g., Visual and Textual) by quantifying the consistency of ordinal relationships. TSI checks if relative similarity from an anchor is preserved (e.g., 'Is $A$ closer to $B$ than to $C$?'). QSI compares relative similarity between distinct pairs (e.g., 'Is $A$ closer to $B$ than $C$ is to $D$?')

model design and behavioral analysis, the reliability of these metrics is paramount for the interpretability of increasingly ubiquitous AI systems.

However, despite their widespread use, current metrics suffer from key shortcomings: *(i)* **Complex to Interpret:** The interpretation of most metrics is inherently difficult as they yield raw scores that lack intuitive meaning and, as noted by Cloos et al. (2025), fail to provide a consistent threshold to distinguish meaningful alignment from spurious similarity. *(ii)* **Sensitive to Outliers:** Most metrics lack robustness to outliers, a fragility so pronounced that two perfectly aligned representations can appear misaligned after introducing only a small number of extreme outliers (Davari et al., 2023). *(iii)* **Computationally Infeasible:** Existing metrics demand significant memory resources and extensive computation time (Nguyen et al., 2021), leading to a reliance on practical, often batched, approximations (Huh et al., 2024). However, these approximations lack theoretical bounds for correctness and can yield inconsistent estimates.

To overcome these challenges, we shift our focus from metrics reliant on precise nominal values to those that emphasize more robust relational properties. Inspiration for our work comes from Non-metric Multidimensional Scaling (NMDS), a data visualization technique that arranges points in a 2D space to preserve the rank order of pre-specified pairwise distances, rather than their exact values (Borg & Groenen, 2005; Agarwal et al., 2007; Van Der Maaten & Weinberger, 2012; Vankadara et al., 2023). The success of NMDS in creating perceptually aligned visualizations, demonstrates that preserving ordinal relationships captures

essential structure needed for alignment. Building on this insight, we measure representational similarity by quantifying the consistency of ordinal relationships between two representation spaces. Specifically, we evaluate the fraction of similarity triplets (of the form "A is more similar to B than to C") and quadruplets (of the form "A is more similar to B than C is to D") for which the relationship holds true in both representations.

To make this ordinal principle concrete, we study it through the Triplet (TSI) and Quadruplet (QSI) Similarity Indices (Figure 1). We analyze the key theoretical properties of ordinal consistency and its connections to existing methods. Our main contributions are:

- **Ordinal Similarity Formulation (Section 4.1):** We formalize ordinal consistency through TSI and QSI, two metrics based on the agreement of relative similarity and distance comparisons.
- **Theoretical Interpretability (Section 4.2):** We show that ordinal similarity has a direct probabilistic interpretation and establish stable baselines for dissimilar representations (Lemma 1).
- **Connection to Neighborhood Consistency (Section 4.3):** We prove a formal equivalence between perfect TSI alignment and Mutual Nearest Neighbors (MNN) across all scales (Corollary 1).
- **Guaranteed Robustness (Section 4.4):** We establish theoretical bounds that guarantee robustness to outliers for ordinal similarity (Lemma 2, Corollary 2).
- **Computational Scalability (Section 4.5):** We provide efficient algorithms for exact calculation (Corollary 3) and a principled approximation scheme with theoretical error bounds (Corollary 4).
- **Empirical Validation (Section 6):** We demonstrate the utility of TSI and QSI in diverse real-world applications, including tracking training representation convergence and CLIP multimodal alignment.

Together, these contributions establish ordinal similarity as a rigorous, scalable and theoretically grounded framework for representation alignment, giving practitioners a reliable way to understand and interpret their models.

## 2. Related Work

This section reviews existing similarity metrics and situates our approach within the broader landscape of representational analysis.

**Similarity Metrics.** Numerous similarity metrics have been proposed to compare representations (Sucholutsky et al., 2025; Klabunde et al., 2025), with formal characterizations provided in Section A. Foundational work includes Canonical Correlation Analysis (CCA) (Hardoon et al., 2004) and its refinements: SVCCA (Raghu et al., 2017), which filters for variance-explaining directions, and PWCCA (Morcos et al., 2018), which weights correlations by explained variance. Another seminal contribution, Centered Kernel Alignment (CKA) (Kornblith et al., 2019), satisfies critical invariance properties by computing a normalized Hilbert-Schmidt Independence Criterion (HSIC) on pairwise distance matrices. This method and the metrics proposed here are both inspired by Representational Similarity Analysis (RSA) (Kriegeskorte et al., 2008), a neuroscience framework for measuring alignment across brain areas (Freiwald & Tsao, 2010) and individuals (Connolly et al., 2012).

More recently, methods focusing on local structure, particularly Mutual Nearest Neighbors (MutualNN) and CKNNA, have gained widespread use. MutualNN directly computes the average number of shared $k$-nearest neighbors between representations, while CKNNA calculates a version of CKA masked to these mutual neighborhoods (Huh et al., 2024). Such local metrics have been instrumental in analyzing representation convergence and the alignment between generative and foundation models (Huh et al., 2024; Duraphe et al., 2025; Yu et al., 2025; Leng et al., 2025).

**Concepts Related to Ordinal Similarity.** Seminal ordinal correlation coefficients such as Kendall's Tau (Kendall, 1938) and Spearman's Rho (Spearman, 1961) motivated the study of relational properties and share important connections with the metrics analyzed in this work. In addition, preserving ordinal relationships in the $d$-dimensional vector space has been studied within the scope of NMDS (Borg & Groenen, 2005; Agarwal et al., 2007; Van Der Maaten & Weinberger, 2012; Vankadara et al., 2023) with important theoretical results on the uniqueness and existence of aligned hyper-dimensional pointclouds derived over the years (Bilu & Linial, 2005; Jamieson & Nowak, 2011; Kleindessner & von Luxburg, 2014; Chatziafratis & Indyk, 2024). From an application perspective, several methodologies have been proposed that leverage ordinal information, such as evaluation metrics assessing whether embeddings satisfy a *pre-defined set* of triplet relationships (Veit et al., 2017; Vankadara et al., 2023; Li et al., 2024), similarity kernels constructed from such sets (Tamuz et al., 2011; Kleindessner & von Luxburg, 2017), and loss functions that optimize them (Schroff et al., 2015). Crucially, these methods typically rely on an externally provided set of triplet relationships as input, rather than evaluating the consistency of ordinal relationships between two representation spaces.

## 3. Background & Notation

Let us consider a set of points from an original input space, $\{z_i\}_{i=1}^N$. Let $f : \mathcal{Z} \to \mathcal{X}$ and $g : \mathcal{Z} \to \mathcal{Y}$ be two embedding functions that map these inputs into two different

representation spaces. This process yields a paired dataset of embeddings $(x_i, y_i)_{i=1}^N$, where each pair is generated from the same source, i.e., $x_i = f(z_i)$ and $y_i = g(z_i)$. We define the complete sets of representations as $X = \{x_1, \ldots, x_N\}$ and $Y = \{y_1, \ldots, y_N\}$. Furthermore, assume that the similarities in $\mathcal{X}$ and $\mathcal{Y}$ are individually modeled through their corresponding distance functions $d_X$ and $d_Y$. The field of similarity metrics aims to quantify the alignment between $(X, d_X)$ and $(Y, d_Y)$ using a metric, which can be viewed as a function $\mathcal{M}_{d_X, d_Y}(X, Y)$ that maps the two sets of representations to a score, typically in the range $[0, 1]$, where higher scores correspond to greater similarity.

## 4. Ordinal Similarity Metrics

To measure representation alignment, we evaluate the consistency of ordinal relationships between spaces via two metrics: the Triplet Similarity Index (TSI), which evaluates anchor-based relative similarity, and the Quadruplet Similarity Index (QSI), which compares distances between disjoint pairs.

Our focus on ordinal relations is deliberate. While discarding absolute distances may appear to inflate similarity, ordinal methods such as NMDS, as well as our TSI and QSI, impose a number of constraints that grows cubically (or faster) with the number of points $N$, sharply restricting admissible configurations (Borg & Groenen, 2005). Moreover, under suitable conditions and as $N \to \infty$, ordinal relationships uniquely determine a representation up to similarity-preserving transformations (e.g., isotropic scaling, orthogonal transforms) (Kleindessner & von Luxburg, 2014), making ordinal consistency a principled and robust measure of geometric alignment.

### 4.1. Formal Definitions

Our metrics are built upon a generalized ordinal comparison function, $O_d(a, b, c, d) = \text{sign}\big(d(a, b) - d(c, d)\big) \in \{-1, 0, 1\}$, which captures the relationship between a pair of distances in a space with metric $d$. A special case of this function, which we denote $O_d^\Delta$, compares the distances from a single anchor point $a$ to two other points, $b$ and $c$. It is defined as $O_d^\Delta(a, b, c) \equiv O_d(a, b, a, c) = \text{sign}\big(d(a, b) - d(a, c)\big)$. Using these functions, we define our metrics as the expected agreement of ordinal comparisons over sets of points.

**Definition 1** (Triplet Similarity Index (TSI)). Let $\mathcal{T} := \{(i, j, k) \in \{1, \ldots N\} | i \neq j, i \neq k, j \neq k\}$ be the set of all triplets of distinct indices. The TSI is the average agreement of the three-point (anchor-based) ordinal comparison function:

$$\text{TSI}_{d_X, d_Y}(X, Y) =$$
$$\mathbb{E}_{(i,j,k) \in \mathcal{T}}\Big[\mathbb{I}\big[O_{d_X}^\Delta(x_i, x_j, x_k) = O_{d_Y}^\Delta(y_i, y_j, y_k)\big]\Big].$$

For QSI, we use the set $\mathcal{Q}$ of all quadruplets of distinct indices.

**Definition 2** (Quadruplet Similarity Index (QSI)). The QSI is the average agreement of the ordinal comparison function over all distinct quadruplets:

$$\text{QSI}_{d_X, d_Y}(X, Y) =$$
$$\mathbb{E}_{(i,j,k,l) \in \mathcal{Q}}\Big[\mathbb{I}\big[O_{d_X}(x_i, x_j, x_k, x_l) = O_{d_Y}(y_i, y_j, y_k, y_l)\big]\Big].$$

This reliance on ordinal comparisons ensures our metrics satisfy a critical set of invariance properties. As highlighted by Kornblith et al. (2019), a similarity metric should be invariant to translations, isotropic scaling, and orthogonal transformations, but not to arbitrary invertible linear transforms. Given that TSI and QSI depend only on the relative ordering of distances, they directly satisfy these criteria in inner product spaces, as is formally analyzed in Section B.

Moreover, by showing that $1 - \text{TSI}$ and $1 - \text{QSI}$ can be formulated as normalized Hamming distances over ordinal relationship arrays, we formally prove that these corresponding distance functions satisfy all three metric axioms: equivalence, symmetry, and triangle inequality (see Section C for a complete analysis). This ensures that TSI and QSI offer theoretically grounded notions of dissimilarity, fulfilling the requirement from Williams et al. (2021) that representation distances should yield proper metrics.

### 4.2. Probabilistic Interpretation

An essential property of our ordinal metrics is their direct probabilistic interpretation. Defined as the expectation of an indicator function, their values represent the probability that a given ordinal relationship is preserved. The TSI, for example, is the probability that for a randomly chosen triplet of points, the relative ordering of distances from an anchor point is the same in both spaces (e.g., "$x_i$ is closer to $x_j$ than to $x_k$"):

$$\text{TSI}_{d_X, d_Y}(X, Y) = P_{(i,j,k) \in \mathcal{T}}\Big(O_{d_X}^\Delta(x_i, x_j, x_k) \\ = O_{d_Y}^\Delta(y_i, y_j, y_k)\Big) \quad (1)$$

Similarly, the QSI is the probability that for a randomly chosen quadruplet of distinct points, the ordinal relationship between a pair of distances is preserved:

$$\text{QSI}_{d_X, d_Y}(X, Y) = P_{(i,j,k,l) \in \mathcal{Q}}\Big(O_{d_X}(x_i, x_j, x_k, x_l) \\ = O_{d_Y}(y_i, y_j, y_k, y_l)\Big) \quad (2)$$

This probabilistic framing highlights the clear semantic value of these metrics. For instance, a TSI of 0.8 means that

80% of triplets agree on their ordering, in sharp contrast to metrics that yield hard-to-interpret opaque values. Additionally, these ordinal relationships yield predictable scores for both perfectly aligned and dissimilar representations. While a score of 1 indicates complete alignment, dissimilarity is captured when the ordinal relations induced by two representations are statistically independent. Sampling independent representations $X \perp\!\!\!\perp Y$ provides a natural and interpretable construction that induces this ordinal independence and therefore serves as a baseline for dissimilarity (see Section D for the rationale). Under this setting, the following lemma demonstrates that the expected score is approximately $\frac{1}{2}$ when the probability of ties is negligible.

**Lemma 1** (Expected Similarity for Independent Representations). *Let $X = \{x_i\}_{i=1}^N$ and $Y = \{y_i\}_{i=1}^N$ be two datasets, with points drawn i.i.d. from arbitrary and independent distributions, $\mathcal{D}_X$ and $\mathcal{D}_Y$, respectively, and equipped with distance functions $d_X$ and $d_Y$. This setting ensures that the ordinal comparisons in $X$ and $Y$ are independent. For TSI, let $a, b, c$ be points drawn i.i.d. from $\mathcal{D}_X$. We define the probability of a tie as $p_X^0 = P_{a,b,c \sim \mathcal{D}_X}\left(d_X(a,b) = d_X(a,c)\right)$. Similarly for $\mathcal{D}_Y$. The expected TSI is:*

$$\mathbb{E}_{\substack{X \sim \mathcal{D}_X^N \\ Y \sim \mathcal{D}_Y^N}}\left[TSI_{d_X, d_Y}(X, Y)\right] = p_X^0 p_Y^0 + \frac{1}{2}(1 - p_X^0)(1 - p_Y^0)$$

*For QSI, let $a, b, c, d$ be points drawn i.i.d. from $\mathcal{D}_X$. We define the probability of a tie as $p_X^0 = P_{a,b,c,d \sim \mathcal{D}_X}\left(d_X(a,b) = d_X(c,d)\right)$. Similarly for $\mathcal{D}_Y$. The expected QSI is:*

$$\mathbb{E}_{\substack{X \sim \mathcal{D}_X^N \\ Y \sim \mathcal{D}_Y^N}}\left[QSI_{d_X, d_Y}(X, Y)\right] = p_X^0 p_Y^0 + \frac{1}{2}(1 - p_X^0)(1 - p_Y^0)$$

**Universal dissimilarity baseline.** The significance of this result, whose proof is provided in Section N.1, lies in its universality: the baseline for independent representations does not depend on the number of samples, their dimensionality, or the type of distribution. This establishes $\frac{1}{2}$ as a consistent reference point for random alignment across diverse applications. While scores below this baseline are mathematically possible, theoretical analysis with 1D representations suggests that geometric constraints might make such extreme misalignment exceedingly rare in practice (see Section E). Consequently, when analyzing their ML models, practitioners can evaluate the TSI/QSI and confidently use $\frac{1}{2}$ as a robust dissimilarity reference, regardless of the architecture or set size. This is key, since previous work from Cloos et al. (2025) reported that existing metrics do not offer consistent thresholds for determining good similarity scores, thus requiring case-specific calibration that hinders method interpretability.

### 4.3. Connection to Neighborhood Consistency

A fundamental characteristic of any similarity metric is its alignment condition, i.e., the set of joint representations for which the metric yields the maximum score of 1. This condition formalizes the notion of perfect alignment and provides a semantic anchor for interpreting lower scores. For TSI, the alignment condition is given by the following equivalence:

**Corollary 1** (Perfect TSI alignment is equivalent to joint Mutual Nearest Neighbors alignment). *Let $(X, d_X)$ and $(Y, d_Y)$ be two sets of $N$ representations without ties. Then, full alignment of the TSI is equivalent to joint alignment of Mutual Nearest Neighbors for all $k$-nearest neighbors with $k$ ranging from 1 to $N - 2$:*

$$TSI_{d_X, d_Y}(X, Y) = 1 \iff$$
$$\forall k \in \{1, \dots, N-2\} : MutualNN_{k, d_X, d_Y}(X, Y) = 1$$

*where $MutualNN_{k, d_X, d_Y}(X, Y)$ is the Mutual Nearest Neighbors score for $k$ neighbors.*

This result, proven in Section N.2.1, is significant because it addresses the limitations of traditional neighborhood descriptors. While Mutual Nearest Neighbors (MNN) is extensively used to analyze representation convergence (Huh et al., 2024) and forms the backbone of metrics like CKNNA, individual MutualNN$_k$ scores suffer from specific limitations: they only measure consistency for a fixed number of neighbors ($k$), ignore their relative ordering within that set, and become ill-defined when ties prevent unique neighborhood determination. Although evaluating neighborhood consistency across all possible scales would provide a more complete view, such an exhaustive approach is typically computationally prohibitive. The equivalence in Corollary 1 shows that TSI overcomes this by jointly evaluating local neighborhood consistency across all scales. Importantly, as we show later, TSI can be efficiently estimated with high precision in time independent of the dataset size $N$, positioning it as a computationally efficient metric for capturing multi-scale neighborhood consistency. We further explore this relationship in Section F, where we show that TSI alignment also serves as a lower bound for a convex combination of MNN coefficients.

### 4.4. Robustness to Outliers

A similarity metric is unreliable if small subsets disproportionately affect its value. TSI and QSI address this by relying on ordinal relationships rather than precise values, which naturally limits the influence of extreme fluctuations. This robustness is formally captured in Lemma 2, which establishes tight bounds on the change in similarity when a subset of representations is transformed.

**Lemma 2** (Bounds for Ordinal Similarity Metrics under Partial Transformations). *Let the set of indices $\{1, \dots, N\}$*

*be partitioned into two disjoint subsets, $U$ (unchanged) and $V$ (transformed). Let $T : \mathcal{X} \to \mathcal{X}$ be a transformation, and define a new dataset $X' = \{x'_i\}_{i=1}^N$ where $x'_i = x_i$ for $i \in U$ and $x'_i = T(x_i)$ for $i \in V$. For similarity scores $S \in \{TSI, QSI\}$, the score of the transformed dataset, $S(X', Y)$, is bounded by the score on the unchanged subset, $S(X_U, Y_U)$, as follows:*

$$c_S \cdot S(X_U, Y_U) \leq S(X', Y) \leq c_S \cdot S(X_U, Y_U) + (1 - c_S)$$

*where $c_S$ is the proportion of comparisons drawn exclusively from the unchanged set $U$: $c_{TSI} = \frac{(|U|)_3}{(N)_3}$ and $c_{QSI} = \frac{(|U|)_4}{(N)_4}$, where $(n)_k = n(n-1)\cdots(n-k+1)$ denotes the falling factorial.*

This result, proven in Section N.3.1, quantifies the extent to which the similarity score remains bounded by the untransformed subset. Within this paradigm, **outliers** (extreme numerical representations) and **corruptions** (perturbations of existing samples) can be viewed as partial transformations, allowing us to bound their deviation from the ground-truth, i.e., the ideal representations we wish to evaluate but which are often inaccessible: (i) the clean subset after removing outliers, $|S(X_U, Y_U) - S(X', Y)|$; and (ii) the original dataset without corruptions, $|S(X, Y) - S(X', Y)|$.

**Corollary 2** (TSI and QSI are Robust to Corruptions and Outliers). *Consider the setup of Lemma 2 where $U$ indices the subset of clean or unchanged points and $V$ corresponds to the subset of corruptions/outliers. For any similarity metric $S \in \{TSI, QSI\}$, the absolute deviation of the similarity score is bounded by:*

$$|S(X_U, Y_U) - S(X', Y)| \leq 1 - c_S \quad \textit{(Outliers)}$$
$$|S(X, Y) - S(X', Y)| \leq 1 - c_S \quad \textit{(Corruptions)}$$

The deviation upper-bounds in Corollary 2 guarantee that neither extreme outliers nor corrupted entries can disproportionately shift the similarity score. Since $(1 - c_S) \to 0$ as the fraction of transformed points $|V|/N$ vanishes, ordinal similarity is theoretically robust to sparse outliers/corruptions (proof in Section N.3.2).

## 4.5. Efficient Computation

A critical requirement for any metric is computational tractability (Räisä et al., 2025). While naive TSI and QSI implementations require $\mathcal{O}(N^3)$ and $\mathcal{O}(N^4)$ time, resulting in prohibitive runtimes, in this section, we show that not only can our metrics be computed exactly with a complexity comparable to existing methods, but they also admit a principled approximation scheme with theoretical guarantees.

**Tractable Exact Computation.** Noting the connection between TSI, QSI, and Kendall's Tau Correlation Coefficient (Kendall, 1938), as explored in Section G, we draw

inspiration from efficient algorithms for the Kendall's Tau computation (Knight, 1966) to significantly reduce the cost of exact computation of TSI and QSI, leading to improved time-complexity bounds:

**Corollary 3** (Computational Complexity of TSI and QSI). *TSI can be computed exactly in $\mathcal{O}(N^2 \log N)$ time and $\mathcal{O}(N)$ space. Assuming a symmetric distance function, QSI can be computed exactly in $\mathcal{O}(N^2 \log N)$ time and $\mathcal{O}(N^2)$ space.*

This quadratic complexity, established in Section H.1, makes our metrics practical for many real-world datasets.

**Principled Approximation for Large-Scale Data.** While quadratic complexity is a significant improvement, it can still be a bottleneck for the massive datasets common in modern machine learning. This has led other metrics to rely on heuristic, batched approximations that often lack theoretical guarantees of convergence (Nguyen et al., 2021; Huh et al., 2024). In contrast, a key advantage of our ordinal framework is that TSI and QSI, being bounded U-statistics, provide high-quality estimates with probabilistic bounds on their deviation error by evaluating the ordinal relationships of a small subset of uniformly sampled triplets/quadruplets, respectively. We formalize these concepts in Corollary 4 and provide supporting formal analysis in Section H.2.

**Corollary 4** (Computational Complexity of Approximate TSI and QSI). *By evaluating $\lceil \frac{1}{2\epsilon^2} \log(\frac{2}{\delta}) \rceil$ triplets or quadruplets sampled uniformly at random, we can estimate TSI or QSI with an additive error of at most $\epsilon$ with a probability of at least $1 - \delta$. This approximation has a time complexity of $\mathcal{O}\left(\frac{1}{\epsilon^2} \log\left(\frac{1}{\delta}\right)\right)$ and an auxiliary space complexity of $\mathcal{O}(1)$.*

Notably, the complexity of this approximation is independent of the dataset size $N$. This yields a theoretically grounded and highly scalable method to estimate representational alignment, ensuring reliable results even on massive datasets where exact computation is infeasible.

## 4.6. Predictable Estimates of True Similarity

Beyond computing a similarity metric on a fixed dataset of size $N$, a fundamental question in practice is how reliably this score estimates the true alignment of the underlying data-generating processes (i.e., the similarity as $N \to \infty$). Because TSI and QSI are bounded U-statistics, we prove in Section I that they exhibit Hoeffding-type concentration bounds, guaranteeing that the probability of deviation from their true population values decays exponentially with $N$. This predictable convergence, a significant property not shown in existing metrics, perfectly complements the fact that, as $N \to \infty$, ordinal constraints uniquely determine representations up to similarity-preserving transformations

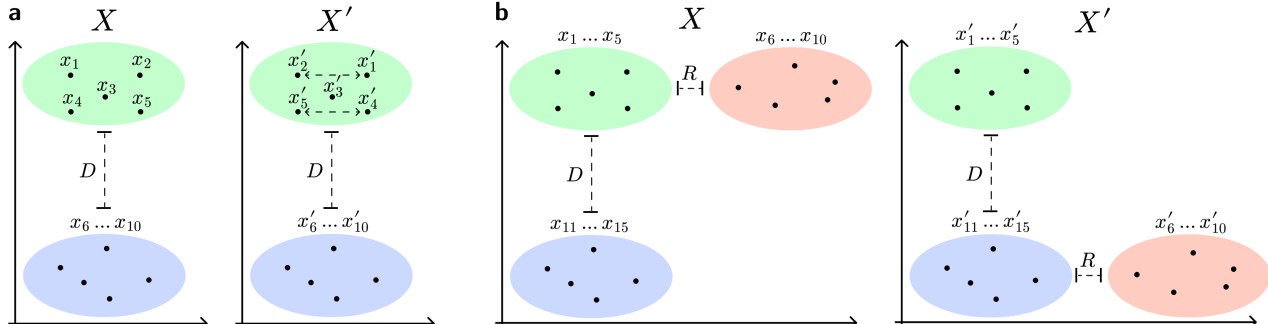

*Figure 2.* **(a)** Local failure mode of MutualNN and CKA: shuffling points within a cluster (green) destroys local geometric structure. **(b)** Global failure mode of MutualNN and CKNNA: rigidly translating one cluster (red) drastically alters the macroscopic layout.

(Kleindessner & von Luxburg, 2014). Because ordinal relationships uniquely dictate geometric structure in the large-sample limit, and our empirical metrics tightly concentrate around their population values, TSI and QSI serve as reliable and robust measurements of alignment between the generating processes of $X$ and $Y$, even for moderate sample sizes.

## 5. Illustrative Examples

We now present illustrative examples that expose concrete failure modes of existing metrics and highlight the complementary nature of TSI and QSI. Each example starts from a representation $X$, applies a structure-disrupting transformation to obtain $X'$, and evaluates whether each metric correctly reports misalignment (i.e., a score $< 1$ between representations $X, X'$). Throughout, all metrics use the Euclidean distance $d_X = \|\cdot\|_2$. For brevity, we only state the critical insights; formal computations are deferred to Section M.

### 5.1. Local Failure Mode of MutualNN and CKA

Metrics relying on neighborhood overlap or centered kernels can be blind to local geometric disruptions (Figure 2a), with MutualNN$_{k-1}(X, X') = 1$ and CKA$(X, X') \to 1$, while TSI$(X, X') < 1$ and QSI$(X, X') < 1$. **Input** $X \subset \mathbb{R}^d$ consists of $N = 2k$ representations ($k \geq 5$) partitioned into two clusters $C_1$ and $C_2$, each of size $k$, with all intra-cluster pairwise distances unique and inter-cluster distance $D \to +\infty$. The **transformed** $X'$ is obtained by applying a permutation $\pi$ that shuffles points *within* $C_1$ ($\pi(C_1) = C_1$), leaving at least one point fixed and swapping two pairs. This destroys local geometric structure while preserving cluster assignments.

### 5.2. Global Failure Mode of MutualNN and CKNNA

Local metrics based on $k$-nearest-neighbor structures can be blind to massive global rearrangements (Figure 2b), yielding MutualNN$_{k-1}(X, X') = 1$ and CKNNA$_{k-1}(X, X') = 1$, while TSI$(X, X') < 1$ and QSI$(X, X') < 1$. **Input** $X \subset$

$\mathbb{R}^d$ consists of $N = 3k$ representations ($k \geq 2$) partitioned into three tightly packed clusters $C_1, C_2, C_3$ with centroids $c_1, c_2, c_3$, inter-cluster distances $\|c_1 - c_2\| \approx D$ and $\|c_1 - c_3\| \approx R$ with $D \gg R \gg 1$, and $c_1 - c_2 \perp c_1 - c_3$. The **transformed** $X'$ translates the entire cluster $C_3$ by $c_2 - c_1$, leaving $C_1$ and $C_2$ unchanged, so that $\|c_1 - c_3'\| \approx \sqrt{D^2 + R^2}$ and $\|c_2 - c_3'\| \approx R$.

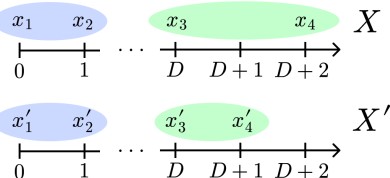

*Figure 3.* Distinguishing TSI and QSI: contracting the upper cluster from gap 2 to 1 preserves every point's nearest-neighbor ordering, yet alters the relative scale between disjoint intra-cluster pairs; TSI remains at 1, while QSI drops to $1/3$.

### 5.3. Distinguishing TSI and QSI

We show that QSI, as opposed to TSI, is sensitive to global geometric changes that leave local neighborhood structure intact (Figure 3), yielding QSI$(X, X') = \frac{1}{3}$ (severe misalignment) despite TSI$(X, X') = 1$ (perfect alignment). **Input** $X$ consists of $N = 4$ points in $\mathbb{R}$: $X = \{0, 1, D, D+2\}$ with $D \gg 1$, forming two remote clusters $\{x_1, x_2\}$ and $\{x_3, x_4\}$. The **transformed** $X'$ contracts the second cluster by mapping $x_4 \mapsto x_4' = D+1$, giving $X' = \{0, 1, D, D+1\}$.

## 6. Experiments

In this section, we compare the interpretability, sensitivity to outliers, and approximation quality of ordinal similarity (TSI/QSI) with metrics from previous work. Furthermore, we explore two real-world case studies: analyzing the convergence of representations during training, and evaluating multimodal alignment in CLIP models. We provide the mathematical formulation of these metrics in Section A and describe the parameter selection used to compute them in Section J. Note that for the experiments in Sections 6.3

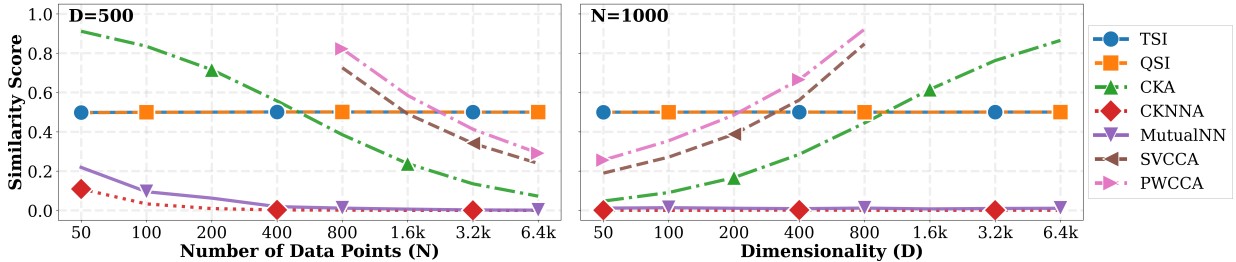

*Figure 4.* Analysis of the output of various similarity metrics in the setting of two statistically independent representation sets composed of $N$ uniformly sampled points from a unitary $D$ dimensional hypercube.

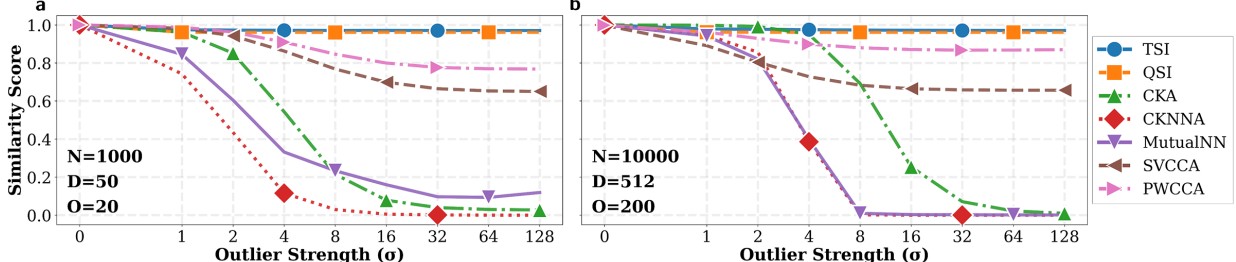

*Figure 5.* Evaluation of similarity metric sensitivity when 2% of the dataset is perturbed by outliers of increasing strength ($\sigma$). a) Synthetic representations sampled from a $D$-dimensional hypercube, b) Representations from a Vision Transformer trained on CIFAR-10.

to 6.5, we do not report SVCCA and PWCCA due to the numerical instability problems highlighted in Section K.1.

### 6.1. Ordinal Similarity Provides Consistent Baseline for Dissimilar Representations

We investigate the stability of dissimilarity scores i.e., scores obtained for two statistically independent representation sets, generated by sampling $N$ points from a $D$-dimensional uniform distribution, $\mathcal{U}([0, 1]^D)$. The evaluation was performed under two distinct conditions, with results averaged over 20 runs: (i) varying the number of samples $N$ with fixed dimensionality $D = 500$, and (ii) varying the dimensionality $D$ with a fixed number of samples $N = 1000$.

The results, presented in Figure 4, demonstrate that dissimilarity scores of several existing metrics are highly sensitive to these parameters. The scores of CKA, SVCCA and PWCCA strongly depend on both sample size and dimensionality, spanning nearly the entire range of possible values; in certain scenarios, they even yield almost perfect alignment for independent representations. While CKNNA and MutualNN appear stable, this is an artifact of their small neighborhood ($k = 10$): their sensitivity becomes noticeable at smaller sample sizes ($N \leq 400$) as $N$ approaches $k$; in the limit $k = N - 1$, CKNNA corresponds to CKA. In contrast, as predicted by our theoretical analysis in Lemma 1, TSI and QSI provide stable scores for statistical independence ($\approx 0.5$), yielding consistent similarity scores regardless of sample size or dimensionality. This stability provides an interpretable baseline, allowing alignment quality to be judged independently of these parameters.

### 6.2. Ordinal Similarity Metrics Are Robust to Outliers

Inspired by the experimental setting in Davari et al. (2023), we assess robustness by comparing an original representation set $X$ to a version $X'$ containing a small fraction of perturbed points. Following Section 4.4, these perturbations can be considered as either transforming normal points into outliers or directly injecting noisy corruptions. We evaluate two settings: a) a synthetic dataset of $N = 1000$ points sampled from $\mathcal{U}([0, 1]^{50})$, and b) 10000 real-world representations from a Vision Transformer (ViT) (Dosovitskiy et al., 2021) trained on CIFAR-10 (Krizhevsky, 2009) (details in Section L). In both cases, 2% of the data are designated for perturbation. To standardize the scale, each perturbed point is displaced by a vector $\bar{d}\sigma\mathbf{v}$, where $\bar{d}$ is the average Euclidean pairwise distance of the set, $\sigma$ is a scalar controlling the perturbation strength, and $\mathbf{v}$ is a uniformly sampled random unit vector. We report the change in similarity as a function of $\sigma$, averaged over 20 runs.

Given that 98% of the data points remain identical between the two representation sets, a reliable metric should output a consistently high similarity score close to 1. However, the results in Figure 5 show significant deviations for several standard metrics. In both the synthetic and real-world settings (panels a and b respectively), CKA, CKNNA, and MutualNN are highly sensitive to the perturbation magnitude $\sigma$, eventually dropping to near-zero values. This is counter-intuitive for neighborhood-based metrics; In Section K.2, we trace this failure to dot-product induced hubness, where a single outlier globally disrupts neighborhood structures. SVCCA and PWCCA are more resilient but still exhibit a noticeable decline as the outlier strength increases.

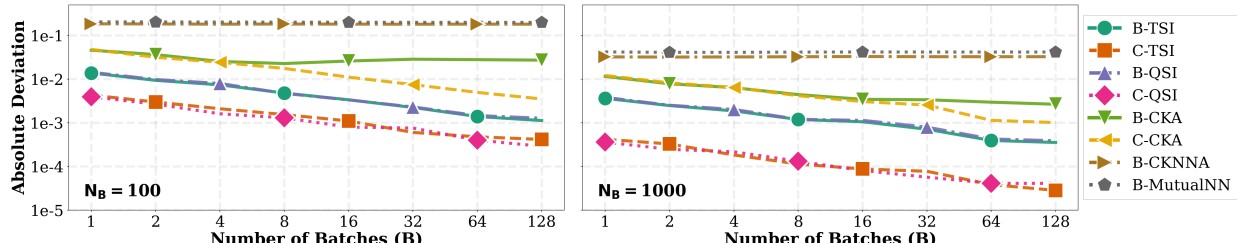

*Figure 6.* Evaluation of the approximation error for various similarity metrics. The plot reports the absolute deviation between the exact and approximate similarity scores as a function of the number of batches ($B$) for varying batch sizes ($N_B$).

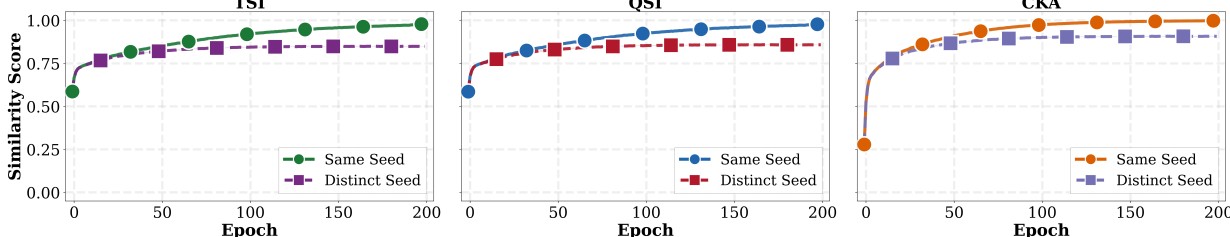

*Figure 7.* Evolution of representation similarity during training on CIFAR-10. We compare the representations of a ViT at epoch $t$ with its final representation (Same Seed) and with the final representation of a separately initialized model (Distinct Seed).

In contrast, TSI and QSI remain remarkably stable, matching the tight theoretical lower bounds from Lemma 2 (e.g., $\approx 94\%$ for TSI in panel a with $N = 1000$ and $|U| = 980$). Since these bounds depend only on the proportion of outliers and are independent of distance function selection, ordinal metrics are immune to failures of other approaches.

### 6.3. Ordinal Similarity Can Be Efficiently Approximated with Negligible Deviation

We evaluate the approximation quality of various metrics by measuring the absolute deviation between exact scores and their estimates as the computational budget increases. The metrics are applied to compute the similarity between initial and final representations of a ViT trained on CIFAR-10 (Section L). Following Section A.1, we compare standard batched approximations (B- prefix), which average over $B$ batches of size $N_B$, with metric-specific schemes (C- prefix). For CKA, the C- prefix denotes batching of individual HSIC coefficients (Nguyen et al., 2021), while for TSI and QSI it represents our approximate sampling approach, using $B \times N_B^2$ samples to match the computational cost of quadratic metrics.

As shown in Figure 6, the metric-specific schemes consistently outperform the standard batched approximations. In the regime of smaller batches ($N_B = 100$), standard approximations yield absolute deviations between $10^{-1}$ and $10^{-2}$. Given that similarity scores are bounded within $[0, 1]$, errors of this magnitude are significant, potentially obscuring meaningful distinctions in representation alignment. While increasing the batch size to $N_B = 1000$ reduces the deviation error for all metrics, standard local metrics (CKNNA, MutualNN) continue to exhibit high approximation errors

that stagnate despite increased computation. In contrast, TSI and QSI demonstrate rapid, monotonic convergence, yielding estimates orders of magnitude more precise than competing approaches. These empirical results validate the tight theoretical bounds established in Corollary 4, confirming that our ordinal framework offers not just scalability, but a mathematically guaranteed confidence in approximation quality that heuristic methods lack.

### 6.4. TSI and QSI Capture Representation Convergence in Real-World Settings

To validate TSI and QSI in a practical training scenario, we analyze representation convergence of a ViT when trained on CIFAR-10, see the setup in Section L. We track the alignment between intermediate model checkpoints at epoch $t$ and two reference points: (i) the final representation of the *same* model, and (ii) the final representation of a model trained with a *distinct* random seed. We compare our metrics against CKA; for the sake of conciseness, we opt not to report results for metrics that only capture local structure (MutualNN, CKNNA). To reflect real-world resource constraints, all metrics are approximated rather than exactly computed. These estimations use $B = 10$ batches of size $N_B = 1000$, utilizing metric-specific approximation schemes where applicable (see Section J for details). All results are averaged over 5 independent runs.

As shown in Figure 7, TSI and QSI accurately capture the training dynamics, exhibiting behavior consistent with established expectations that similarity increases as training progresses. In the "Same Seed" regime, both metrics start near the random dissimilarity baseline ($\approx 0.5$) and monotonically increase to perfect alignment ($1.0$) as the model

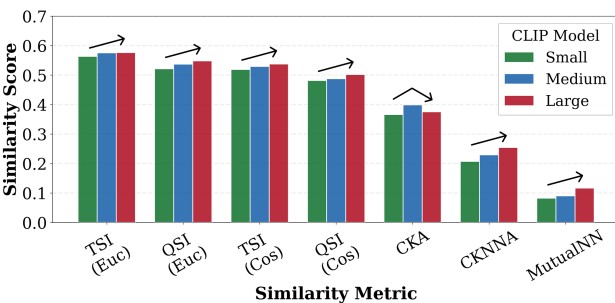

*Figure 8.* Multimodal alignment analysis using CLIP models of increasing scale on the ImageNet validation set.

converges to itself. In the "Distinct Seed" regime, the metrics correctly identify that while models trained from different initializations converge to similar representations, they do not reach identical alignment. These results confirm that ordinal metrics effectively measure representation evolution in real-world deep learning workflows, offering the utility of established methods like CKA while maintaining the theoretical robustness and interpretability advantages demonstrated in previous sections.

### 6.5. Ordinal Similarity Consistently Tracks Multimodal Alignment Across Model Scales

A major real-world application of similarity metrics is evaluating multimodal alignment, such as vision-language consistency in CLIP (Radford et al., 2021). We analyze three CLIP variants from `sentence-transformers` (Reimers & Gurevych, 2019)—Small (ViT-B/32), Medium (ViT-B/16), and Large (ViT-L/14)—which exhibit increasing ImageNet zero-shot accuracy (63.3%, 68.1%, and 75.4%) on the validation set of ImageNet containing 50k datapoints. Following the original CLIP evaluation, we compare image representations with prompt-based label embeddings ("A photo of {label}"). As CLIP optimizes cosine similarity between modalities, larger models with higher accuracy should contain increasingly aligned representations. This is especially true for the alignment measured with dot products or cosine similarity, which is the case for all reported metrics except for TSI and QSI, which by default uses Euclidean distance. For these reasons, we also report TSI and QSI using negative cosine similarity. While this distance does not satisfy metric axioms and may therefore lack clear practical lower bounds (Section E) or exhibit the counter-intuitive behaviors discussed in Section K.2, its direct link to the CLIP objective provides valuable insight into multimodal alignment. To enable realistic evaluation, all metrics are approximated rather than exactly computed. These estimations are computed using $B = 10$ batches of size $N_B = 1000$, utilizing metric-specific approximation schemes where applicable (see Section J for details).

The empirical results presented in Figure 8 largely confirm

these expectations, though they also reveal critical failure modes in established metrics. While TSI and QSI correctly capture the monotonic increase in alignment using both Euclidean distance and negative cosine similarity, CKA exhibits a counter-intuitive behavior: the similarity score drops when transitioning from the Medium to the Large model. This drop coincides with a change in embedding dimensionality, as both the Small and Medium models have 512 dimensions while the Large model has 768, further highlighting CKA's sensitivity to varying dimensions as previously discussed in Section 6.1. While local metrics like CKNNA and MutualNN also capture the alignment increase, they are intrinsically limited as they only evaluate the consistency of very reduced local substructures. In contrast, our ordinal metrics capture the global geometric structure composed of these local substructures, as formally established by the connection between MutualNN and TSI in Section 4.3.

This use-case underscores the practical advantages of our approach. Beyond enabling scalable, high-confidence evaluation on large datasets like ImageNet, TSI and QSI offer immediate interpretability: a score of $0.57$, for instance, indicates that $57\%$ of relative distance relationships are preserved across modalities. This provides a clear geometric measure of alignment relative to the $0.5$ random baseline, underscoring the metrics' utility for practitioners seeking interpretable results at scale.

## 7. Conclusion

In this paper, we introduced TSI and QSI, redefining representational similarity via ordinal relationship consistency. By prioritizing distance ordering over raw values, these metrics provide stable, interpretable baselines and a direct probabilistic interpretation across varying dimensions and sample sizes. We theoretically established the equivalence of ordinal alignment to multi-scale neighborhood consistency and proved robustness to outliers where metrics like CKA fail. Methodologically, we developed efficient algorithms for exact computation and a Monte Carlo sampling scheme with provable approximation guarantees independent of dataset size.

Empirical validation on Vision Transformer training dynamics and multimodal CLIP patterns confirms the practical utility of our approach. TSI and QSI correctly track alignment across varying model scales and training regimes, even as embedding dimensions change, where metrics like CKA often fail. Furthermore, our proposed sampling schemes achieve approximation precision orders of magnitude higher than standard batching, enabling high-confidence evaluation on massive datasets like ImageNet. By combining mathematical rigor with scalable computation, ordinal similarity provides a principled and reliable framework for evaluating representational alignment in modern machine learning.

## Impact Statement

This paper presents work whose goal is to advance the field of Machine Learning. There are many potential societal consequences of our work, none which we feel must be specifically highlighted here.

## Code Availability

Code is available at
https://github.com/diogosoares22/ordinal-similarity-metrics.

## Acknowledgments

This project has received funding from the European Research Council (ERC) under the European Funding Union's Horizon 2020 research and innovation programme (grant agreement No 810115 – DOG-AMP).

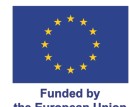 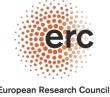

Conflict of Interest: Projects in Szczurek lab at the University of Warsaw are cofunded by Merck Healthcare.

## LLM Usage

This paper was written with the assistance of Gemini2.5 (Comanici et al., 2025), which was used to enhance language clarity and flow. All content has been reviewed and edited to ensure originality and accuracy.

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

# Appendix

We organize the Appendix as follows:

- Section A: We provide formal mathematical definitions for all similarity metrics evaluated in our experiments.

- Section B: We analyze the invariance properties of TSI and QSI under various transformations in inner product spaces.

- Section C: We show that the complementary distances $1 - \text{TSI}$ and $1 - \text{QSI}$ satisfy the metric axioms.

- Section D: We provide a causal perspective on why statistical independence serves as a natural baseline for representational dissimilarity.

- Section E: We analyze the occurrence of low ordinal similarity scores for representations in $\mathbb{R}$ and compute practical lower bounds.

- Section F: We establish a theoretical connection between TSI and Mutual Nearest Neighbors (MNN) by deriving a TSI upper bound consisting of a convex combination of MNN scores.

- Section G: We explore the connection between our ordinal metrics and an adjusted version of the Kendall rank correlation coefficient.

- Section H: We describe efficient algorithms for both exact and approximate computation of TSI and QSI.

- Section I: We define population similarity and prove concentration bounds for empirical TSI and QSI as a function of dataset size.

- Section J: We detail the experimental setup, including metric parameters and model training configurations.

- Section K: We discuss the impact of distance proxy selection on the sensitivity of local neighborhood-based metrics to outliers; Additionally, we provide intuition for the numerical instability of SVCCA and PWCCA.

- Section L: We describe the training procedure of a Vision Transformer used to generate CIFAR-10 latent representations for our experiments.

- Section M: We present full derivations for the illustrative counterexamples summarized in the main text (local/global failure modes of baseline metrics, and TSI/QSI distinction).

- Section N: We present the complete mathematical proofs for all lemmas and corollaries introduced in this manuscript.

## A. Similarity Metrics

This section provides a brief mathematical formulation of the similarity metrics considered in this paper. For notation purposes, we assume $X$ and $Y$ are ordered sets of $N$ data points. When these are vectorial representations, they can be viewed as matrices $X \in \mathbb{R}^{N \times L_X}$ and $Y \in \mathbb{R}^{N \times L_Y}$. Similarity Metrics are functions of the form $\mathcal{M}_{\text{parameters}}(X, Y) \in [0, 1]$, where 1 indicates complete similarity and 0 indicates complete dissimilarity. Although, as shown in Section 6.1, it is often the case that the practical baseline dissimilarity is higher than 0.

The CCA, SVCCA, and PWCCA metrics assume the representations are vectorial and operate directly on them, in contrast to other metrics that leverage either distance functions or kernel operators on the underlying representation spaces, denoted as $d_X, d_Y$, and $k_X, k_Y$ respectively.

**CCA** Canonical correlation finds bases for two sets of representations, $X$ and $Y$, such that, when the original matrices are projected onto these bases, the correlation is maximized. The overall similarity is the average of the correlations of multiple, mutually orthogonal pairs of such projections, known as canonical correlations.

Let $m = \min(L_X, L_Y)$ be the number of canonical components. For each component $i$ from 1 to $m$, the canonical correlation $\rho_i$ is found by solving:

$$\rho_i(X,Y) = \max_{\mathbf{w}_X^i \in \mathbb{R}^{L_X}, \mathbf{w}_Y^i \in \mathbb{R}^{L_Y}} \text{corr}(X\mathbf{w}_X^i, Y\mathbf{w}_Y^i)$$
$$\text{subject to} \quad \forall j < i \quad \text{cov}(X\mathbf{w}_X^i, X\mathbf{w}_X^j) = 0$$
$$\forall j < i \quad \text{cov}(Y\mathbf{w}_Y^i, Y\mathbf{w}_Y^j) = 0.$$

The final CCA score is the unweighted mean of these correlations:

$$\text{CCA}(X,Y) = \frac{1}{m} \sum_{i=1}^{m} \rho_i(X,Y)$$

**SVCCA** Singular Vector CCA (SVCCA) improves upon CCA by first performing a dimensionality reduction step. Instead of performing CCA on the raw representations, it first identifies and projects the data onto the most significant principal components, i.e., those that explain most of the variance.

First, a Singular Value Decomposition (SVD) is performed on the centered data matrices, $X = U_X \Sigma_X V_X^T$ and $Y = U_Y \Sigma_Y V_Y^T$. The number of components to retain, $k_X$ and $k_Y$, is chosen to explain at least 99% of the variance in each space, based on the singular values ($\sigma_{X,i}, \sigma_{Y,i}$):

$$k_X = \min \left\{ k \mid \frac{\sum_{i=1}^{k} \sigma_{X,i}^2}{\sum_{i=1}^{L_X} \sigma_{X,i}^2} \geq 0.99 \right\} \quad \text{and} \quad k_Y = \min \left\{ k \mid \frac{\sum_{i=1}^{k} \sigma_{Y,i}^2}{\sum_{i=1}^{L_Y} \sigma_{Y,i}^2} \geq 0.99 \right\}$$

The original representations are then projected onto these principal directions:

$$X' = XV_X^{1:k_X} \quad \text{and} \quad Y' = YV_Y^{1:k_Y}$$

where $V^{1:k}$ denotes the matrix containing the first $k$ columns of $V$. SVCCA is then defined as the CCA score between these dimensionality-reduced representations:

$$\text{SVCCA}(X,Y) = \text{CCA}(X',Y')$$

**PWCCA** Projection Weighted CCA (PWCCA) refines this approach by acknowledging that not all canonical directions equally reflect the underlying similarity of the original data. It computes a weighted average of the canonical correlations, where the weights are proportional to the amount of information explained by each of the CCA projection directions.

The importance weight $\tilde{\alpha}_i$ for each normalized canonical direction $w_X^i$ is calculated as the sum of the absolute projections of the original feature vectors (columns of the data matrix $X$) onto that direction:

$$\tilde{\alpha}_i(X,Y) = \sum_{j=1}^{L_X} \frac{|\langle Xw_X^i, X_{:,j} \rangle|}{\|Xw_X^i\|}$$

The final PWCCA score is the weighted average of the canonical correlations using normalized weights:

$$\text{PWCCA}(X,Y) = \sum_{i=1}^{m} \frac{\tilde{\alpha}_i(X,Y)}{\sum_{k=1}^{m} \tilde{\alpha}_k(X,Y)} \rho_i(X,Y)$$

where $m$ is the number of canonical components.

**CKA** Centered Kernel Alignment (CKA) is a similarity metric based on the Hilbert-Schmidt Independence Criterion (HSIC) (Gretton et al., 2005), which measures the dependence between two kernel matrices derived from the representations. First, pairwise relationship matrices $K_X$ and $K_Y$ are computed using a kernel function, $k$. Common choices include the Linear Kernel ($k(a,b) = a^\top b$) and the RBF Kernel ($k(a,b) = \exp(-\|a-b\|^2/2\sigma^2)$). These functions map pairs of vectors to a similarity score, but they do not satisfy the axioms of a distance metric.

The kernel matrices are defined as:

$$(K_X)_{ij} = k_X(x_i, x_j) \quad \text{and} \quad (K_Y)_{ij} = k_Y(y_i, y_j)$$

The HSIC statistic is then computed as $\text{HSIC}(A, B) = \frac{1}{(N-1)^2} \operatorname{tr}(AHBH)$, where $H = I_N - \frac{1}{N} J_N$ is a centering matrix. CKA normalizes the HSIC between the two representations' kernel matrices by the HSIC of each kernel matrix with itself. This process ensures the resulting score is invariant to isotropic scaling and lies within the range $[0, 1]$.

$$\text{CKA}_{k_X, k_Y}(X, Y) = \frac{\text{HSIC}(K_X, K_Y)}{\sqrt{\text{HSIC}(K_X, K_X)\text{HSIC}(K_Y, K_Y)}}$$

**MutualNN** Mutual Nearest Neighbors (MutualNN) quantifies similarity by assessing the preservation of local neighborhoods. For a chosen hyperparameter $k$, the metric evaluates the overlap between the $k$-nearest neighbors of each point in the two representation spaces. Specifically, the set of shared neighbors for a point $i$ is the intersection of its neighbor sets from space $\mathcal{X}$ and space $\mathcal{Y}$:

$$\text{SharedNN}_{k,d_X,d_Y}(i) := \text{KNN}_{k,d_X}(X \setminus \{x_i\}, x_i) \cap \text{KNN}_{k,d_Y}(Y \setminus \{y_i\}, y_i)$$

The final MutualNN score is the average fraction of shared neighbors across all data points, a value naturally normalized between 0 and 1.

$$\text{MutualNN}_{k,d_X,d_Y}(X, Y) = \frac{1}{N} \sum_{i=1}^{N} \frac{|\text{SharedNN}_{k,d_X,d_Y}(i)|}{k}$$

Here, $|\cdot|$ denotes the cardinality of the set, measuring the size of the intersection.

**CKNNA** Centered Kernel Nearest-Neighbor Alignment (CKNNA) integrates CKA's global perspective with MutualNN's focus on local structure. It achieves this by calculating a CKA-like score restricted only to the mutual nearest neighbors of each point. The process begins by constructing a binary mask matrix, $M$, based on the shared nearest neighbors (as defined in MutualNN) for a given $k$:

$$M(k)_{ij} = \mathbb{I}[j \in \text{SharedNN}_{k,d_X,d_Y}(i)]$$

This mask is then applied element-wise to the kernel matrices, effectively zeroing out relationships between points that are not in the same local neighborhood. The masked HSIC is defined as:

$$\text{HSIC}_{\text{masked}}(A, B, M) = \text{HSIC}(A \odot M, B \odot M)$$

Finally, the CKNNA score uses the standard CKA normalization but with the masked HSIC, thus measuring alignment only within these corresponding local neighborhoods.

$$\text{CKNNA}_{k,k_X,k_Y,d_X,d_Y}(X, Y) = \frac{\text{HSIC}_{\text{masked}}(K_X, K_Y, M(k))}{\sqrt{\text{HSIC}_{\text{masked}}(K_X, K_X, M(k))\text{HSIC}_{\text{masked}}(K_Y, K_Y, M(k))}}$$

This formulation is unique in that it requires both a distance function to construct the nearest-neighbor mask and a kernel function for the subsequent HSIC calculation. While these are treated as independent choices, it is worth noting that a distance function can be derived from a similarity kernel provided certain conditions are met (Schölkopf, 2000).

### A.1. Batched Approximations

Due to the significant computational complexity of the aforementioned similarity metrics, which often scale quadratically with the number of samples $N$, their exact calculation on large datasets is frequently impractical. Consequently, batched approximation methods are widely used (Huh et al., 2024).

A general approach employed in these works is to estimate similarity by examining the metric's scores over a randomly sampled smaller mini-batch. Here, we expand this approach by averaging over $B$ mini-batches. Let $(X_b, Y_b)_{b=1}^{B}$ denote the mini-batches sampled from the full datasets, then, the metric $\mathcal{M}$ can be approximated as:

$$\widehat{\mathcal{M}}_{\text{parameters}}(X, Y) = \frac{1}{B} \sum_{b=1}^{B} \mathcal{M}_{\text{parameters}}(X_b, Y_b)$$

This approach can be readily applied to CKA, CKNNA, MutualNN, SVCCA and PWCCA. Although straightforward, directly averaging the final score of these metrics (like MutualNN and CKNNA) can result in biased estimates.

For CKA, a more principled approximation was proposed by Nguyen et al. (2021), which avoids this source of bias by separately estimating the numerator and denominator of the CKA formula. This method relies on an unbiased HSIC estimator, denoted $HSIC_1$, which differs from the biased estimator used in the standard definition. The unbiased estimator is defined as:

$$\text{HSIC}_1(K, L) = \frac{1}{n(n-3)} \left( \text{tr}(\tilde{K}\tilde{L}) + \frac{\mathbf{1}^\top \tilde{K}\mathbf{1}\mathbf{1}^\top \tilde{L}\mathbf{1}}{(n-1)(n-2)} - \frac{2}{n-2}\mathbf{1}^\top \tilde{K}\tilde{L}\mathbf{1} \right)$$

With $K_{X_b}$ and $K_{Y_b}$ as the kernel matrices for a mini-batch $b$, the batched CKA estimator is constructed as the ratio of the averaged unbiased HSIC estimates:

$$\widehat{\text{CKA}}_{k_X, k_Y}(X, Y) = \frac{\frac{1}{B} \sum_{b=1}^{B} \text{HSIC}_1(K_{X_b}, K_{Y_b})}{\sqrt{\left( \frac{1}{B} \sum_{b=1}^{B} \text{HSIC}_1(K_{X_b}, K_{X_b}) \right) \left( \frac{1}{B} \sum_{b=1}^{B} \text{HSIC}_1(K_{Y_b}, K_{Y_b}) \right)}}$$

Crucially, Nguyen et al. (2021) highlight that when sampling with replacement and as $B \to \infty$, the average of the batch-wise $HSIC_1$ terms converges to the ground-truth $HSIC_1$ value for the full dataset, ensuring that the overall CKA estimate converges to the true unbiased CKA score.

## B. Invariances in Inner Product Spaces

An important property of similarity metrics is their invariance to specific geometric transformations. In inner product spaces, metrics should be invariant to translations, isotropic scaling, and orthogonal transformations, but not to arbitrary linear transforms (Kornblith et al., 2019; Klabunde et al., 2025). In this section, we analyze whether ordinal similarities meet these requirements, starting with the definition of metric invariance in Definition 3.

**Definition 3** (Metric Invariance). For datasets $X \subset \mathcal{X}$ and $Y \subset \mathcal{Y}$, the metric $\mathcal{M}_{d_X, d_Y}(X, Y)$ is invariant in its first argument to a transformation $T : \mathcal{X} \to \mathcal{X}$ if $\mathcal{M}_{d_X, d_Y}(T(X), Y) = \mathcal{M}_{d_X, d_Y}(X, Y)$, where $T(X) = \{T(x) \mid x \in X\}$. If $\mathcal{M}$ is symmetric, this invariance extends to the second argument. Thus, we simply say that $\mathcal{M}$ is invariant to $T$.

Following this definition, we consider an inner product space $(\mathcal{X}, \langle \cdot, \cdot \rangle)$ and its induced metric $d_X(x_1, x_2) = \sqrt{\langle x_1 - x_2, x_1 - x_2 \rangle}$. For such metrics, we derive the following invariance results:

**Corollary 5** (Ordinal Similarity Metric Invariances in Inner Product Spaces). *Let $(\mathcal{X}, \langle \cdot, \cdot \rangle)$ be an inner product space with its induced metric $d_X$, then TSI and QSI are invariant to the following transformations:*

1. ***Translation:*** *$T(x) = x + c$, for any constant vector $c \in \mathcal{X}$.*

2. ***Isotropic Scaling:*** *$T(x) = \alpha x$, for any non-zero scalar $\alpha \in \mathbb{R}$.*

3. ***Orthogonal Transformation:*** *A linear map $T$ that preserves the inner product, i.e., $\langle T(x_1), T(x_2) \rangle = \langle x_1, x_2 \rangle$ for all $x_1, x_2 \in \mathcal{X}$.*

*Furthermore, TSI and QSI are **not** generally invariant to:*

1. ***Invertible Linear Transformations:*** *A transformation $T(x) = Ax$, where $A$ is a linear operator for which an inverse $A^{-1}$ exists such that $AA^{-1} = A^{-1}A = I$.*

These results, proven in Section N.5, have practical implications in common Machine Learning settings. For instance, when evaluating the continuous latent representations of neural networks, this means that rotating, translating, or isotropically scaling these representations does not affect the outcome of ordinal similarity metrics.

## C. TSI/QSI-based distances satisfy the metric axioms

The study of metric properties for representational similarity was first introduced by Williams et al. (2021), who motivated the importance of derived distances satisfying the metric axioms to offer theoretically grounded scores that can be reliably used for other applications.

**Corollary 6** (Complementary TSI and QSI Distances Satisfy the Metric Axioms). *Let $(X, d_X)$, $(Y, d_Y)$, and $(Z, d_Z)$ be sets of $N$ representations equipped with their respective distance functions. For a similarity metric $S \in \{TSI, QSI\}$, define the distance function $d_S((X, d_X), (Y, d_Y)) = 1 - S_{d_X, d_Y}(X, Y)$. Then, $d_S$ satisfies the following metric axioms, as originally defined in* Williams et al. (2021):

1. **Equivalence:** $d_S((X, d_X), (Y, d_Y)) = 0 \iff (X, d_X) \sim (Y, d_Y)$

2. **Symmetry:** $d_S((X, d_X), (Y, d_Y)) = d_S((Y, d_Y), (X, d_X))$

3. **Triangle inequality:** $d_S((X, d_X), (Z, d_Z)) \leq d_S((X, d_X), (Y, d_Y)) + d_S((Y, d_Y), (Z, d_Z))$

*Therefore, these are proper metrics over the space of representation sets defined by the equivalence relation $S_{d_X, d_Y}(X, Y) = 1$.*

This result, proven in Section N.6, has important practical implications. By establishing that $1 - $ TSI and $1 - $ QSI satisfy the metric axioms, we ensure they provide mathematically rigorous notions of distance between representation spaces. Consequently, these distance functions can be reliably employed in downstream applications that rely on formal metric properties, such as clustering, nearest-neighbor retrieval, and topological data analysis.

## D. Why do Statistically Independent Representations represent Dissimilarity?

An alternative perspective on similarity metrics can be achieved through the lens of causal analysis. Following the notation from Section 3, the sets of representations $X = \{f(z_i)\}_{i=1}^N$ and $Y = \{g(z_i)\}_{i=1}^N$ are generated from a common set of source samples $Z = \{z_i\}_{i=1}^N$. The set $Z$ can typically be interpreted as sampling $N$ times from a probability distribution $\mathcal{D}_Z$. In effect, this means that $X$ and $Y$ can be constructed from a two-step process of first sampling from $\mathcal{D}_Z$ and then transforming the samples with $f$ and $g$ respectively. This process establishes the causal graph shown in Figure 9.

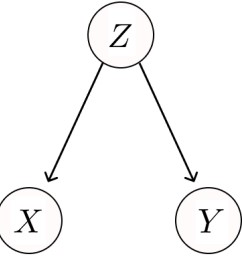

*Figure 9.* Causal graph illustrating that the source concepts, $Z$, are a common cause for the two representation sets, $X$ and $Y$. The similarity between $X$ and $Y$ is induced by this shared origin.

This causal structure reveals that the relationship between $X$ and $Y$ is induced by $Z$ acting as a common cause. This becomes apparent with the cats and dogs example from Figure 1. If $z_i$ represents the abstract concept of a "grey cat," its corresponding representations—$x_i$ as text and $y_i$ as an image—can be considered similar precisely because they originate from this shared source.

Conversely, if this causal link were broken, $x_i$ and $y_i$ would become statistically independent. This scenario is equivalent to comparing representations of unrelated concepts, such as the text for an "orange cat" and the image of a "brown dog." Even for well-defined embedding functions f and g, such a comparison between uncorrelated inputs is expected to yield dissimilarity. Therefore, statistical independence serves as a natural baseline for evaluating representational dissimilarity.

## E. Practical Lower Bounds for Ordinal Dissimilarity

Building on the expected similarity for independent representations (Lemma 1), we now investigate whether ordinal similarity can drop significantly below the $\frac{1}{2}$ statistical baseline. While such scores are mathematically possible, we hypothesize that adversarial configurations become increasingly rare as $N$ grows. To ground this intuition, we investigate the 1D Euclidean setting, where geometric constraints prohibit zero-similarity for even small $N$. We conjecture that as $N$ increases, the consistency required for extreme misalignment becomes so restrictive that the range of achievable scores is pulled towards the statistical baseline, though we do not formally prove this convergence here.

**Corollary 7** (Computability of Global Lower Bounds in $\mathbb{R}$). *For any number of considered representations $N$, the global lower bound of the ordinal similarity $S \in \{TSI, QSI\}$ over sets of $N$ distinct points in $\mathbb{R}$ is theoretically computable. Specifically, let $s_{\min} = \min_{X,Y \in \mathbb{R}^N \text{ with distinct entries}} S(X, Y)$. Then:*

$$s_{\min} = \min_{X,Y \in \mathcal{X}_{rep}} \left( \min_{\sigma \in Perm(N)} S(X, Y^\sigma) \right)$$

*where $\mathcal{X}_{rep}$ denotes a representative set as defined in Definition 7. Note that $\mathcal{X}_{rep}$ is finite and we can computationally obtain at least one such set (specifically a Monotonically Ordered Representative Set via Algorithm 6).*

This result, whose proof is provided in Section N.10, enables us to formally investigate the limits of dissimilarity by reducing the global search to a finite representative set. Crucially, this allows for the exact calculation of lower bounds for small $N$ through a computer-assisted search, which can then be used to extrapolate conclusions for all $N$. In particular, we show in Corollary 8 that for any $N$ beyond a small constant, complete misalignment in $\mathbb{R}$ is geometrically impossible, ensuring strictly positive similarity for all representation mappings.

**Corollary 8** (Universal Strict Positivity of Representations in $\mathbb{R}$). *Let $X, Y \subset \mathbb{R}$ be ordered sets of $N$ scalar representations equipped with the standard Euclidean distance $d_X(a, b) = d_Y(a, b) = |a - b|$. For any configuration where both $X$ and $Y$ consist of distinct points ($x_i \neq x_j$ and $y_i \neq y_j$ for $i \neq j$) and $N \geq 5$, it is impossible to obtain a similarity score of zero. In other words, the similarity is strictly positive for all such mappings:*

$$\forall X, Y \subset \mathbb{R} \text{ with distinct entries}, \quad TSI_{d_X, d_Y}(X, Y) > 0$$

*The same strict positivity holds for QSI if $N \geq 6$:*

$$\forall X, Y \subset \mathbb{R} \text{ with distinct entries}, \quad QSI_{d_X, d_Y}(X, Y) > 0$$

*Proof.* The result follows by construction; it suffices to verify the base cases $N = 5$ for TSI and $N = 6$ for QSI. Since these metrics are expectations over triplets and quadruplets respectively, any dataset larger than these base cases necessarily contains sub-configurations with non-zero scores, ensuring the global metric is strictly positive. To show this statement for a particular $N$, we do so via a computer-assisted proof leveraging the computability of the global lower bound for TSI and QSI as shown in Corollary 7. In Table 1, we present the obtained global lower bounds for TSI and QSI for all $N$ from 3 to 6, and observe that TSI is lower bounded by 0.13 for $N = 5$ and QSI is lower bounded by 0.11 for $N = 6$. $\qquad\square$

As shown in Table 1, the global lower bounds are non-decreasing with $N$, consistent with our hypothesis. For example, the TSI lower bound rises from 0.13 ($N = 5$) to 0.17 ($N = 6$). This behavior suggests that geometric constraints compound as the dataset grows, potentially pulling the minimum achievable similarity toward the statistical baseline. Computations are restricted to $N \leq 6$ due to the rapid expansion of the representative search space.

The increasing difficulty of constructing misaligned representations stems from the rigidity of geometric constraints. In 1D Euclidean space, the triangle identity $d(i, k) = d(i, j) + d(j, k)$ creates a dense network of dependencies; forcing misalignment in one triplet often necessitates alignment in another. As $N$ grows, these interlocking rules make adversarial "zero-similarity" configurations impossible and support our hypothesis that the global minimum is pulled toward the $\frac{1}{2}$ baseline. We expect this pattern to generalize to any distance satisfying the triangle inequality, reinforcing the statistical baseline as a practical lower bound for dissimilarity in real-world applications.

**Reproducibility of Computed Practical Lower Bounds**   To compute the values in Table 1, we implemented and executed the procedures described in Algorithms 5 and 6. For the construction of the Monotonically Ordered Representative Sets

*Table 1.* Global lower bounds for TSI and QSI for all $X, Y \subset \mathbb{R}$ consisting of $N$ distinct points. As $N$ increases, the geometric constraints of the real line prevent the existence of perfectly misaligned (zero-similarity) configurations.

| | Lower Bound | |
|---|---|---|
| **Number of Points ($N$)** | **TSI** | **QSI** |
| 3 | 0.00 | – |
| 4 | 0.00 | 0.00 |
| 5 | 0.13 | 0.00 |
| 6 | 0.17 | 0.11 |

(MORS), we utilized the linear programming solver from the `scipy.optimize` module (Virtanen et al., 2020), with a precision threshold of $\Delta = 10^{-6}$. The computational time varied significantly with $N$: while the evaluation for $N \leq 5$ was completed in less than 2 minutes, the exhaustive search for $N = 6$ required approximately 7 days of computation time due to the combinatorial expansion of the representative space. These results provide a concrete validation of our theoretical framework and establish the practical lower bounds for ordinal similarity in one-dimensional spaces.

## F. TSI's connection to Mutual Nearest Neighbors

In this section, we deepen the theoretical connection between TSI and Mutual Nearest Neighbors (MNN). We first clarify that the standard MNN coefficient (Section A) is fundamentally ill-defined in the presence of ties. Specifically, the set of $k$-nearest neighbors becomes ambiguous when multiple points share the same distance to an anchor, such that the transition from $k$ to $k+1$ is not uniquely determined. To maintain mathematical rigor, both Corollaries 1 and 9 assume representations without ties.

While Corollary 1 establishes a formal equivalence between TSI and MNN at perfect alignment (score = 1), it is natural to investigate whether a relationship persists for lower similarity values. In Corollary 9, we prove that TSI is upper-bounded by a convex combination of MNN coefficients across all possible scales $k$. Interestingly, the weights in this combination increase with $k$, suggesting that MNN descriptors at larger scales carry more information regarding the global ordinal structure captured by TSI. This result demonstrates that TSI and MNN are deeply intertwined even in regimes of partial alignment, positioning TSI as a robust proxy for multi-scale neighborhood consistency.

**Corollary 9** (TSI alignment is upper-bounded by a convex combination of Mutual Nearest Neighbors coefficients). *Let $(X, d_X)$ and $(Y, d_Y)$ be two sets of $N$ representations without ties, i.e., $\forall i, j, k \in \{1, \ldots, N\}, d_X(x_i, x_j) = d_X(x_i, x_k) \implies j = k$ (same for $Y$). Then, the TSI alignment is upper-bounded by a convex combination of Mutual Nearest Neighbors coefficients:*

$$TSI_{d_X, d_Y}(X, Y) \leq \frac{1}{\binom{N-1}{2}} \sum_{k=1}^{N-2} k \cdot MutualNN_{k, d_X, d_Y}(X, Y)$$

*where $MutualNN_{k, d_X, d_Y}(X, Y)$ is the Mutual Nearest Neighbors score for $k$ neighbors as defined in Section A.*

This bound, whose proof is detailed in Section N.2.2, formally bridges the gap between local compositional properties and global representational similarity, showing that high TSI alignment necessitates a strong preservation of neighborhood structures across multiple scales.

## G. Connection to Kendall Rank Correlation Coefficient

TSI and QSI can be intuitively understood through their connection to the Kendall rank correlation coefficient (Kendall, 1938), also known as Kendall's $\tau$. The Kendall-$\tau$ coefficient is a non-parametric statistic that measures the ordinal association between two quantities. Let $A = \{a_1, \ldots, a_M\}$ and $B = \{b_1, \ldots, b_M\}$ be two sequences of paired observations. The coefficient is defined as:

$$\text{Kendall-}\tau(A, B) = \frac{1}{M(M-1)} \sum_{j \neq k} \text{sign}(a_j - a_k)\text{sign}(b_j - b_k) \tag{3}$$

The product of sign functions captures agreement between non-tied pairs. To create a comprehensive similarity score that is equivalent to our ordinal metrics, we must also account for ties. We introduce an adjustment term, as specified in Equation (4), that rewards agreement on ties and penalizes disagreements involving ties.

$$
\begin{aligned}
\text{Adj-Ties}(A, B) = \frac{1}{M(M-1)} \sum_{j \neq k} \Big( &\mathbb{I}\big[\text{sign}(a_j - a_k) = \text{sign}(b_j - b_k) = 0\big] \\
&- \mathbb{I}\big[\text{sign}(a_j - a_k) = 0 \wedge \text{sign}(b_j - b_k) \neq 0\big] \\
&- \mathbb{I}\big[\text{sign}(a_j - a_k) \neq 0 \wedge \text{sign}(b_j - b_k) = 0\big] \Big)
\end{aligned}
\tag{4}
$$

This allows us to define the Adjusted Kendall Tau Coefficient:

$$
\text{Adj-Kendall-}\tau(A, B) = \frac{\text{Kendall-}\tau(A, B) + \text{Adj-Ties}(A, B) + 1}{2}
\tag{5}
$$

Besides incorporating ties, this transformation also maps the standard correlation from the range $[-1, 1]$ to $[0, 1]$ to ensure range consistency with similarity metrics. Equipped with this coefficient, we can provide an alternative formulation for TSI. As the following corollary shows, TSI can be interpreted as the average Adjusted Kendall Tau with respect to the distances from each anchor, a connection that grounds our metric in established statistical methods.

**Corollary 10** (TSI as an Average of Anchor-based Kendall's Tau). *For each anchor point $i$, let $D_X^i = \{d_X(x_i, x_t) | t \neq i\}$ and $D_Y^i = \{d_Y(y_i, y_t) | t \neq i\}$ denote the collections of distances from that anchor, treated as paired sequences ordered by the index $t$. Then TSI can be expressed as:*

$$
TSI_{d_X, d_Y}(X, Y) = \frac{1}{N} \sum_{i=1}^{N} Adj\text{-}Kendall\text{-}\tau(D_X^i, D_Y^i)
$$

*where the Adjusted Kendall Tau is calculated according to Equation (5).*

We provide a formal proof for this statement in Section N.7.2. Additionally, this perspective can be expanded to reveal a deeper relationship encompassing both TSI and QSI. The next corollary demonstrates that, assuming symmetric distance functions, the Adjusted Kendall Tau over all unique pairwise distances can be decomposed into a convex combination of TSI and QSI, directly reinforcing the connection between our proposed metrics and conventional statistical methods.

**Corollary 11** (Unified Kendall's Tau Formulation for TSI and QSI). *Let $D_X^{all} = \{d_X(x_i, x_t) | i < t\}$ and $D_Y^{all} = \{d_Y(y_i, y_t) | i < t\}$ denote the collections of all unique pairwise distances, treated as paired sequences ordered lexicographically by $(i, t)$. Then, for distance functions $d_X$ and $d_Y$ that satisfy the symmetry property, i.e., $d(a, b) = d(b, a)$, a weighted sum of TSI and QSI can be expressed as a function of Kendall's Tau coefficient:*

$$
\frac{4}{N+1} TSI_{d_X, d_Y}(X, Y) + \frac{N-3}{N+1} QSI_{d_X, d_Y}(X, Y) = Adj\text{-}Kendall\text{-}\tau(D_X^{all}, D_Y^{all})
$$

*where the Adjusted Kendall Tau is calculated according to Equation (5).*

Beyond the proof provided in Section N.7.3, we note to the advanced reader that the term Adj-Kendall-$\tau(D_X^{\text{all}}, D_Y^{\text{all}})$ is conceptually related to Representational Similarity Analysis (RSA) (Kriegeskorte et al., 2008) when computed using Kendall's $\tau$ correlation between distance matrices. Since RSA provides a versatile framework for measuring similarity, of which the above formulation represents a specific instantiation, our analysis of the Kendall's $\tau$ connection inherently captures these relationships.

Additionally, as we show in the following sections, these alternative formulations provide additional benefits, as they lay the foundation for the efficient computation of TSI and QSI.

# H. Efficient Computation of Ordinal Similarity Metrics

In this section, we explore algorithms for the efficient computation of TSI and QSI, analyzing their respective asymptotic time and space complexities. We divide our analysis into two key approaches: exact computation, for scenarios where the precise value of the metric is required, and approximate computation, where a degree of precision is relinquished in exchange of significant gains in computational speed, making the metrics applicable to large-scale applications.

## H.1. Exact Computation

Building upon the connection to the Kendall Tau Correlation coefficient explored in Section G, we can devise efficient algorithms for our metrics. The foundation of this approach is the efficient computation of the Adj-Kendall-$\tau$ coefficient. Inspired by algorithms for Kendall's Tau (Knight, 1966), we define a fast computation procedure in Algorithm 1, with its corresponding asymptotic complexity analysis provided in Corollary 12.

---

**Algorithm 1** Adjusted Kendall Tau Correlation Coefficient

---

**Require:** Two sequences of paired observations $A, B$
1: Let $\pi_A, \pi_B$ be the rank orderings of points based on $A, B$.
2: $T_A \leftarrow \text{CountTies}(\pi_A)$
3: $T_B \leftarrow \text{CountTies}(\pi_B)$
4: $T_{AB} \leftarrow \text{CountJointTies}(\pi_A, \pi_B)$
5: $I \leftarrow \text{CountInversions}(\pi_A, \pi_B)$
6: $C \leftarrow \frac{|A| \times (|A|-1)}{2} - I - T_A - T_B + T_{AB}$
7: $J \leftarrow 2 \times (C + T_{AB})$
8: **return** $J/(|A| \times (|A| - 1))$

---

**Corollary 12** (Computational Complexity of Adj-Kendall-$\tau$). *Let $A = \{a_1, \ldots, a_M\}$ and $B = \{b_1, \ldots, b_M\}$ be two sequences of paired observations. The Adj-Kendall-$\tau(A, B)$, described in Equation (5), can be computed in $\mathcal{O}(M \log M)$ time and $\mathcal{O}(M)$ space complexity.*

Equipped with this efficient subroutine (proof in Section N.8), we leverage the anchor-based formulation from Corollary 10 to construct Algorithm 2. The overall asymptotic complexity analysis for TSI, as summarized in Corollary 3, is derived by applying the $\mathcal{O}(M \log M)$ procedure for each of the $N$ anchor points ($M = N - 1$), resulting in a total time complexity of $\mathcal{O}(N^2 \log N)$ and a space complexity of $\mathcal{O}(N)$.

---

**Algorithm 2** Efficient TSI

---

**Require:** Two sets of $N$ representations $X, Y$, distance functions $d_X, d_Y$
1: $A_{\text{total}} \leftarrow 0$
2: **for** $i = 1 \rightarrow N$ **do**
3:     Let $D_X^i \leftarrow \{d_X(x_i, x_j)\}_{j \neq i}$ and $D_Y^i \leftarrow \{d_Y(y_i, y_j)\}_{j \neq i}$
4:     $A_{\text{total}} \leftarrow A_{\text{total}} + \text{Adj-Kendall-}\tau(D_X^i, D_Y^i)$
5: **end for**
6: **return** $A_{\text{total}}/N$

---

A similar approach, based on the joint formulation in Corollary 11, yields Algorithm 3 for the exact computation of QSI. This method, which requires a symmetric distance function, computes Adj-Kendall-$\tau$ over all $M = \binom{N}{2}$ unique pairwise distances. As formalized in the asymptotic complexity analysis in Corollary 3, this leads to a time complexity of $\mathcal{O}(M \log M) = \mathcal{O}(N^2 \log N)$ and a space complexity of $\mathcal{O}(M) = \mathcal{O}(N^2)$. While QSI calculation also requires TSI, the latter does not affect the overall asymptotic complexity of the combined procedure.

---

**Algorithm 3** Efficient QSI (for symmetric distances)

---

**Require:** Two sets of $N$ representations $X, Y$, distance functions $d_X, d_Y$
1: Let $D_X^{\text{all}} = \{d_X(x_i, x_t) | i < t\}$ and $D_Y^{\text{all}} = \{d_Y(y_i, y_t) | i < t\}$
2: **return**$\big((N+1) \times \text{Adj-Kendall-}\tau(D_X^{\text{all}}, D_Y^{\text{all}}) - 4 \times \text{TSI}_{d_X, d_Y}(X, Y)\big)/(N-3)$

---

**Corollary 3** (Computational Complexity of TSI and QSI). *TSI can be computed exactly in $\mathcal{O}(N^2 \log N)$ time and $\mathcal{O}(N)$ space. Assuming a symmetric distance function, QSI can be computed exactly in $\mathcal{O}(N^2 \log N)$ time and $\mathcal{O}(N^2)$ space.*

*Proof.* Algorithms 2 and 3 achieve the stated complexities by leveraging the $\mathcal{O}(M \log M)$ complexity of the Adjusted Kendall-$\tau$ subroutine (Corollary 12). For TSI, the subroutine is called $N$ times with $M = N - 1$, yielding $\mathcal{O}(N^2 \log N)$. For QSI, it is called once with $M = \binom{N}{2} \approx N^2/2$, yielding $\mathcal{O}(N^2 \log(N^2)) = \mathcal{O}(N^2 \log N)$. □

## H.2. Approximate Computation

To ensure scalability for modern large-scale datasets, we leverage the statistical structure of our metrics to enable fast approximation. Since TSI and QSI can be formulated as bounded U-statistics, they admit efficient approximation via Monte Carlo methods. By independently sampling a sufficient number of triplets or quadruplets, we can obtain an estimate that is arbitrarily close to the true value with high probability. This is formalized in Lemma 3.

**Lemma 3** (Approximation Bounds for Ordinal Similarity Metrics). *Let $\widehat{TSI}_n$ be the estimator for TSI obtained by sampling $n$ triplets uniformly at random with replacement from the set $\mathcal{T}$ and calculating the fraction of samples where the ordinal relationship is preserved. Similarly, let $\widehat{QSI}_n$ be the analogous estimator for QSI from $n$ random quadruplets. For any error tolerance $\epsilon > 0$, the estimators are bounded as follows:*

$$P\left(\left|\widehat{TSI}_n - TSI_{d_X, d_Y}(X, Y)\right| \geq \epsilon\right) \leq 2 \exp\left(-2n\epsilon^2\right)$$

$$P\left(\left|\widehat{QSI}_n - QSI_{d_X, d_Y}(X, Y)\right| \geq \epsilon\right) \leq 2 \exp\left(-2n\epsilon^2\right)$$

This lemma, whose proof is in Section N.4, has a direct practical consequence: by inverting the bound, we can determine the number of samples $n$ required to guarantee an estimate with additive error $\epsilon$ with probability at least $1 - \delta$. This calculation, $n = \lceil \frac{1}{2\epsilon^2} \log(\frac{2}{\delta}) \rceil$, is precisely what underpins the principled approximation method in Algorithm 4.

---

**Algorithm 4** Approximate Ordinal Similarity (TSI or QSI)

---

**Require:** Two sets of representations $X, Y$, distance functions $d_X, d_Y$, metric type $S \in \{\text{TSI}, \text{QSI}\}$, maximum additive error $\epsilon > 0$, and failure probability $\delta \in (0, 1)$.
1: $n \leftarrow \lceil \frac{1}{2\epsilon^2} \log(\frac{2}{\delta}) \rceil$
2: $A_{\text{total}} \leftarrow 0$
3: **for** $i = 1 \rightarrow n$ **do**
4:     **if** $S = \text{TSI}$ **then**
5:         Sample a triplet of distinct indices $(i, j, k)$ uniformly at random from $\mathcal{T}$.
6:         $A_{\text{total}} \leftarrow A_{\text{total}} + \mathbb{I}\big[O_{d_X}^{\Delta}(x_i, x_j, x_k) = O_{d_Y}^{\Delta}(y_i, y_j, y_k)\big]$
7:     **else**
8:         Sample a quadruplet of distinct indices $(i, j, k, l)$ uniformly at random from $\mathcal{Q}$.
9:         $A_{\text{total}} \leftarrow A_{\text{total}} + \mathbb{I}\big[O_{d_X}(x_i, x_j, x_k, x_l) = O_{d_Y}(y_i, y_j, y_k, y_l)\big]$
10:    **end if**
11: **end for**
12: **return**$A_{\text{total}}/n$

---

It is straightforward to see that computing this algorithm requires $\mathcal{O}\left(\frac{1}{\epsilon^2} \log\left(\frac{1}{\delta}\right)\right)$ time complexity and $\mathcal{O}(1)$ auxiliary space complexity.

**Corollary 4** (Computational Complexity of Approximate TSI and QSI). *By evaluating $\lceil \frac{1}{2\epsilon^2} \log(\frac{2}{\delta}) \rceil$ triplets or quadruplets sampled uniformly at random, we can estimate TSI or QSI with an additive error of at most $\epsilon$ with a probability of at least $1 - \delta$. This approximation has a time complexity of $\mathcal{O}\left(\frac{1}{\epsilon^2} \log\left(\frac{1}{\delta}\right)\right)$ and an auxiliary space complexity of $\mathcal{O}(1)$.*

*Proof.* Direct application of Lemma 3 by setting the number of samples $n = \lceil \frac{1}{2\epsilon^2} \log(\frac{2}{\delta}) \rceil$. The time complexity follows from evaluating the ordinal predicate for $n$ samples, and space complexity is $\tilde{\mathcal{O}}(1)$ as only the running sum needs to be stored. $\qquad\square$

Finally, it is possible to combine both the exact and approximate approaches by averaging the exact computation of TSI and QSI over a predetermined set of randomly selected batches. This approach can sometimes work well in practice, as it evaluates a much larger number of triplets/quadruplets for the same unit of computation. However, the theoretical guarantees are not as tight as the aforementioned approximation approaches due to the statistical dependence between the evaluated triplets/quadruplets. Therefore, we do not explore this in detail.

## I. Concentration Bounds of True Population Similarity as a Function of Dataset Size

As introduced in Section 3, representation sets $X$ and $Y$ are generated by applying mappings $f$ and $g$ to a common source set $Z$. In most practical applications, this set $Z$ can be viewed as being sampled from a source distribution $\mathcal{D}_Z$. Ultimately, the goal is not merely to measure the similarity of specific finite sets, but to assess the alignment of the underlying data-generating process itself. We can formalize this by treating the pair generation as sampling from a joint distribution $\mathcal{D}_{XY}$. This motivates the concept of *population similarity*: the expected metric value as the sample size grows to infinity, which directly quantifies the true alignment of the underlying generating processes.

**Definition 4** (Population Similarity). Let $\mathcal{D}_{XY}$ be a joint probability distribution over pairs of representations $(x, y)$ in spaces equipped with distance functions $d_X$ and $d_Y$. For a similarity metric $S$ evaluated on finite datasets of size $N$, we define the *population similarity* (or true similarity) as the expected value of the metric as the sample size approaches infinity:

$$S^*_{d_X, d_Y}(\mathcal{D}_{XY}) = \lim_{N \to +\infty} \mathbb{E}_{(X,Y) \sim \mathcal{D}^N_{XY}} \left[ S_{d_X, d_Y}(X, Y) \right]$$

A fundamental question in practice is how reliably a similarity score computed on finite representation sets $X$ and $Y$ estimates this true population similarity. Strikingly, the mathematical structure of TSI and QSI allows us to derive tight concentration bounds that depend solely on the dataset size.

**Lemma 4** (Concentration Bounds for Empirical TSI and QSI). *Let $(X, Y) \sim \mathcal{D}^N_{XY}$ be a finite dataset of $N$ independent pairs sampled from the joint distribution $\mathcal{D}_{XY}$. Because TSI and QSI are U-Statistics with bounded indicator kernels in $[0, 1]$ (of degrees $m = 3$ and $m = 4$, respectively), their empirical estimations tightly concentrate around their true population similarities $S^* \in \{TSI^*, QSI^*\}$. For any $\epsilon > 0$, the probabilities of deviation are bounded by:*

$$\mathbb{P}\left[ \left| TSI_{d_X, d_Y}(X, Y) - TSI^*_{d_X, d_Y}(\mathcal{D}_{XY}) \right| \geq \epsilon \right] \leq 2 \exp\left( -2 \lfloor \tfrac{N}{3} \rfloor \epsilon^2 \right)$$

$$\mathbb{P}\left[ \left| QSI_{d_X, d_Y}(X, Y) - QSI^*_{d_X, d_Y}(\mathcal{D}_{XY}) \right| \geq \epsilon \right] \leq 2 \exp\left( -2 \lfloor \tfrac{N}{4} \rfloor \epsilon^2 \right)$$

This lemma has significant practical implications: it guarantees that empirical TSI and QSI scores predictably and rapidly converge to their true population values as the dataset grows, providing researchers with mathematically bounded confidence in their measurements. While a comprehensive convergence analysis of all existing similarity metrics is beyond the scope of this work, we anticipate that other metrics either exhibit slower convergence rates or possess bounds that depend on the specific geometry of the underlying distributions, in contrast to the distribution-free, purely sample-dependent bounds of TSI and QSI.

## J. Experimental Setting

In this section, we describe the parameters used to define each of the similarity metrics in our experiments. This parameter choice reflects the implementation in the corresponding original work, as summarized in Table 2.

To run the external metrics, we utilized the code from their respective original work. This was achieved by utilizing the code repository proposed by Cloos et al. (2024), as this repository centralizes existing implementations of similarity metrics to facilitate comparisons across studies.

For the approximate computations of these metrics, following the parameter selection of other works (Huh et al., 2024), we use a batch size of $N_B = 1000$ for all external metrics, and sample $B$ batches with replacement (typically $B = 10$

*Table 2.* Parameter settings for the similarity metrics used in the experiments. Settings are chosen to align with the original implementations. The specified distance and kernel functions are the same for both representation spaces (i.e., $d_X = d_Y$ and $k_X = k_Y$).

| Metric | Distance Func. | Kernel Func. | Other Params | Original Work |
|--------|---------------|--------------|--------------|---------------|
| SVCCA | - | - | Var. explained $\geq 0.99$ | (Raghu et al., 2017) |
| PWCCA | - | - | - | (Morcos et al., 2018) |
| CKA | - | $z_1^\top z_2$ | - | (Kornblith et al., 2019) |
| MutualNN | $-z_1^\top z_2$ | - | $k = 10$ | (Huh et al., 2024) |
| CKNNA | $-z_1^\top z_2$ | $z_1^\top z_2$ | $k = 10$ | (Huh et al., 2024) |
| TSI (Ours) | $\|z_1 - z_2\|_2$ | - | - | This work |
| QSI (Ours) | $\|z_1 - z_2\|_2$ | - | - | This work |

unless specified otherwise). Specifically, when computing similarity metrics in the experiments of Sections 6.4 and 6.5, we utilized the custom CKA principled approximation from Nguyen et al. (2021) (see Section A.1), whereas for the other external metrics we simply use a default batch approach. For TSI and QSI, we do not use batches but rather sample $B \times N_B^2$ triplets/quadruplets and directly leverage Algorithm 4. This ensures a computationally comparable approximation to other similarity metrics that have quadratic time/space complexity.

## K. Comments on Experimental Outcomes of External Metrics

### K.1. Instability of SVCCA and PWCCA Computation

SVCCA and PWCCA require solving a CCA problem between two matrices, which involves estimating batch covariances and computing inverses (or inverse square root) via eigendecomposition/SVD of the sample covariances. In our setting, this step is particularly fragile in the batched approximation regime used for scalability evaluation. For $N_B$ high-dimensional representations with dimension $d$, the covariance estimates can be rank-deficient if $d > N_B$ (noting that after centering, the rank is at most $N_B - 1$), therefore SVCCA and PWCCA are not computed in this setting. However, even with $N_B \geq d$, strong collinearity and low effective rank can lead to challenges where small perturbations in the batch produce disproportionate numerical effects when performing CCA analysis, a phenomenon most frequent when $d$ is similar to $N_B$. Empirically, this shows up as non-deterministic behavior across random batches/seeds and occasional numerical failures (e.g., "SVD did not converge", "NaNs/Inf"), which prevents reliable averaging across batches and consistent approximation-quality evaluation. We therefore do not offer comparisons with SVCCA and PWCCA in Sections 6.3 to 6.5.

### K.2. Counter-Intuitive Sensitivity of MutualNN and CKNNA

The zero-like, low similarity scores of MutualNN and CKNNA reported in the outlier experiment (Section 6.2) may initially seem counter-intuitive. Since these metrics quantify alignment based on local neighborhood overlaps, perturbing a single point $x_i$ should theoretically have a bounded impact. Intuitively, this perturbation should only alter the neighborhood calculations for a constant number of points: $x_i$ itself, the set of points that lost $x_i$ as a neighbor, and the set of points that newly acquired $x_i$ as a neighbor. Typically, this is expected to constitute a constant number of points, resulting in a negligible expected score deviation of $\frac{1}{N} \times \mathcal{O}(1)$.

However, this intuition fails because standard implementations of these metrics typically utilize the negative dot product as a proxy for distance. Unlike Euclidean distance, the negative dot product does not satisfy the triangle inequality and does not have a lower bound. When an outlier is introduced with a large magnitude (as defined in our robustness protocol), it produces extreme dot product values across the entire dataset that in the limit, for large magnitude values, either converge to $-\infty$, $0$ or $+\infty$. Consequently, this single outlier can potentially create a Hubness effect, i.e., it invades the top-$k$ nearest neighbor lists of a vast number of points (approaching $\mathcal{O}(N)$), effectively displacing their true neighbors globally. This systemic disruption causes the similarity score to degrade disproportionately, resulting in a $\frac{1}{N} \times \mathcal{O}(N)$ impact.

This observation is empirically validated in Figure 10, where we utilize the same experimental setup as in the original outlier experiment (see Section 6.2) to compare the robustness of CKNNA and MutualNN using both dot product and Euclidean distance. As the outlier strength $\sigma$ increases, the similarity scores for the variants using dot product (standard in many

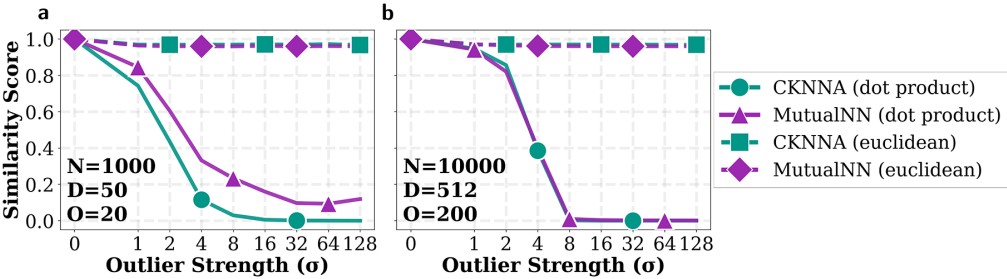

*Figure 10.* Robustness of local metrics to outliers when using different distance proxies. a) Synthetic representations ($N = 1000$, $D = 50$, $K = 20$), b) Vision Transformer representations on CIFAR-10 ($N = 10000$, $D = 512$, $K = 200$). While CKNNA and MutualNN implemented with the dot product (solid lines) drop to near-zero scores as outlier strength $\sigma$ increases, their Euclidean counterparts (dashed lines) remain robust to these perturbations.

implementations) drop sharply towards zero, reflecting the hubness-driven systemic disruption of local neighborhoods. In contrast, the versions of these metrics utilizing Euclidean distance remain robust, maintaining alignment scores near 1.0 even under severe perturbations. This confirms the aforementioned limitations associated with selecting distance proxies that do not satisfy the triangle inequality.

While replacing the negative dot product addresses the issue in certain practical settings, it does not provide theoretical guarantees of robustness. There are still scenarios where MutualNN and CKNNA remain dramatically affected by outliers. Furthermore, even with proper metrics, these methods suffer from fundamental structural limitations: they are blind to the relative ordering of neighbors within the $k$-neighborhood, and they are inherently limited to a single neighborhood scale ($k$). In contrast, unlike local metrics, the theoretical robustness bounds of TSI and QSI presented in Corollary 2 are applicable to all distance functions, even including the negative dot product, making them theoretically robust.

## L. Training Details of Vision Transformer on CIFAR-10

To evaluate the robustness and approximation qualities of the similarity metrics in a realistic setting, we utilized representations derived from a Vision Transformer (ViT) (Dosovitskiy et al., 2021) trained on the CIFAR-10 dataset (Krizhevsky, 2009). All similarity evaluations described in Sections 6.2 to 6.4 were computed using the representations of the 10,000 images comprising the official CIFAR-10 validation set.

The model architecture was adapted to accommodate the lower resolution of CIFAR-10 inputs ($32 \times 32$) by reducing the patch size. The specific architectural hyperparameters used for the experiments are detailed in Table 3.

*Table 3.* Architectural hyperparameters for the Vision Transformer trained on CIFAR-10.

| Parameter | Value |
| --- | --- |
| Input Resolution | $32 \times 32 \times 3$ |
| Patch Size | $4 \times 4$ |
| Latent Dimension ($D$) | 512 |
| Depth (Transformer Layers) | 6 |
| Attention Heads | 8 |
| MLP Dimension | 512 |
| Dropout Rate | 0.1 |
| Output Classes | 10 |

The model was trained for 200 epochs on a Tesla V100 GPU, requiring approximately 2 hours to complete. Figure 11 illustrates the validation accuracy progression over epochs, confirming that the representations used for analysis correspond to a well-optimized model capable of extracting semantic features.

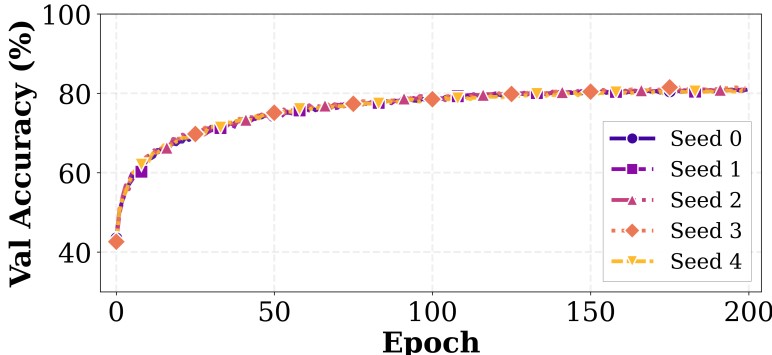

*Figure 11.* Epoch-wise Validation Accuracy of the ViT model trained on CIFAR-10.

## L.1. Details of Representations Used for Each Experiment

In the following, we explicitly define the source of the representations ($X$ and $Y$) used for the similarity computations in Sections 6.2 to 6.4. Crucially, for all experiments listed below, the representation sets are constructed by extracting the latent features, i.e., the corresponding [CLS] token representations, of the 10,000 images comprising the CIFAR-10 validation set.

*Table 4.* Configuration of representation sets used for each experimental evaluation.

| Experiment Section | Representation Set X | Representation Set Y |
|---|---|---|
| Outlier Robustness (Section 6.2) | Fully trained ViT (final epoch, Seed 0) | Fully trained ViT (final epoch, Seed 0) with synthetic outliers |
| Efficient Approximation (Section 6.3) | Untrained ViT (initial weights, Seed 0) | Fully trained ViT (final epoch, Seed 0) |
| Representation Convergence (Section 6.4) | Intermediate trained ViT (all epochs, Seed $\{0, 1, 2, 3, 4\}$) | Fully trained ViT (final epoch, Seed $\{0, 1, 2, 3, 4\}$) |

## M. Illustrative Examples

**Local Transformation Failure Mode of MutualNN and CKA Addressed by TSI and QSI**    To illustrate a setting where TSI and QSI successfully capture misalignment while MutualNN and CKA fail, we construct the following counterexample.

Let $X \subset \mathbb{R}^d$ be a dataset of $N = 2k$ representations equipped with the standard Euclidean distance ($d_X = \|\cdot\|_2$), partitioned into two distinct clusters $C_1$ and $C_2$, each containing $k \geq 5$ points. We assume that all pairwise distances within each cluster are unique (no ties). Furthermore, let the distance between the two clusters be arbitrarily large; that is, the inter-cluster distance $D = \min_{x \in C_1, y \in C_2} d(x, y)$ satisfies $D \to +\infty$.

Now, consider a transformed representation space $X'$ constructed by applying a permutation $\pi$ to the elements of cluster $C_1$ (i.e., $\pi(C_1) = C_1$ with points locally shuffled). We make the technical assumption that for this cluster, $\pi$ leaves at least one point unpermuted, i.e. $\pi(i) = i$, and permutes two pairs of points between them $\pi(j_1) = k_1 \wedge \pi(k_1) = j_1$ and $\pi(j_2) = k_2 \wedge \pi(k_2) = j_2$. This transformation introduces a severe local structural deformation while perfectly preserving the global cluster assignments.

We evaluate the impact of this local deformation on the respective metrics:

- **Mutual Nearest Neighbors (MutualNN$_{k-1, d_X, d_X}$):** Because the inter-cluster distance $D \to +\infty$, the $k - 1$ nearest neighbors of any point $x \in C_i$ are precisely the other $k - 1$ points in its own cluster $C_i$. Since the permutation $\pi$ only shuffles points within $C_1$, the set of intra-cluster nearest neighbors for each point is perfectly preserved. Therefore, the neighborhood overlap remains identical, yielding MutualNN$_{k-1}(X, X) = $ MutualNN$_{k-1}(X, X') = 1$.

- **Centered Kernel Alignment (CKA$_{d_X, d_X}$):** Following the definition from Section A, CKA first computes the pairwise distance matrices $K_X$ with $(K_X)_{ij} = d_X(x_i, x_j)$ and $K_{X'}$ analogously. It then computes the Hilbert-Schmidt Independence Criterion (HSIC) on the centered pairwise distance matrices: $\text{HSIC}(K_X, K_{X'}) = \frac{1}{(N-1)^2} \text{tr}(K_X H K_{X'} H)$, where $H = I_N - \frac{1}{N} J_N$ is the centering matrix. We show that as $D \to +\infty$, $\text{HSIC}(K_X, K_{X'}) \to \text{HSIC}(K_X, K_X)$ and $\text{HSIC}(K_{X'}, K_{X'}) \to \text{HSIC}(K_X, K_X)$, which implies $\text{CKA}(X, X') \to \text{CKA}(X, X) = 1$.

  To see this, consider the centered matrix $K_X H$. The right-centering operation effectively subtracts the row means from each entry. For any point, there are $k$ intra-cluster distances $d_{\text{intra}}$ such that $0 \leq d_{\text{intra}} \leq \epsilon$ for some small

$\epsilon > 0$, and $k$ inter-cluster distances $d_{\text{inter}}$ such that $D \leq d_{\text{inter}} \leq D + \epsilon$. Thus, the mean distance $\mu$ from any point to all other points satisfies $D/2 \leq \mu \leq D/2 + \epsilon$. Consequently, after centering, the centered intra-cluster distances $d_{\text{intra}} - \mu$ satisfy $-D/2 - \epsilon \leq d_{\text{intra}} - \mu \leq -D/2 + \epsilon$, and the centered inter-cluster distances $d_{\text{inter}} - \mu$ satisfy $D/2 - \epsilon \leq d_{\text{inter}} - \mu \leq D/2 + \epsilon$. Because this asymptotic behavior depends exclusively on the global cluster assignments, which are invariant under the intra-cluster permutation $\pi$, the entries of the centered distance matrices $K_X H$ and $K_{X'} H$ are identical up to some entry deviation of at most $2\epsilon$. Let us define the deviation matrix $\Delta$ such that $K_{X'} H = K_X H + \Delta$, where the entries of $\Delta$ are bounded by $2\epsilon$. Using the linearity of the trace operator, we can decompose $\text{HSIC}(K_X, K_{X'})$ and $\text{HSIC}(K_{X'}, K_{X'})$ in terms of $\text{HSIC}(K_X, K_X)$:

$$
\begin{aligned}
(N-1)^2 \text{HSIC}(K_X, K_{X'}) &= \text{tr}(K_X H K_{X'} H) \\
&= \text{tr}(K_X H (K_X H + \Delta)) \\
&= \text{tr}(K_X H K_X H) + \text{tr}(K_X H \Delta) \\
&= (N-1)^2 \text{HSIC}(K_X, K_X) + \text{tr}(K_X H \Delta)
\end{aligned}
$$

Similarly, since the right centered matrix can be written as $K_{X'} H = K_X H + \Delta$, we have:

$$
\begin{aligned}
(N-1)^2 \text{HSIC}(K_{X'}, K_{X'}) &= \text{tr}((K_{X'} H)^2) \\
&= \text{tr}((K_X H + \Delta)^2) \\
&= \text{tr}((K_X H)^2) + 2 \text{tr}(K_X H \Delta) + \text{tr}(\Delta^2) \\
&= (N-1)^2 \text{HSIC}(K_X, K_X) + 2 \text{tr}(K_X H \Delta) + \text{tr}(\Delta^2)
\end{aligned}
$$

Since the entries of the centered distance matrix $K_X H$ grow linearly with the inter-cluster distance $D$, the term $\text{HSIC}(K_X, K_X)$ is of the order $\mathcal{O}(D^2)$. In contrast, because the entries of $\Delta$ are bounded by $2\epsilon = \mathcal{O}(1)$, the cross-term $\text{tr}(K_X H \Delta)$ is $\mathcal{O}(D)$, and the residual term $\text{tr}(\Delta^2)$ is $\mathcal{O}(1)$. Therefore, as $D \to +\infty$, the $\mathcal{O}(D^2)$ terms dominate, causing both $\text{HSIC}(K_X, K_{X'})$ and $\text{HSIC}(K_{X'}, K_{X'})$ to converge to $\text{HSIC}(K_X, K_X)$.

- **Triplet Similarity Index (TSI$_{d_X, d_X}$):** Under our technical assumptions, we select the unpermuted anchor index $i$ (i.e., $\pi(i) = i$) and the permuted pair $j_1$ and $k_1$. Because all intra-cluster distances are unique (no ties), the original ordinal relationship between $j_1$ and $k_1$ evaluates to either 1 or $-1$ (i.e., $O_{d_X}^{\Delta}(x_i, x_{j_1}, x_{k_1}) \in \{1, -1\}$). Since the permutation swaps the points at indices $j_1$ and $k_1$, their distances from the unpermuted anchor $i$ are strictly exchanged: $d_X(x_i', x_{j_1}') = d_X(x_i, x_{k_1})$ and $d_X(x_i', x_{k_1}') = d_X(x_i, x_{j_1})$. This means the relative distance ordering from the anchor is inverted, so the new ordinal relationship must evaluate to the opposite sign. Because the ordinal value is necessarily different, we have $O_{d_X}^{\Delta}(x_i, x_{j_1}, x_{k_1}) \neq O_{d_X}^{\Delta}(x_i', x_{j_1}', x_{k_1}')$. The presence of this strictly discordant triplet guarantees that the expected agreement across all possible triplets is not perfect, thus $\text{TSI}_{d_X, d_X}(X, X') < 1$.

- **Quadruplet Similarity Index (QSI$_{d_X, d_X}$):** To show a discordant quadruplet, we select the four indices involved in the two swapped pairs: $j_1, j_2, k_1, k_2$. In the original space, the disjoint pairs $(j_1, j_2)$ and $(k_1, k_2)$ have unique distances, so their ordinal relationship evaluates to $O_{d_X}(x_{j_1}, x_{j_2}, x_{k_1}, x_{k_2}) \in \{1, -1\}$. After applying the permutation, the points at these indices are swapped: $x_{j_1}' = x_{k_1}, x_{j_2}' = x_{k_2}, x_{k_1}' = x_{j_1}$, and $x_{k_2}' = x_{j_2}$. Consequently, the new distance between indices $j_1$ and $j_2$ becomes the original distance between $k_1$ and $k_2$ (i.e., $d_X(x_{j_1}', x_{j_2}') = d_X(x_{k_1}, x_{k_2})$), and vice versa. This strictly flips the distance ordering between the two disjoint pairs, yielding a necessarily different opposite sign: $O_{d_X}(x_{j_1}, x_{j_2}, x_{k_1}, x_{k_2}) \neq O_{d_X}(x_{j_1}', x_{j_2}', x_{k_1}', x_{k_2}')$. This discordant quadruplet ensures that $\text{QSI}_{d_X, d_X}(X, X') < 1$.

This example demonstrates a critical failure mode: MutualNN and CKA can be completely insensitive to intra-cluster permutations. Consequently, points within a cluster can be arbitrarily shuffled, destroying local alignment without affecting these metrics. In contrast, both TSI and QSI successfully detect that the representations are no longer locally aligned.

**Global Transformation Failure Mode of MutualNN and CKNNA Addressed by TSI and QSI**   To illustrate a setting where TSI and QSI successfully capture misalignment while MutualNN and CKNNA fail to detect a massive global transformation, we construct the following counterexample.

Let $X \subset \mathbb{R}^d$ be a dataset of $N = 3k$ representations equipped with the standard Euclidean distance ($d_X = \|\cdot\|_2$), partitioned into three distinct clusters $C_1$, $C_2$, and $C_3$, each containing $k \geq 2$ points. Let $c_1, c_2$, and $c_3$ denote the respective centroids of these clusters. We assume the clusters are tightly packed such that intra-cluster distances are

arbitrarily small compared to inter-cluster distances. We define the inter-cluster distances as follows: $\|c_1 - c_2\|_2 \approx D$ and $\|c_1 - c_3\|_2 \approx R$, with $D \gg R \gg 1$. Furthermore, we assume that the vector $c_1 - c_2$ is orthogonal to $c_1 - c_3$, which implies that $\|c_2 - c_3\|_2 \approx \sqrt{D^2 + R^2}$.

Now, consider a transformed representation space $X'$ constructed by simply translating the entire third cluster $C_3$ by the vector $c_2 - c_1$, while leaving $C_1$ and $C_2$ unchanged. The new centroid for the third cluster becomes $c'_3 = c_3 + (c_2 - c_1)$. Consequently, the new inter-cluster distances become $\|c'_1 - c'_3\|_2 = \|c_1 - (c_3 + c_2 - c_1)\|_2 = \|(c_1 - c_3) + (c_1 - c_2)\|_2 \approx \sqrt{D^2 + R^2}$ (due to orthogonality), and $\|c'_2 - c'_3\|_2 = \|c_2 - (c_3 + c_2 - c_1)\|_2 = \|c_1 - c_3\|_2 \approx R$. The distance between $C_1$ and $C_2$ remains $\approx D$. This transformation represents a severe global structural change, significantly altering the macroscopic layout of the clusters.

We evaluate the impact of this global transformation on the respective metrics:

- **Mutual Nearest Neighbors (MutualNN$_{k-1, d_X, d_X}$):** Because $D \gg R \gg 1$ and the clusters are tightly packed, $k - 1$ nearest neighbors of any point $x \in C_i$ are exclusively other points within its own cluster $C_i$. Since the transformation only translates the entire cluster $C_3$ rigidly and preserves the large inter-cluster separations ($R \gg 1$), the intra-cluster nearest neighbors for all points remain perfectly intact. Thus, the neighborhood overlap is identical, yielding MutualNN$_{k-1}(X, X') = 1$.

- **Centered Kernel Nearest-Neighbor Alignmen (CKNNA$_{k-1, d_X, d_X}$):** CKNNA relies on the adjacency matrices of the $k$-nearest neighbor graphs. As established above, the $k - 1$-nearest neighbor graph connects only points within the same cluster. This local connectivity structure is completely invariant to the rigid translation of $C_3$. Consequently, the local adjacency matrices for $X$ and $X'$ are identical, leading to CKNNA$_{k-1}(X, X') = 1$ (analogous to the self-similarity CKA$(Z, Z) = 1$ when the underlying distance matrices are identical).

- **Triplet Similarity Index (TSI$_{d_X, d_X}$):** We select an anchor point $x_1 \in C_1$, and two target points $x_2 \in C_2$ and $x_3 \in C_3$. In the original space $X$, the distances from the anchor are $d_X(x_1, x_2) \approx D$ and $d_X(x_1, x_3) \approx R$. Since $D > R$, the ordinal relationship evaluates to $O_{d_X}^{\Delta}(x_1, x_2, x_3) = 1$. In the transformed space $X'$, the points map to $x'_1, x'_2, x'_3$. The new distances from the anchor become $d_X(x'_1, x'_2) \approx D$ and $d_X(x'_1, x'_3) \approx \sqrt{D^2 + R^2}$. Because $D < \sqrt{D^2 + R^2}$, the relative distance ordering is inverted, yielding the opposite sign: $O_{d_X}^{\Delta}(x'_1, x'_2, x'_3) = -1$. The existence of this strictly discordant triplet implies that the expected agreement across all triplets is not perfect, hence TSI$_{d_X, d_X}(X, X') < 1$.

- **Quadruplet Similarity Index (QSI$_{d_X, d_X}$):** We select two distinct points $x_{1a}, x_{1b} \in C_1$, one point $x_2 \in C_2$, and one point $x_3 \in C_3$. In the original space $X$, the distance between the disjoint pairs $(x_{1a}, x_2)$ and $(x_{1b}, x_3)$ are roughly $D$ and $R$, respectively. Since $D > R$, the ordinal relationship is $O_{d_X}(x_{1a}, x_2, x_{1b}, x_3) = 1$. In the transformed space $X'$, the distances become approximately $D$ and $\sqrt{D^2 + R^2}$. Since $D < \sqrt{D^2 + R^2}$, the distance ordering between the two pairs is flipped, yielding $O_{d_X}(x'_{1a}, x'_2, x'_{1b}, x'_3) = -1$. This discordant quadruplet ensures that QSI$_{d_X, d_X}(X, X') < 1$.

This example illustrates a fundamental limitation: metrics strictly based on $k$-nearest neighbor graphs, such as MutualNN and CKNNA, can be completely blind to massive global transformations as long as the local intra-cluster neighborhoods remain intact. In contrast, TSI and QSI evaluate distances at all scales and successfully identify the macroscopic misalignment.

**Distinguishing the Scope of TSI and QSI** To illustrate the difference in sensitivity between TSI and QSI, consider a 1-dimensional dataset containing $N = 4$ representations forming two distinct clusters: $X = \{x_1, x_2, x_3, x_4\} = \{0, 1, D, D + 2\}$, equipped with the standard Euclidean distance. Let $D \gg 1$ be an arbitrarily large positive constant. The representations can be conceptually grouped into two remote clusters: $\{x_1, x_2\}$ and $\{x_3, x_4\}$.

Now, consider a transformed dataset $X'$ obtained by applying a local contraction to the second cluster, specifically mapping $x_4 \rightarrow x'_4 = D + 1$. Thus, $X' = \{0, 1, D, D + 1\}$.

We can evaluate the impact of this local structural deformation on both metrics:

- **TSI:** Because TSI relies strictly on triplets (a shared anchor point), it only measures distance rankings relative to each individual point. Since the inter-cluster distance ($\approx D$) dominates any intra-cluster changes, the relative distance ordering from any single anchor point remains entirely unaffected. Therefore, TSI$_{d_X, d_X}(X, X') = 1$.

- **QSI:** QSI evaluates disjoint pairs of distances. In the original space $X$, the intra-cluster distance of the first cluster is smaller than the intra-cluster distance of the second cluster: $d_X(x_1, x_2) = 1 < 2 = d_X(x_3, x_4)$. However, in the

contracted space $X'$, this relationship turns equal: $d_X(x_1', x_2') = 1 = d_X(x_3', x_4')$. Because QSI captures this global relative scale variation between disjoint pairs, the metric penalizes the transformation, yielding $\text{QSI}_{d_X, d_X}(X, X') = \frac{1}{3}$ (the ordinal sign changes for $(1, 2, 3, 4)$ and $(1, 3, 2, 4)$ due to the contraction, but is preserved for $(1, 4, 2, 3)$, where each tuple $(i, j, k, l)$ denotes the disjoint-pair comparison $d(x_i, x_j)$ vs. $d(x_k, x_l)$).

This minimal example highlights that while TSI is highly robust to localized, non-isotropic scale variations provided the local ranking structure is preserved, QSI strictly enforces the preservation of global relative scale across disjoint regions of the representation space.

# N. Proofs

## N.1. Expected Similarity for Independent Representations

This proof derives the expected value of TSI and QSI for independently sampled representations (Lemma 1). The key insight is that for i.i.d. samples, the probability of one distance being larger than another is, by symmetry, equal to the probability of it being smaller.

**Lemma 1** (Expected Similarity for Independent Representations). *Let $X = \{x_i\}_{i=1}^N$ and $Y = \{y_i\}_{i=1}^N$ be two datasets, with points drawn i.i.d. from arbitrary and independent distributions, $\mathcal{D}_X$ and $\mathcal{D}_Y$, respectively, and equipped with distance functions $d_X$ and $d_Y$. This setting ensures that the ordinal comparisons in $X$ and $Y$ are independent. For TSI, let $a, b, c$ be points drawn i.i.d. from $\mathcal{D}_X$. We define the probability of a tie as $p_X^0 = P_{a,b,c\sim\mathcal{D}_X}\big(d_X(a, b) = d_X(a, c)\big)$. Similarly for $\mathcal{D}_Y$. The expected TSI is:*

$$\mathbb{E}_{\substack{X\sim\mathcal{D}_X^N \\ Y\sim\mathcal{D}_Y^N}}[TSI_{d_X, d_Y}(X, Y)] = p_X^0 p_Y^0 + \frac{1}{2}(1 - p_X^0)(1 - p_Y^0)$$

*For QSI, let $a, b, c, d$ be points drawn i.i.d. from $\mathcal{D}_X$. We define the probability of a tie as $p_X^0 = P_{a,b,c,d\sim\mathcal{D}_X}\big(d_X(a, b) = d_X(c, d)\big)$. Similarly for $\mathcal{D}_Y$. The expected QSI is:*

$$\mathbb{E}_{\substack{X\sim\mathcal{D}_X^N \\ Y\sim\mathcal{D}_Y^N}}\big[QSI_{d_X, d_Y}(X, Y)\big] = p_X^0 p_Y^0 + \frac{1}{2}(1 - p_X^0)(1 - p_Y^0)$$

*Proof.* The proof for TSI is presented here; the argument for QSI is identical, simply replacing triplets with quadruplets and $O_d^\triangle$ with $O_d$.

From its probabilistic interpretation (Equation (1)), the TSI over two representation sets is the probability that the ordinal comparison agrees for a single random triplet drawn from the underlying distributions. Let $O_X = O_{d_X}^\triangle(x_i, x_j, x_k)$ and $O_Y = O_{d_Y}^\triangle(y_i, y_j, y_k)$ for randomly drawn points. The TSI is $P(O_X = O_Y)$.

We can decompose this probability by conditioning on the possible outcomes $\{-1, 0, 1\}$ for the sign comparison in each space:

$$P(O_X = O_Y) = \sum_{v\in\{-1,0,1\}} P(O_X = v \wedge O_Y = v)$$

Since the datasets $X$ and $Y$ are drawn from independent distributions, the outcomes of their ordinal comparisons are independent. Thus, we can factor the joint probabilities:

$$P(O_X = O_Y) = P(O_X = -1)P(O_Y = -1) + P(O_X = 0)P(O_Y = 0) + P(O_X = 1)P(O_Y = 1)$$

Let's analyze the probabilities for a single space, say $X$. We denote $p_X^- = P(O_X = -1)$, $p_X^0 = P(O_X = 0)$, and $p_X^+ = P(O_X = 1)$. By definition, $p_X^0$ is the probability of a tie, as defined in the lemma.

Now, w.l.o.g consider the event $O_X = -1$, which corresponds to $d_X(x_i, x_j) < d_X(x_i, x_k)$. Since the points $x_j$ and $x_k$ are drawn i.i.d. from the same distribution $\mathcal{D}_X$, their labels are interchangeable. Swapping the roles of $x_j$ and $x_k$ gives the event $d_X(x_i, x_k) < d_X(x_i, x_j)$, which corresponds to $O_X = 1$. This means that for any arbitrary triplet, there exists an equally probable permutation that transforms $O_X = -1$ to $O_X = 1$, therefore, the probabilities of these two events must be equal:

$$p_X^- = P\big(d_X(x_i, x_j) < d_X(x_i, x_k)\big) = P\big(d_X(x_i, x_k) < d_X(x_i, x_j)\big) = p_X^+$$

Since the probabilities of all outcomes must sum to 1, we have $p_X^- + p_X^0 + p_X^+ = 1$. Substituting $p_X^- = p_X^+$, we get $2p_X^+ + p_X^0 = 1$, which implies:

$$p_X^+ = p_X^- = \frac{1 - p_X^0}{2}$$

The same logic applies to space $Y$, so $p_Y^+ = p_Y^- = \frac{1-p_Y^0}{2}$.

Finally, we substitute these back into the expression for the expected TSI:

$$\begin{aligned}
\mathbb{E}[\text{TSI}] &= (p_X^-)(p_Y^-) + (p_X^0)(p_Y^0) + (p_X^+)(p_Y^+) \\
&= \left(\frac{1 - p_X^0}{2}\right)\left(\frac{1 - p_Y^0}{2}\right) + p_X^0 p_Y^0 + \left(\frac{1 - p_X^0}{2}\right)\left(\frac{1 - p_Y^0}{2}\right) \\
&= 2 \cdot \frac{(1 - p_X^0)(1 - p_Y^0)}{4} + p_X^0 p_Y^0 \\
&= \frac{(1 - p_X^0)(1 - p_Y^0)}{2} + p_X^0 p_Y^0
\end{aligned}$$

This completes the proof. $\qquad\square$

### N.2. Connection to Neighborhood Consistency

This section provides the formal proofs for the relationship between TSI and local neighborhood structures. We first establish the equivalence between perfect TSI alignment and joint Mutual Nearest Neighbors (MNN) consistency across all scales (Corollary 1). Subsequently, we prove that the TSI alignment score is upper-bounded by a convex combination of individual MNN coefficients (Corollary 9). The proof strategy relies on reducing these global relationships to independent point-wise comparisons for each anchor point, leveraging the shared ordinal structure between TSI and neighborhood-based similarity metrics.

#### N.2.1. PERFECT TSI ALIGNMENT IS EQUIVALENT TO JOINT MUTUAL NEAREST NEIGHBORS ALIGNMENT

**Corollary 1** (Perfect TSI alignment is equivalent to joint Mutual Nearest Neighbors alignment). *Let $(X, d_X)$ and $(Y, d_Y)$ be two sets of $N$ representations without ties. Then, full alignment of the TSI is equivalent to joint alignment of Mutual Nearest Neighbors for all $k$-nearest neighbors with $k$ ranging from 1 to $N - 2$:*

$$\begin{aligned}
\text{TSI}_{d_X, d_Y}(X, Y) = 1 &\iff \\
\forall k \in \{1, \ldots, N - 2\} &: \text{MutualNN}_{k, d_X, d_Y}(X, Y) = 1
\end{aligned}$$

*where $\text{MutualNN}_{k, d_X, d_Y}(X, Y)$ is the Mutual Nearest Neighbors score for $k$ neighbors.*

*Proof.* The ( $\implies$ ) direction is straightforward: $\text{TSI}_{d_X, d_Y}(X, Y) = 1$ implies that for every anchor point $i$, the relative ordering of distances to all other points is identical in both spaces. This ensures that $\text{KNN}_{k, d_X}(X \setminus \{x_i\}, x_i) = \text{KNN}_{k, d_Y}(Y \setminus \{y_i\}, y_i)$ for any $k$, yielding $\text{MutualNN}_{k, d_X, d_Y}(X, Y) = 1$.

For the ( $\impliedby$ ) direction, if $\text{MutualNN}_{k, d_X, d_Y}(X, Y) = 1$ for all $k \in \{1, \ldots, N - 2\}$, then $|\text{SharedNN}_{k, d_X, d_Y}(i)| = k$ for all $i$ and $k$, which implies $\text{KNN}_{k, d_X}(X \setminus \{x_i\}, x_i) = \text{KNN}_{k, d_Y}(Y \setminus \{y_i\}, y_i)$. Let $v_k^{d_X}(i)$ and $v_k^{d_Y}(i)$ be the unique $k$-th nearest neighbors of point $i$ in each space. Since $\{v_k^{d_X}(i)\} = \text{KNN}_{k, d_X}(X \setminus \{x_i\}, x_i) \setminus \text{KNN}_{k-1, d_X}(X \setminus \{x_i\}, x_i)$ (and similarly for $Y$), it follows that $v_k^{d_X}(i) = v_k^{d_Y}(i)$ for all $k \in \{1, \ldots, N - 1\}$. Thus, the distance rank-ordering from any anchor is preserved, which is equivalent to $\text{TSI}_{d_X, d_Y}(X, Y) = 1$. $\qquad\square$

#### N.2.2. TSI ALIGNMENT IS UPPER-BOUNDED BY CONVEX COMBINATION OF MUTUAL NEAREST NEIGHBORS COEFFICIENTS

**Corollary 9** (TSI alignment is upper-bounded by a convex combination of Mutual Nearest Neighbors coefficients). *Let $(X, d_X)$ and $(Y, d_Y)$ be two sets of $N$ representations without ties, i.e., $\forall i, j, k \in \{1, \ldots, N\}, d_X(x_i, x_j) =$*

$d_X(x_i, x_k) \implies j = k$ *(same for $Y$). Then, the TSI alignment is upper-bounded by a convex combination of Mutual Nearest Neighbors coefficients:*

$$\mathit{TSI}_{d_X, d_Y}(X, Y) \leq \frac{1}{\binom{N-1}{2}} \sum_{k=1}^{N-2} k \cdot \mathit{MutualNN}_{k, d_X, d_Y}(X, Y)$$

*where $\mathit{MutualNN}_{k, d_X, d_Y}(X, Y)$ is the Mutual Nearest Neighbors score for $k$ neighbors as defined in Section A.*

*Proof.* The proof proceeds by showing that the global inequality,

$$\mathrm{TSI}_{d_X, d_Y}(X, Y) \leq \frac{1}{\binom{N-1}{2}} \sum_{k=1}^{N-2} k \cdot \mathrm{MutualNN}_{k, d_X, d_Y}(X, Y), \tag{6}$$

can be reduced to a simpler point-wise comparison. We first express both sides as averages over individual data points $i \in \{1, \ldots, N\}$. Using the anchor-based formulation of TSI (Corollary 10) and the definition of Mutual Nearest Neighbors (Section A), the desired inequality is equivalent to:

$$\frac{1}{N} \sum_{i=1}^{N} \text{Adj-Kendall-}\tau(D_X^i, D_Y^i) \leq \frac{1}{\binom{N-1}{2}} \sum_{k=1}^{N-2} k \cdot \left( \frac{1}{N} \sum_{i=1}^{N} \frac{|\text{SharedNN}_{k, d_X, d_Y}(i)|}{k} \right)$$

$$\iff \frac{1}{N} \sum_{i=1}^{N} \text{Adj-Kendall-}\tau(D_X^i, D_Y^i) \leq \frac{1}{N} \sum_{i=1}^{N} \left( \frac{1}{\binom{N-1}{2}} \sum_{k=1}^{N-2} |\text{SharedNN}_{k, d_X, d_Y}(i)| \right).$$

By linearity of summation, this global condition holds if the following point-wise inequality is satisfied for every $i$:

$$\text{Adj-Kendall-}\tau(D_X^i, D_Y^i) \overset{?}{\leq} \frac{1}{\binom{N-1}{2}} \sum_{k=1}^{N-2} |\text{SharedNN}_{k, d_X, d_Y}(i)|. \tag{7}$$

Establishing this point-wise comparison thus completes the reduction.

To establish this, let $\sigma$ be the permutation that sorts the distances in $D_X^i$ increasingly, such that $d_X(x_i, x_{\sigma(1)}) < d_X(x_i, x_{\sigma(2)}) < \cdots < d_X(x_i, x_{\sigma(N-1)})$. The existence of a unique such permutation is guaranteed by the no-ties assumption. We define the ordinal rank of a point $j$ relative to $i$ as $r(j, D^i) = N - 1 - t$, where $j$ is the $t$-th closest neighbor to $i$ with respect to distances $D^i$. By construction, $r(\sigma(j), D_X^i) = N - 1 - j$.

With these definitions, and leveraging the no-ties assumption to ensure the sets $\text{SharedNN}_k$ are well-defined as having exactly $k$ elements, we can rewrite the weighted sum of shared neighbors as:

$$\sum_{k=1}^{N-2} |\text{SharedNN}_{k, d_X, d_Y}(i)| = \sum_{k=1}^{N-2} \sum_{j=1}^{N-1} \mathbb{I}[j \in \text{SharedNN}_{k, d_X, d_Y}(i)]$$

$$= \sum_{j=1}^{N-1} \min(r(j, D_X^i), r(j, D_Y^i))$$

$$= \sum_{j=1}^{N-1} \min(r(\sigma(j), D_X^i), r(\sigma(j), D_Y^i))$$

$$= \sum_{j=1}^{N-2} \min(r(\sigma(j), D_X^i), r(\sigma(j), D_Y^i))$$

Observe that for $j = N - 1$, we have $r(\sigma(N - 1), D_X^i) = 0$, so the $(N-1)$-th term in this sum is always zero.

Next, we simplify the Adjusted Kendall-$\tau$ using the identity from Lemma 6. By indexing the sum over the permutation $\sigma$

and leveraging the symmetry of the sign function, we obtain:

$$
\begin{aligned}
\text{Adj-Kendall-}\tau(D_X^i, D_Y^i) &= \frac{1}{(N-1)(N-2)} \sum_{j \neq t} \mathbb{I}\big[\text{sign}(d_X(x_i,x_j) - d_X(x_i,x_t)) = \text{sign}(d_Y(y_i,y_j) - d_Y(y_i,y_t))\big] \\
&= \frac{1}{(N-1)(N-2)} \sum_{j \neq t} \mathbb{I}\big[\text{sign}(d_X(x_i,x_{\sigma(j)}) - d_X(x_i,x_{\sigma(t)})) = \text{sign}(d_Y(y_i,y_{\sigma(j)}) - d_Y(y_i,y_{\sigma(t)}))\big] \\
&= \frac{1}{\binom{N-1}{2}} \sum_{j=1}^{N-2} \sum_{t=j+1}^{N-1} \mathbb{I}\big[\text{sign}(d_X(x_i,x_{\sigma(j)}) - d_X(x_i,x_{\sigma(t)})) = \text{sign}(d_Y(y_i,y_{\sigma(j)}) - d_Y(y_i,y_{\sigma(t)}))\big] \\
&= \frac{1}{\binom{N-1}{2}} \sum_{j=1}^{N-2} \sum_{t=j+1}^{N-1} \mathbb{I}\big[d_Y(y_i,y_{\sigma(j)}) < d_Y(y_i,y_{\sigma(t)})\big],
\end{aligned}
$$

where the last equality follows because $d_X(x_i, x_{\sigma(j)}) < d_X(x_i, x_{\sigma(t)})$ for all $j < t$ by the definition of $\sigma$ and the no-ties assumption. Substituting these reformulated terms into our point-wise comparison, we want to show that:

$$
\frac{1}{\binom{N-1}{2}} \sum_{j=1}^{N-2} \sum_{t=j+1}^{N-1} \mathbb{I}\big[d_Y(y_i,y_{\sigma(j)}) < d_Y(y_i,y_{\sigma(t)})\big] \overset{?}{\leq} \frac{1}{\binom{N-1}{2}} \sum_{j=1}^{N-2} \min(r(\sigma(j), D_X^i), r(\sigma(j), D_Y^i)). \tag{8}
$$

Therefore, to prove the corollary, it suffices to show that for each $j \in \{1, \dots, N-2\}$:

$$
\sum_{t=j+1}^{N-1} \mathbb{I}\big[d_Y(y_i,y_{\sigma(j)}) < d_Y(y_i,y_{\sigma(t)})\big] \overset{?}{\leq} \min(r(\sigma(j), D_X^i), r(\sigma(j), D_Y^i)). \tag{9}
$$

To do this, we establish two upper bounds on the sum. First, by the definition of the sum, we have:

$$
\sum_{t=j+1}^{N-1} \mathbb{I}\big[d_Y(y_i,y_{\sigma(j)}) < d_Y(y_i,y_{\sigma(t)})\big] \leq \sum_{t=j+1}^{N-1} 1 = N - j - 1 = r(\sigma(j), D_X^i).
$$

Second, we observe that the sum is also bounded by the total number of points $y_{\sigma(t)}$ that are further from $y_i$ than $y_{\sigma(j)}$:

$$
\sum_{t=j+1}^{N-1} \mathbb{I}\big[d_Y(y_i,y_{\sigma(j)}) < d_Y(y_i,y_{\sigma(t)})\big] \leq \sum_{t=1, t \neq j}^{N-1} \mathbb{I}\big[d_Y(y_i,y_{\sigma(j)}) < d_Y(y_i,y_{\sigma(t)})\big] = r(\sigma(j), D_Y^i).
$$

Since both bounds hold, the term-wise inequality in Equation (9) is satisfied for all $j \in \{1, \dots, N-2\}$. By summing over $j$, this directly implies Equation (8), which establishes the point-wise comparison in Equation (7). Finally, by averaging over all anchor points $i$, we obtain the global inequality in Equation (6), completing the proof.

$\square$

## N.3. Outlier Robustness

This subsection establishes the theoretical guarantees for the robustness of TSI and QSI to both representation corruptions and outliers. We first derive tight bounds for ordinal similarity under partial transformations (Lemma 2) using a combinatorial approach that partitions comparisons based on their involvement with transformed points. We then leverage these bounds to formally prove the robustness of our metrics (Corollary 2), demonstrating that their sensitivity to arbitrary modifications or sparse outliers is inherently bounded.

### N.3.1. BOUNDS FOR ORDINAL SIMILARITY METRICS UNDER PARTIAL TRANSFORMATIONS

**Lemma 2** (Bounds for Ordinal Similarity Metrics under Partial Transformations). *Let the set of indices $\{1, \dots, N\}$ be partitioned into two disjoint subsets, $U$ (unchanged) and $V$ (transformed). Let $T : \mathcal{X} \to \mathcal{X}$ be a transformation, and define a new dataset $X' = \{x_i'\}_{i=1}^N$ where $x_i' = x_i$ for $i \in U$ and $x_i' = T(x_i)$ for $i \in V$. For similarity scores $S \in \{TSI, QSI\}$, the score of the transformed dataset, $S(X', Y)$, is bounded by the score on the unchanged subset, $S(X_U, Y_U)$, as follows:*

$$
c_S \cdot S(X_U, Y_U) \leq S(X', Y) \leq c_S \cdot S(X_U, Y_U) + (1 - c_S)
$$

*where $c_S$ is the proportion of comparisons drawn exclusively from the unchanged set $U$: $c_{TSI} = \frac{(|U|)_3}{(N)_3}$ and $c_{QSI} = \frac{(|U|)_4}{(N)_4}$, where $(n)_k = n(n-1)\cdots(n-k+1)$ denotes the falling factorial.*

*Proof.* Let's first present the proof for TSI, as the argument for QSI follows the exact same structure. By definition, the TSI for the transformed dataset $X'$ is:

$$\text{TSI}_{d_X, d_Y}(X', Y) = \frac{1}{(N)_3} \sum_{(i,j,k) \in \mathcal{T}} \mathbb{I}\Big[O_{d_X}^{\Delta}(x_i', x_j', x_k') = O_{d_Y}^{\Delta}(y_i, y_j, y_k)\Big]$$

We partition the sum based on whether all indices $\{i, j, k\}$ belong to the untransformed set $U$. Let $\mathcal{T}_U$ be the set of triplets where all indices are in $U$.

$$(N)_3 \cdot \text{TSI}_{d_X, d_Y}(X', Y) = \sum_{(i,j,k) \in \mathcal{T}_U} \mathbb{I}\Big[\ldots\Big] + \sum_{(i,j,k) \in \mathcal{T} \setminus \mathcal{T}_U} \mathbb{I}\Big[\ldots\Big]$$

For any triplet $(i, j, k) \in \mathcal{T}_U$, the points are unchanged ($x_i' = x_i, x_j' = x_j, x_k' = x_k$). Therefore, the ordinal comparison is identical to that of the original dataset:

$$O_{d_X}^{\Delta}(x_i', x_j', x_k') = O_{d_X}^{\Delta}(x_i, x_j, x_k)$$

The first sum is thus the number of aligned triplets within the subset $U$, which by definition is:

$$\sum_{(i,j,k) \in \mathcal{T}_U} \mathbb{I}\Big[O_{d_X}^{\Delta}(x_i, x_j, x_k) = O_{d_Y}^{\Delta}(y_i, y_j, y_k)\Big] = (|U|)_3 \cdot \text{TSI}_{d_X, d_Y}(X_U, Y_U)$$

For the second sum over triplets with at least one transformed point, we can establish bounds. Since the indicator function $\mathbb{I}[\cdot]$ is always non-negative, a lower bound for this sum is 0. This gives us:

$$(N)_3 \cdot \text{TSI}_{d_X, d_Y}(X', Y) \geq (|U|)_3 \cdot \text{TSI}_{d_X, d_Y}(X_U, Y_U)$$

Dividing by $(N)_3$ gives the lower bound of the lemma.

For the upper bound, we use the fact that any indicator is at most 1. The second sum is therefore bounded by the number of triplets in its domain, which is $|\mathcal{T} \setminus \mathcal{T}_U| = (N)_3 - (|U|)_3$.

$$(N)_3 \cdot \text{TSI}_{d_X, d_Y}(X', Y) \leq (|U|)_3 \cdot \text{TSI}_{d_X, d_Y}(X_U, Y_U) + ((N)_3 - (|U|)_3)$$

Dividing by $(N)_3$ yields the upper bound:

$$\begin{aligned}
\text{TSI}_{d_X, d_Y}(X', Y) &\leq \frac{(|U|)_3}{(N)_3} \text{TSI}_{d_X, d_Y}(X_U, Y_U) + 1 - \frac{(|U|)_3}{(N)_3} \\
&= c_{\text{TSI}} \cdot \text{TSI}_{d_X, d_Y}(X_U, Y_U) + (1 - c_{\text{TSI}})
\end{aligned}$$

The same derivation holds for QSI by replacing the set of triplets $\mathcal{T}$ with the set of quadruplets $\mathcal{Q}$, the falling factorial $(n)_3$ with $(n)_4$ for both $n = N$ and $n = |U|$, and the triplet comparison function $O_d^{\Delta}$ with the quadruplet comparison function $O_d$. This yields the stated bounds for QSI and completes the proof. $\square$

### N.3.2. TSI AND QSI ARE ROBUST TO CORRUPTIONS AND OUTLIERS

**Corollary 2** (TSI and QSI are Robust to Corruptions and Outliers). *Consider the setup of Lemma 2 where $U$ indices the subset of clean or unchanged points and $V$ corresponds to the subset of corruptions/outliers. For any similarity metric $S \in \{TSI, QSI\}$, the absolute deviation of the similarity score is bounded by:*

$$\begin{aligned}
|S(X_U, Y_U) - S(X', Y)| &\leq 1 - c_S \quad \text{(Outliers)} \\
|S(X, Y) - S(X', Y)| &\leq 1 - c_S \quad \text{(Corruptions)}
\end{aligned}$$

*Proof.* Both inequalities follow from the bounds established in Lemma 2. Let $I$ be the interval $[c_S S(X_U, Y_U), c_S S(X_U, Y_U) + (1 - c_S)]$, which has length $\ell(I) = 1 - c_S$.

For the **Corruptions** bound, we observe that both the original dataset $X$ and the corrupted dataset $X'$ result from a transformation of the subset $V$ (where $X$ uses the identity transformation). By Lemma 2, both scores must lie within $I$:

$$S(X, Y) \in [c_S S(X_U, Y_U), c_S S(X_U, Y_U) + (1 - c_S)]$$
$$S(X', Y) \in [c_S S(X_U, Y_U), c_S S(X_U, Y_U) + (1 - c_S)]$$

Since both values belong to the same interval of length $1 - c_S$, their absolute difference is bounded by the interval length:

$$|S(X, Y) - S(X', Y)| \leq 1 - c_S.$$

For the **Outliers** bound, subtracting $S(X_U, Y_U)$ from the bounds in Lemma 2 yields:

$$(c_S - 1)S(X_U, Y_U) \leq S(X', Y) - S(X_U, Y_U) \leq (c_S - 1)S(X_U, Y_U) + (1 - c_S).$$

Since $0 \leq S(X_U, Y_U) \leq 1$ and $(c_S - 1) \leq 0$, we have $(c_S - 1)S(X_U, Y_U) \geq c_S - 1 = -(1 - c_S)$ for the lower bound, and $(c_S - 1)S(X_U, Y_U) + (1 - c_S) \leq 1 - c_S$ for the upper bound. This leads to:

$$-(1 - c_S) \leq S(X', Y) - S(X_U, Y_U) \leq 1 - c_S.$$

As the deviation is contained within $[-(1 - c_S), 1 - c_S]$, it follows that $|S(X_U, Y_U) - S(X', Y)| \leq 1 - c_S.$ $\qquad\square$

### N.4. Approximation Bounds for Ordinal Similarity Metrics

This proof establishes the approximation bounds presented in Lemma 3 for both TSI and QSI. The key observation is that evaluating the predicate of TSI/QSI for a randomly sampled triplet/quadruplet is equivalent to sampling directly from a Bernoulli distribution with the underlying expectation equating to the true TSI and QSI outputs for the considered representations.

**Lemma 3** (Approximation Bounds for Ordinal Similarity Metrics). *Let $\widehat{TSI}_n$ be the estimator for TSI obtained by sampling $n$ triplets uniformly at random with replacement from the set $\mathcal{T}$ and calculating the fraction of samples where the ordinal relationship is preserved. Similarly, let $\widehat{QSI}_n$ be the analogous estimator for QSI from $n$ random quadruplets. For any error tolerance $\epsilon > 0$, the estimators are bounded as follows:*

$$P\left( \left| \widehat{TSI}_n - TSI_{d_X, d_Y}(X, Y) \right| \geq \epsilon \right) \leq 2 \exp\left( -2n\epsilon^2 \right)$$

$$P\left( \left| \widehat{QSI}_n - QSI_{d_X, d_Y}(X, Y) \right| \geq \epsilon \right) \leq 2 \exp\left( -2n\epsilon^2 \right)$$

*Proof.* The proof for TSI is presented here; the argument for QSI is identical. The TSI is defined as the expectation of an indicator function over all possible triplets, and our estimator, $\widehat{TSI}_n$, is the sample mean of $n$ independent random draws from this set.

To formalize this, we define a Bernoulli random variable $Z$ for a single triplet $(i, j, k)$ sampled uniformly from $\mathcal{T}$. Let $Z = 1$ if the ordinal comparison agrees between the two spaces and $Z = 0$ otherwise:

$$Z = \mathbb{I}\big[ O^\Delta_{d_X}(x_i, x_j, x_k) = O^\Delta_{d_Y}(y_i, y_j, y_k) \big]$$

By definition, $Z$ is a Bernoulli random variable. Its expected value is the probability of success, which is precisely the true TSI:

$$\mathbb{E}[Z] = P(Z = 1) = P_{(i,j,k) \in \mathcal{T}}\left( O^\Delta_{d_X}(x_i, x_j, x_k) = O^\Delta_{d_Y}(y_i, y_j, y_k) \right) = TSI_{d_X, d_Y}(X, Y)$$

Our estimator $\widehat{TSI}_n$ is the sample mean of $n$ independent and identically distributed random variables $Z_1, \ldots, Z_n$ drawn from the distribution of $Z$:

$$\widehat{TSI}_n = \frac{1}{n} \sum_{t=1}^{n} Z_t$$

The expected value of the estimator is the true TSI: $\mathbb{E}[\widehat{\text{TSI}}_n] = \frac{1}{n}\sum_{t=1}^{n}\mathbb{E}[Z_t] = \mathbb{E}[Z] = \text{TSI}_{d_X,d_Y}(X,Y)$.

We can now apply Hoeffding's inequality (Hoeffding, 1963), which provides a bound on the probability that a sum of bounded independent random variables deviates from its expected value. For a sample mean $\bar{S}_n = \frac{1}{n}\sum Z_t$ where $Z_t \in [a,b]$, the inequality states:

$$P(|\bar{S}_n - \mathbb{E}[\bar{S}_n]| \geq \epsilon) \leq 2\exp\left(-\frac{2n\epsilon^2}{(b-a)^2}\right)$$

In our case, the random variables are bounded between $a = 0$ and $b = 1$, which means $(b-a)^2 = 1$. Substituting our estimator and its expected value into the inequality gives the desired result:

$$P\left(\left|\widehat{\text{TSI}}_n - \text{TSI}_{d_X,d_Y}(X,Y)\right| \geq \epsilon\right) \leq 2\exp\left(-2n\epsilon^2\right)$$

This completes the proof. $\qquad\square$

## N.5. Invariant Transformations for Ordinal Similarity Metrics

This proof establishes the invariance properties of ordinal similarity metrics, as stated in Corollary 5. The invariances for translation, isotropic scaling, and orthogonal transformations are straightforward. The non-invariance to arbitrary invertible linear transforms is proven by a counterexample where a non-isotropic scaling alters an existing ordinal relationship.

**Corollary 5** (Ordinal Similarity Metric Invariances in Inner Product Spaces). *Let $(\mathcal{X}, \langle\cdot,\cdot\rangle)$ be an inner product space with its induced metric $d_X$, then TSI and QSI are invariant to the following transformations:*

1. **Translation:** $T(x) = x + c$, *for any constant vector $c \in \mathcal{X}$.*

2. **Isotropic Scaling:** $T(x) = \alpha x$, *for any non-zero scalar $\alpha \in \mathbb{R}$.*

3. **Orthogonal Transformation:** *A linear map $T$ that preserves the inner product, i.e., $\langle T(x_1), T(x_2)\rangle = \langle x_1, x_2\rangle$ for all $x_1, x_2 \in \mathcal{X}$.*

*Furthermore, TSI and QSI are **not** generally invariant to:*

1. **Invertible Linear Transformations:** *A transformation $T(x) = Ax$, where $A$ is a linear operator for which an inverse $A^{-1}$ exists such that $AA^{-1} = A^{-1}A = I$.*

*Proof.* We first establish an auxiliary lemma that provides sufficient conditions for a transformation to leave our ordinal similarity metrics unchanged.

**Lemma 5** (Sufficient Conditions for Ordinal Similarity Invariance). *Let $T : \mathcal{X} \to \mathcal{X}$ be a transformation. If there exists a strictly increasing function $S : \mathbb{R}_0^+ \to \mathbb{R}_0^+$ such that for all $x_i, x_j \in X$, the transformed distance is given by $d_X(T(x_i), T(x_j)) = S(d_X(x_i, x_j))$, then TSI and QSI are invariant to $T$.*

*Proof.* The calculation of both TSI and QSI depends on the sign of a difference between two distances, i.e., $\text{sign}(d_1 - d_2)$. We show that this term remains unchanged after applying the transformation $T$. Since $S$ is strictly increasing, for any $a, b \in \mathbb{R}_0^+$, we have $a > b \iff S(a) > S(b)$. This implies that the sign of the difference is preserved:

$$\text{sign}(a - b) = \text{sign}(S(a) - S(b))$$

For TSI, the comparison is $\text{sign}(d_X(x_i, x_j) - d_X(x_i, x_k))$. For QSI, it is $\text{sign}(d_X(x_i, x_j) - d_X(x_k, x_l))$. In both cases, applying the transformation $T$ yields a comparison of the form $\text{sign}(S(d_1) - S(d_2))$, which is equal to the original sign, $\text{sign}(d_1 - d_2)$.

Since the predicate inside the expectation of the TSI and QSI definitions remains unchanged for every triplet or quadruplet, the expectation itself is unchanged. Thus, the metrics are invariant to $T$. $\qquad\square$

We now use Lemma 5 to prove the specific invariances. For each transformation, we will identify the corresponding function $S$ and show that it is strictly increasing.

**Translation**   Let $T(x) = x + c$ for a constant vector $c \in \mathcal{X}$. The transformed distance is:

$$d_X(T(x_i), T(x_j)) = \sqrt{\langle (x_i + c) - (x_j + c), (x_i + c) - (x_j + c) \rangle}$$
$$= \sqrt{\langle x_i - x_j, x_i - x_j \rangle}$$
$$= d_X(x_i, x_j)$$

Here, $S(d) = d$, which is the identity function and is strictly increasing. Therefore, by Lemma 5, TSI and QSI are invariant to translations.

**Isotropic Scaling**   Let $T(x) = \alpha x$ for a non-zero scalar $\alpha \in \mathbb{R}$. The transformed distance is:

$$d_X(T(x_i), T(x_j)) = \sqrt{\langle \alpha x_i - \alpha x_j, \alpha x_i - \alpha x_j \rangle}$$
$$= \sqrt{\alpha^2 \langle x_i - x_j, x_i - x_j \rangle}$$
$$= |\alpha| d_X(x_i, x_j)$$

Here, $S(d) = |\alpha| d$. Since $\alpha \neq 0$, $|\alpha|$ is a positive constant, making $S(d)$ a strictly increasing function. Therefore, by Lemma 5, TSI and QSI are invariant to isotropic scaling.

**Orthogonal Transformation**   Let $T$ be a linear map that preserves the inner product. The transformed distance is:

$$d_X(T(x_i), T(x_j)) = \sqrt{\langle T(x_i) - T(x_j), T(x_i) - T(x_j) \rangle}$$
$$= \sqrt{\langle T(x_i - x_j), T(x_i - x_j) \rangle} \quad \text{(by linearity of T)}$$
$$= \sqrt{\langle x_i - x_j, x_i - x_j \rangle}$$
$$= d_X(x_i, x_j)$$

Again, $S(d) = d$, which is strictly increasing. Therefore, by Lemma 5, TSI and QSI are invariant to orthogonal transformations.

**Non-Invariance to Invertible Linear Transformations**   Let $T$ be an invertible linear transformation, $T(x) = Ax$. Using the Singular Value Decomposition (SVD), $A$ can be written as $A = UDV^T$, where $U$ and $V$ are orthogonal transformations and $D = \text{diag}(\sigma_1, \ldots, \sigma_n)$ is a diagonal matrix with positive singular values $\sigma_i > 0$. Since TSI and QSI are invariant to orthogonal transformations (as shown previously), their invariance to $A$ is equivalent to their invariance to the diagonal scaling $D$. If the scaling is isotropic, i.e., $D = \alpha I$ for some scalar $\alpha > 0$, then the metrics are invariant.

We now show by counterexample that if the scaling is **non-isotropic**, TSI and QSI are not invariant. We will do this by constructing a dataset $X$ and a non-isotropic scaling $D$ such that the self-similarity score is no longer perfect. While it is trivially true that $\text{TSI}_{d_X, d_X}(X, X) = 1$, we will show that $\text{TSI}_{d_X, d_X}(D(X), X) < 1$. This inequality directly proves that the metric has changed, i.e., $\text{TSI}_{d_X, d_X}(D(X), X) \neq \text{TSI}_{d_X, d_X}(X, X)$.

By definition, a non-isotropic scaling requires at least two singular values to be different. Let $\sigma_a \neq \sigma_b$, with corresponding orthonormal basis vectors $\{e_a, e_b\}$.

- **For TSI**, consider the triplet $X = \{x_i, x_j, x_k\} = \{e_a + e_b, e_b, e_a\}$. In the original space, the distances from the anchor $x_i$ are:

$$d_X(x_i, x_j) = \|(e_a + e_b) - e_b\| = \|e_a\| = 1$$
$$d_X(x_i, x_k) = \|(e_a + e_b) - e_a\| = \|e_b\| = 1$$

  The ordinal comparison in the original space is thus $O_{d_X}^{\triangle}(x_i, x_j, x_k) = \text{sign}(1 - 1) = 0$. After applying the scaling $D$, the transformed points are $D(x_i) = \sigma_a e_a + \sigma_b e_b$, $D(x_j) = \sigma_b e_b$, and $D(x_k) = \sigma_a e_a$. The new distances in the

transformed space are:

$$d_X(D(x_i), D(x_j)) = \|(\sigma_a e_a + \sigma_b e_b) - \sigma_b e_b\| = \|\sigma_a e_a\| = \sigma_a$$
$$d_X(D(x_i), D(x_k)) = \|(\sigma_a e_a + \sigma_b e_b) - \sigma_a e_a\| = \|\sigma_b e_b\| = \sigma_b$$

Since $\sigma_a \neq \sigma_b$, the ordinal comparison in the transformed space is $O_{d_X}^{\Delta}(D(x_i), D(x_j), D(x_k)) = \text{sign}(\sigma_a - \sigma_b) \neq 0$.

For this triplet, the ordinal relationship in the original space (0) does not match the one in the transformed space ($\neq 0$). Therefore, the indicator function for this triplet in the calculation of $\text{TSI}_{d_X, d_X}(D(X), X)$ is 0. Since at least one term in the expectation is 0, the final score must be less than 1.

- **For QSI**, a similar argument holds. Consider the quadruplet $X = \{x_i, x_j, x_k, x_l\} = \{2e_a, e_a, 2e_b, e_b\}$. The initial distances are:

$$d_X(x_i, x_j) = \|2e_a - e_a\| = \|e_a\| = 1$$
$$d_X(x_k, x_l) = \|2e_b - e_b\| = \|e_b\| = 1$$

This results in an ordinal comparison of $O_{d_X}(x_i, x_j, x_k, x_l) = \text{sign}(1 - 1) = 0$. After applying $D$, the points become $D(x_i) = 2\sigma_a e_a$, $D(x_j) = \sigma_a e_a$, $D(x_k) = 2\sigma_b e_b$, and $D(x_l) = \sigma_b e_b$. The new distances are:

$$d_X(D(x_i), D(x_j)) = \|2\sigma_a e_a - \sigma_a e_a\| = \|\sigma_a e_a\| = \sigma_a$$
$$d_X(D(x_k), D(x_l)) = \|2\sigma_b e_b - \sigma_b e_b\| = \|\sigma_b e_b\| = \sigma_b$$

The new ordinal comparison is $O_{d_X}(D(x_i), D(x_j), D(x_k), D(x_l)) = \text{sign}(\sigma_a - \sigma_b) \neq 0$.

In both cases, the ordinal relationship has changed for the chosen set of points. This guarantees that $\text{TSI}_{d_X, d_X}(D(X), X) < 1$ and $\text{QSI}_{d_X, d_X}(D(X), X) < 1$, proving that the metrics are not generally invariant to non-isotropic scaling and thus not generally invariant to invertible linear transformations. $\square$

### N.6. TSI/QSI-based distances satisfy the metric axioms

**Corollary 6** (Complementary TSI and QSI Distances Satisfy the Metric Axioms)**.** *Let $(X, d_X)$, $(Y, d_Y)$, and $(Z, d_Z)$ be sets of $N$ representations equipped with their respective distance functions. For a similarity metric $S \in \{TSI, QSI\}$, define the distance function $d_S((X, d_X), (Y, d_Y)) = 1 - S_{d_X, d_Y}(X, Y)$. Then, $d_S$ satisfies the following metric axioms, as originally defined in* [Williams et al. (2021)](#)*:*

1. ***Equivalence:*** $d_S((X, d_X), (Y, d_Y)) = 0 \iff (X, d_X) \sim (Y, d_Y)$

2. ***Symmetry:*** $d_S((X, d_X), (Y, d_Y)) = d_S((Y, d_Y), (X, d_X))$

3. ***Triangle inequality:*** $d_S((X, d_X), (Z, d_Z)) \leq d_S((X, d_X), (Y, d_Y)) + d_S((Y, d_Y), (Z, d_Z))$

*Therefore, these are proper metrics over the space of representation sets defined by the equivalence relation $S_{d_X, d_Y}(X, Y) = 1$.*

*Proof.* We will provide the proof for TSI ($S = \text{TSI}$). The proof for QSI follows identically by replacing the set of triplets $\mathcal{T}$ with the set of quadruplets $\mathcal{Q}$, and the triplet ordinal function $O_d^{\Delta}$ with the quadruplet ordinal function $O_d$.

Recall that $\text{TSI}_{d_X, d_Y}(X, Y)$ is defined as the expected agreement of the ordinal functions over all triplets:

$$\text{TSI}_{d_X, d_Y}(X, Y) = \frac{1}{(N)_3} \sum_{(i,j,k) \in \mathcal{T}} \mathbb{I}\big[O_{d_X}^{\Delta}(x_i, x_j, x_k) = O_{d_Y}^{\Delta}(y_i, y_j, y_k)\big]$$

To simplify notation, let the ordinal profile of a dataset $(X, d_X)$ be denoted by the vector $\mathcal{P}_X \in \{-1, 0, 1\}^{(N)_3}$, where the elements correspond to the ordinal outcomes $O_{d_X}^{\Delta}(x_i, x_j, x_k)$ for all $(i, j, k) \in \mathcal{T}$.

We can now prove each of the three metric axioms for the distance function $d_{\text{TSI}}((X, d_X), (Y, d_Y)) = 1 - \text{TSI}_{d_X, d_Y}(X, Y)$:

**1. Equivalence:** Two representation sets are equivalent in their corresponding spaces if their ordinal profiles match perfectly, meaning $\mathcal{P}_X = \mathcal{P}_Y$.

$$d_{\text{TSI}}((X, d_X), (Y, d_Y)) = 0 \iff \text{TSI}_{d_X, d_Y}(X, Y) = 1$$
$$\iff \forall (i, j, k) \in \mathcal{T} : O_{d_X}^{\Delta}(x_i, x_j, x_k) = O_{d_Y}^{\Delta}(y_i, y_j, y_k)$$
$$\iff \mathcal{P}_X = \mathcal{P}_Y$$

**2. Symmetry:** Since the indicator function representing equality is commutative:

$$\mathbb{I}\big[O_{d_X}^{\Delta}(x_i, x_j, x_k) = O_{d_Y}^{\Delta}(y_i, y_j, y_k)\big] = \mathbb{I}\big[O_{d_Y}^{\Delta}(y_i, y_j, y_k) = O_{d_X}^{\Delta}(x_i, x_j, x_k)\big] \iff$$
$$\text{TSI}_{d_X, d_Y}(X, Y) = \text{TSI}_{d_Y, d_X}(Y, X) \iff$$
$$1 - \text{TSI}_{d_X, d_Y}(X, Y) = 1 - \text{TSI}_{d_Y, d_X}(Y, X) \iff$$
$$d_{\text{TSI}}((X, d_X), (Y, d_Y)) = d_{\text{TSI}}((Y, d_Y), (X, d_X))$$

**3. Triangle inequality:** Notice that the disagreement in the ordinal functions can be interpreted as the normalized Hamming distance between the ordinal profiles:

$$d_{\text{TSI}}((X, d_X), (Y, d_Y)) = 1 - \frac{1}{(N)_3} \sum_{(i,j,k) \in \mathcal{T}} \mathbb{I}\big[O_{d_X}^{\Delta}(x_i, x_j, x_k) = O_{d_Y}^{\Delta}(y_i, y_j, y_k)\big]$$
$$= \frac{1}{(N)_3} \sum_{(i,j,k) \in \mathcal{T}} \mathbb{I}\big[O_{d_X}^{\Delta}(x_i, x_j, x_k) \neq O_{d_Y}^{\Delta}(y_i, y_j, y_k)\big]$$
$$= \frac{1}{(N)_3} d_{\text{hamming}}(\mathcal{P}_X, \mathcal{P}_Y)$$

Since the Hamming distance over vector spaces satisfies the triangle inequality, it directly follows that:

$$d_{\text{hamming}}(\mathcal{P}_X, \mathcal{P}_Z) \leq d_{\text{hamming}}(\mathcal{P}_X, \mathcal{P}_Y) + d_{\text{hamming}}(\mathcal{P}_Y, \mathcal{P}_Z) \iff$$
$$\frac{1}{(N)_3} d_{\text{hamming}}(\mathcal{P}_X, \mathcal{P}_Z) \leq \frac{1}{(N)_3} d_{\text{hamming}}(\mathcal{P}_X, \mathcal{P}_Y) + \frac{1}{(N)_3} d_{\text{hamming}}(\mathcal{P}_Y, \mathcal{P}_Z) \iff$$
$$d_{\text{TSI}}((X, d_X), (Z, d_Z)) \leq d_{\text{TSI}}((X, d_X), (Y, d_Y)) + d_{\text{TSI}}((Y, d_Y), (Z, d_Z))$$

This concludes the proof. $\square$

## N.7. Kendall Tau Based Formulations

This subsection derives the connection between ordinal similarity metrics and Kendall's Tau correlation coefficient, as described in Corollary 10 and Corollary 11. To prove these statements, we first show that the Adjusted Kendall Tau coefficient described in Equation (5) is equivalent to a formulation that leverages a similar ordinal predicate to TSI and QSI. Next, we reformulate TSI as an average of anchor-based Adjusted Kendall's Tau correlations. Finally, we establish that the TSI and QSI can be jointly expressed via a global Adjusted Kendall Tau Coefficient.

N.7.1. ALTERNATIVE FORMULATION OF ADJUSTED KENDALL TAU

**Lemma 6** (Adjusted Kendall Tau Equivalent Formulation). *Let $A = \{a_1, \ldots, a_M\}$ and $B = \{b_1, \ldots, b_M\}$ be two sequences of paired observations and let Adj-Kendall-$\tau(A, B)$ denote the Adjusted Kendall Tau coefficient defined in Equation (5). Then, the following identity holds:*

$$\textit{Adj-Kendall-}\tau(A, B) = \frac{1}{M(M-1)} \sum_{j \neq k} \mathbb{I}\big[\text{sign}(a_j - a_k) = \text{sign}(b_j - b_k)\big]$$

*Proof.* Let $s_a = \text{sign}(a_j - a_k)$ and $s_b = \text{sign}(b_j - b_k)$ for any pair of indices $j, k$. The expression $2\mathbb{I}[s_a = s_b] - 1$ evaluates to 1 if the ordinal relationship is preserved ($s_a = s_b$) and $-1$ otherwise. We can verify by cases that the term inside the

summation of Kendall-$\tau(A, B)$+Adj-Ties$(A, B)$, which is $s_a s_b + \mathbb{I}[s_a = s_b = 0] - \mathbb{I}[s_a = 0 \wedge s_b \neq 0] - \mathbb{I}[s_a \neq 0 \wedge s_b = 0]$, also evaluates to 1 for agreement (concordant non-ties and joint ties) and $-1$ for disagreement (discordant non-ties, and ties in only one sequence). This implies that:

$$\text{Kendall-}\tau(A, B) + \text{Adj-Ties}(A, B) = \frac{1}{M(M-1)} \sum_{j \neq k} \left( 2\mathbb{I}\big[\text{sign}(a_j - a_k) = \text{sign}(b_j - b_k)\big] - 1 \right)$$

Thereby, enabling us to simplify the expression within the main summation of Adj-Kendall-$\tau(A, B)$:

$$
\begin{aligned}
\text{Adj-Kendall-}\tau(A, B) &= \frac{\text{Kendall-}\tau(A, B) + \text{Adj-Ties}(A, B) + 1}{2} \\
&= \frac{1}{2} \left[ \frac{1}{M(M-1)} \sum_{j \neq k} (2\mathbb{I}[\dots] - 1) + 1 \right] \\
&= \frac{1}{2} \left[ \frac{2}{M(M-1)} \sum_{j \neq k} \mathbb{I}[\dots] - \frac{M(M-1)}{M(M-1)} + 1 \right] \\
&= \frac{1}{M(M-1)} \sum_{j \neq k} \mathbb{I}\big[\text{sign}(a_j - a_k) = \text{sign}(b_j - b_k)\big]
\end{aligned}
$$

$\square$

### N.7.2. TSI as an Average of Anchor-based Kendall's Tau

**Corollary 10** (TSI as an Average of Anchor-based Kendall's Tau). *For each anchor point $i$, let $D_X^i = \{d_X(x_i, x_t) | t \neq i\}$ and $D_Y^i = \{d_Y(y_i, y_t) | t \neq i\}$ denote the collections of distances from that anchor, treated as paired sequences ordered by the index $t$. Then TSI can be expressed as:*

$$TSI_{d_X, d_Y}(X, Y) = \frac{1}{N} \sum_{i=1}^{N} \text{Adj-Kendall-}\tau(D_X^i, D_Y^i)$$

*where the Adjusted Kendall Tau is calculated according to Equation* (5).

*Proof.* The proof proceeds by reformulating the right-hand side (RHS) of the corollary's equation. Let $A = D_X^i$ and $B = D_Y^i$ be the sequences of distances from an anchor $i$, each of length $M = N - 1$.

First, we leverage Lemma 6 to simplify the Adjusted Kendall's $\tau$ coefficient. Notably, we see that:

$$\text{Adj-Kendall-}\tau(A, B) = \frac{1}{M(M-1)} \sum_{j \neq k} \mathbb{I}\big[\text{sign}(a_j - a_k) = \text{sign}(b_j - b_k)\big]$$

Now, we substitute this result back into the RHS of the corollary, with $M = N - 1$. Furthermore, note that $D_X^i = \{d_X(x_i, x_t) | t \neq i\}$ is ordered by letting $t$ range from 1 to $N$ (excluding $i$). Therefore, for $j' \neq k', j', k' \in \{1, \dots, N-1\}$, we can form $j \neq k, j, k \in \{1, \dots, N\} \setminus \{i\}$ such that $(D_X^i)_{j'} = d_X(x_i, x_j)$ and $(D_Y^i)_{j'} = d_Y(y_i, y_j)$. Therefore, we can rewrite the RHS of the equation into:

$$\text{RHS} = \frac{1}{N} \sum_{i=1}^{N} \left( \frac{1}{(N-1)(N-2)} \sum_{j' \neq k'} \mathbb{I}\big[\text{sign}((D_X^i)_{j'} - (D_X^i)_{k'}) = \text{sign}((D_Y^i)_{j'} - (D_Y^i)_{k'})\big] \right)$$

$$= \frac{1}{N} \sum_{i=1}^{N} \left( \frac{1}{(N-1)(N-2)} \sum_{\substack{j,k \neq i \\ j \neq k}} \mathbb{I}\big[\text{sign}(d_X(x_i,x_j) - d_X(x_i,x_k)) = \text{sign}(d_Y(y_i,y_j) - d_Y(y_i,y_k))\big] \right)$$

$$= \frac{1}{N(N-1)(N-2)} \sum_{i=1}^{N} \sum_{\substack{j,k \neq i \\ j \neq k}} \mathbb{I}\big[\ldots\big]$$

The double summation iterates over all triplets of distinct indices $(i,j,k)$, of which there are $N(N-1)(N-2) = |\mathcal{T}|$. The expression is therefore the expectation over all such triplets, which is the definition of TSI:

$$\text{RHS} = \mathbb{E}_{(i,j,k) \in \mathcal{T}}\Big[\mathbb{I}\big[\text{sign}(d_X(x_i,x_j) - d_X(x_i,x_k)) = \text{sign}(d_Y(y_i,y_j) - d_Y(y_i,y_k))\big]\Big]$$

$$= \mathbb{E}_{(i,j,k) \in \mathcal{T}}\Big[\mathbb{I}\big[O_{d_X}^{\Delta}(x_i,x_j,x_k) = O_{d_Y}^{\Delta}(y_i,y_j,y_k)\big]\Big] = \text{TSI}_{d_X,d_Y}(X,Y) \qquad \square$$

### N.7.3. UNIFIED KENDALL'S TAU FORMULATION FOR TSI AND QSI

**Corollary 11** (Unified Kendall's Tau Formulation for TSI and QSI). *Let* $D_X^{all} = \{d_X(x_i,x_t)|i < t\}$ *and* $D_Y^{all} = \{d_Y(y_i,y_t)|i < t\}$ *denote the collections of all unique pairwise distances, treated as paired sequences ordered lexicographically by* $(i,t)$*. Then, for distance functions* $d_X$ *and* $d_Y$ *that satisfy the symmetry property, i.e.,* $d(a,b) = d(b,a)$*, a weighted sum of TSI and QSI can be expressed as a function of Kendall's Tau coefficient:*

$$\frac{4}{N+1} TSI_{d_X,d_Y}(X,Y) + \frac{N-3}{N+1} QSI_{d_X,d_Y}(X,Y) = \text{Adj-Kendall-}\tau(D_X^{all}, D_Y^{all})$$

*where the Adjusted Kendall Tau is calculated according to Equation* (5).

*Proof.* The proof proceeds by reformulating the right-hand side (RHS) of the corollary's equation. Firstly, we leverage the results from Lemma 6 to simplify the RHS of the corollary to:

$$\text{Adj-Kendall-}\tau(D_X^{\text{all}}, D_Y^{\text{all}}) =$$

$$\frac{1}{M(M-1)} \sum_{j \neq k} \mathbb{I}\big[\text{sign}((D_X^{\text{all}})_j - (D_X^{\text{all}})_k) = \text{sign}((D_Y^{\text{all}})_j - (D_Y^{\text{all}})_k)\big]$$

Furthermore, by noting that $D_X^{\text{all}} = \{d_X(x_i,x_t)|i < t\}$ and $D_Y^{\text{all}} = \{d_Y(y_i,y_t)|i < t\}$, we know that $M = \frac{N(N-1)}{2}$. Additionally, we can also restructure the summation in terms of $i$ and $t$:

$$\frac{1}{M(M-1)} \sum_{j' \neq k'} \mathbb{I}\big[\text{sign}((D_X^{\text{all}})_{j'} - (D_X^{\text{all}})_{k'}) = \text{sign}((D_Y^{\text{all}})_{j'} - (D_Y^{\text{all}})_{k'})\big] =$$

$$\frac{1}{(\frac{N(N-1)}{2})_2} \sum_{\substack{i=1 \\ i<t}}^{N} \sum_{\substack{j=1 \\ j<k \\ (i,t) \neq (j,k)}}^{N} \mathbb{I}\big[\text{sign}(d_X(x_i,x_t) - d_X(x_j,x_k)) = \text{sign}(d_Y(y_i,y_t) - d_Y(y_j,y_k))\big] =$$

$$\frac{4}{(N+1)_4} \sum_{\substack{i=1 \\ i<t}}^{N} \sum_{\substack{j=1 \\ j<k \\ (i,t) \neq (j,k)}}^{N} \mathbb{I}\big[\text{sign}(d_X(x_i,x_t) - d_X(x_j,x_k)) = \text{sign}(d_Y(y_i,y_t) - d_Y(y_j,y_k))\big]$$

Here $(n)_k = n \times (n-1) \times \cdots \times (n-k+1)$ denotes the falling factorial. Further, since $d_X$ is symmetric, we have that $\text{sign}(d_X(x_i, x_t) - d_X(x_j, x_k))$ is invariant to the permutation of $i$ and $t$, as well as to the permutation of $j$ and $k$. The same holds for $d_Y$. Therefore, we can equivalently consider all of the aforementioned permutations if we adjust our sum by a factor of 4, resulting in the following equation:

$$
\frac{1}{(N+1)_4} \sum_{\substack{i=1 \\ i \neq t}}^{N} \sum_{\substack{j=1 \\ j \neq k \\ (i,t) \neq (j,k) \\ (t,i) \neq (j,k)}}^{N} \mathbb{I}\big[\text{sign}(d_X(x_i, x_t) - d_X(x_j, x_k)) = \text{sign}(d_Y(y_i, y_t) - d_Y(y_j, y_k))\big]
$$

Now, for clarity, we define the current indices of the summation as a set $\mathcal{S}_O := \{(i,t,j,k)|(i,t,j,k) \in \{1,\ldots N\}^4 : i \neq t \wedge j \neq k \wedge (i,t) \neq (j,k) \wedge (t,i) \neq (j,k)\}$. This set can be decomposed into two disjoint sets $\mathcal{S}_\mathcal{T} := \{(i,t,j,k)|(i,t,j,k) \in \mathcal{S}_O : i = j \vee i = k \vee t = j \vee t = k\}$ and $\mathcal{S}_\mathcal{Q} := \{(i,t,j,k)|(i,t,j,k) \in \mathcal{S}_O : i \neq j \wedge i \neq k \wedge t \neq j \wedge t \neq k\}$. This is trivially true, as one set's predicate is the negation of the other.

Further analyzing these sets, we can alternatively described them as: The set $\mathcal{S}_\mathcal{Q}$ corresponding to all possible quadruplets for which all elements are distinct and in $\{1, \ldots N\}$, and the set $\mathcal{S}_\mathcal{T}$ corresponding to all possible quadruplets for which exactly two elements are identical and in $\{1, \ldots N\}$.

Therefore, we directly conclude that:

$$
\sum_{\substack{i=1 \\ t \neq i}}^{N} \sum_{\substack{j=1 \\ k \neq j \\ (i,t) \neq (j,k) \\ (t,i) \neq (j,k)}}^{N} \mathbb{I}[\ldots] = \sum_{(i,t,j,k) \in \mathcal{S}_O} \mathbb{I}[\ldots] = \sum_{(i,t,j,k) \in \mathcal{S}_\mathcal{T}} \mathbb{I}[\ldots] + \sum_{(i,t,j,k) \in \mathcal{S}_\mathcal{Q}} \mathbb{I}[\ldots]
$$

We now analyze the sums over the two disjoint sets $\mathcal{S}_\mathcal{Q}$ (where all four indices are distinct) and $\mathcal{S}_\mathcal{T}$ (where exactly two indices are identical) separately.

**Analysis of Quadruplet Terms ($S_Q$)** The set $\mathcal{S}_\mathcal{Q}$ consists of all ordered quadruplets $(i, t, j, k)$ with four distinct indices. This set is equivalent to $\mathcal{Q}$, the set of quadruplets over which QSI is defined, and contains $|\mathcal{Q}| = (N)_4$ elements. The sum over this set corresponds to the QSI metric:

$$
\sum_{(i,t,j,k) \in \mathcal{S}_\mathcal{Q}} \mathbb{I}\big[\text{sign}(d_X(x_i, x_t) - d_X(x_j, x_k)) = \text{sign}(d_Y(y_i, y_t) - d_Y(y_j, y_k))\big]
$$

$$
= \sum_{(i,t,j,k) \in \mathcal{Q}} \mathbb{I}\big[O_{d_X}(x_i, x_t, x_j, x_k) = O_{d_Y}(y_i, y_t, x_j, x_k)\big]
$$

$$
= (N)_4 \cdot \mathbb{E}_{(i,t,j,k) \in \mathcal{Q}}\Big[\mathbb{I}\big[O_{d_X}(x_i, x_t, x_j, x_k) = O_{d_Y}(y_i, y_t, x_j, x_k)\big]\Big]
$$

$$
= (N)_4 \cdot \text{QSI}_{d_X, d_Y}(X, Y)
$$

**Analysis of Triplet Terms ($S_T$)** The set $\mathcal{S}_\mathcal{T}$ consists of all quadruplets where exactly two indices are identical. This occurs in four disjoint and symmetric cases: (1) $i = j$, (2) $i = k$, (3) $t = j$, and (4) $t = k$. Each case contributes an equal amount to the sum. We analyze the case where $i = j$: the predicate becomes $\mathbb{I}\big[\text{sign}(d_X(x_i, x_t) - d_X(x_i, x_k)) = \text{sign}(d_Y(y_i, y_t) - d_Y(y_i, y_k))\big]$, where $i, t, k$ are distinct. This is the predicate for TSI over the triplet $(i, t, k)$. The sum for this case is over all $(N)_3$ distinct triplets:

$$
\sum_{\substack{i,t \neq k \\ i \neq t}} \mathbb{I}\big[\text{sign}(d_X(x_i, x_t) - d_X(x_i, x_k)) = \text{sign}(d_Y(y_i, y_t) - d_Y(y_i, y_k))\big]
$$

$$
= \sum_{(i,t,k) \in \mathcal{T}} \mathbb{I}\big[O_{d_X}^\Delta(x_i, x_t, x_k) = O_{d_Y}^\Delta(y_i, y_t, y_k)\big]
$$

$$
= (N)_3 \cdot \text{TSI}_{d_X, d_Y}(X, Y)
$$

Since there are four such symmetric cases, the total sum over $\mathcal{S}_\mathcal{T}$ is four times this value:

$$\sum_{(i,t,j,k)\in\mathcal{S}_\mathcal{T}} \mathbb{I}[\dots] = 4(N)_3 \cdot \text{TSI}_{d_X,d_Y}(X,Y)$$

We combine the results for the two partitions and substitute them back into the expression for the RHS of the corollary:

$$\begin{aligned}
\text{RHS} &= \frac{1}{(N+1)_4}\left(\sum_{(i,t,j,k)\in\mathcal{S}_\mathcal{T}} \mathbb{I}[\dots] + \sum_{(i,t,j,k)\in\mathcal{S}_\mathcal{Q}} \mathbb{I}[\dots]\right)\\
&= \frac{1}{(N+1)_4}\left(4(N)_3 \cdot \text{TSI}_{d_X,d_Y}(X,Y) + (N)_4 \cdot \text{QSI}_{d_X,d_Y}(X,Y)\right)\\
&= \frac{4(N)_3}{(N+1)(N)_3}\text{TSI}_{d_X,d_Y}(X,Y) + \frac{(N)_4}{(N+1)(N)_3}\text{QSI}_{d_X,d_Y}(X,Y)\\
&= \frac{4}{N+1}\text{TSI}_{d_X,d_Y}(X,Y) + \frac{N-3}{N+1}\text{QSI}_{d_X,d_Y}(X,Y) \qquad\qquad \square
\end{aligned}$$

### N.8. Computational Complexity of Adjusted Kendall Tau

This proof establishes the computational complexity results presented in Corollary 12 for the computation of the Adjusted Kendall Tau coefficient using Algorithm 1. The approach is inspired by efficient algorithms for calculating Kendall's Tau, such as the one proposed by Knight (1966), and relies on reformulating the metric in terms of concordant and tied pairs.

**Corollary 12** (Computational Complexity of Adj-Kendall-$\tau$). *Let $A = \{a_1,\dots,a_M\}$ and $B = \{b_1,\dots,b_M\}$ be two sequences of paired observations. The Adj-Kendall-$\tau(A,B)$, described in Equation (5), can be computed in $\mathcal{O}(M\log M)$ time and $\mathcal{O}(M)$ space complexity.*

*Proof.* We first prove that Algorithm 1 correctly computes the Adjusted Kendall Tau and then analyze its time and space complexity.

**Correctness**    From Lemma 6, the Adjusted Kendall Tau is the fraction of ordered pairs of indices $(j,k)$ where the ordinal relationship agrees between sequences $A$ and $B$:

$$\text{Adj-Kendall-}\tau(A,B) = \frac{1}{M(M-1)}\sum_{j\neq k}\mathbb{I}\big[\text{sign}(a_j - a_k) = \text{sign}(b_j - b_k)\big]$$

An agreement occurs if the unordered pair $\{j,k\}$ is either strictly concordant (e.g., $a_j > a_k$ and $b_j > b_k$) or jointly tied ($a_j = a_k$ and $b_j = b_k$). Let $C$ be the number of strictly concordant unordered pairs and $T_{AB}$ be the number of jointly tied unordered pairs. Each such unordered pair corresponds to two agreeing ordered pairs, $(j,k)$ and $(k,j)$. Therefore, the numerator of the expression—the total count of agreeing ordered pairs—is equal to $2(C + T_{AB})$.

Algorithm 1 correctly computes this value. The number of strictly concordant pairs, $C$, is calculated using the identity that the total number of unordered pairs, $\frac{M(M-1)}{2}$, is the sum of concordant, discordant ($I$), and tied pairs. By subtracting the number of inversions ($I$) and ties in each sequence ($T_A, T_B$) from the total, and adding back the number of joint ties ($T_{AB}$) which were subtracted twice, we isolate $C$. The algorithm then computes the total number of agreeing ordered pairs as $J = 2(C + T_{AB})$ (line 7) and normalizes by the total number of ordered pairs, $M(M-1)$ (line 8), thus correctly computing the Adjusted Kendall Tau.

**Computational Time Complexity**    The time complexity of Algorithm 1 is determined by its most expensive steps. Let $M$ be the length of the input sequences $A$ and $B$.

- **Ranking (line 1):** Computing the rank orderings $\pi_A$ and $\pi_B$ requires sorting, which takes $\mathcal{O}(M\log M)$ time.

- **Counting Ties (lines 2-4):** After sorting, ties can be counted in a single pass, which takes $\mathcal{O}(M)$ time.

- **Counting Inversions (line 5):** The number of inversions between two permutations can be computed efficiently using an approach similar to merge sort in $\mathcal{O}(M \log M)$ time.

The remaining steps take constant time. Therefore, the dominant factor is the ranking and inversion counting, resulting in a total time complexity of $\mathcal{O}(M \log M)$.

**Computational Space Complexity** The space complexity is determined by the storage required for the data structures used.

- The input sequences $A$ and $B$, along with their ranks $\pi_A$ and $\pi_B$, require $\mathcal{O}(M)$ space.

- The merge sort-based algorithm for counting inversions requires $\mathcal{O}(M)$ space complexity.

Thus, the total space complexity of the algorithm is $\mathcal{O}(M)$. □

### N.9. Concentration Bounds for Empirical TSI and QSI

**Lemma 4** (Concentration Bounds for Empirical TSI and QSI). *Let $(X, Y) \sim \mathcal{D}_{XY}^N$ be a finite dataset of $N$ independent pairs sampled from the joint distribution $\mathcal{D}_{XY}$. Because TSI and QSI are U-Statistics with bounded indicator kernels in $[0,1]$ (of degrees $m = 3$ and $m = 4$, respectively), their empirical estimations tightly concentrate around their true population similarities $S^* \in \{TSI^*, QSI^*\}$. For any $\epsilon > 0$, the probabilities of deviation are bounded by:*

$$\mathbb{P}\left[\left|TSI_{d_X,d_Y}(X, Y) - TSI^*_{d_X,d_Y}(\mathcal{D}_{XY})\right| \geq \epsilon\right] \leq 2\exp\left(-2\lfloor\tfrac{N}{3}\rfloor\epsilon^2\right)$$

$$\mathbb{P}\left[\left|QSI_{d_X,d_Y}(X, Y) - QSI^*_{d_X,d_Y}(\mathcal{D}_{XY})\right| \geq \epsilon\right] \leq 2\exp\left(-2\lfloor\tfrac{N}{4}\rfloor\epsilon^2\right)$$

*Proof.* **1. Definition of U-Statistics.** A U-Statistic provides a method for obtaining unbiased estimators of population parameters. Given a set of $N$ independent observations $Z = \{z_1, \ldots, z_N\}$, and a symmetric kernel function $h(z_1, \ldots, z_m)$ of degree $m \leq N$, the corresponding U-Statistic is defined as the average of the kernel over all combinations of $m$ items from the input set:

$$U(Z) = \frac{1}{\binom{N}{m}} \sum_{1 \leq i_1 < \cdots < i_m \leq N} h(z_{i_1}, \ldots, z_{i_m})$$

**2. Hoeffding's Bound Applied to U-Statistics.** Because the terms in the sum of a U-Statistic share overlapping subsets of data, they are not independent. Consequently, the standard Hoeffding's Inequality cannot be applied directly. However, Hoeffding (1963) derived a specific concentration inequality for U-statistics. For a kernel bounded in $[0, 1]$, the bound is given by:

$$\mathbb{P}\left[|U(Z) - \mathbb{E}[U(Z)]| \geq \epsilon\right] \leq 2\exp\left(-2\lfloor\tfrac{N}{m}\rfloor\epsilon^2\right)$$

**3. Equivalence of Expectations.** Because all subsets of size $m$ are drawn from the same underlying distribution, the expected value of the kernel over any combination is identical. Using the linearity of expectation, the expected value of the U-Statistic over $N$ samples collapses strictly to the expected value of its kernel over $m$ samples:

$$\begin{aligned}
\mathbb{E}[U(Z)] &= \mathbb{E}\left[\frac{1}{\binom{N}{m}} \sum_{1 \leq i_1 < \cdots < i_m \leq N} h(z_{i_1}, \ldots, z_{i_m})\right] \\
&= \frac{1}{\binom{N}{m}} \sum_{1 \leq i_1 < \cdots < i_m \leq N} \mathbb{E}\left[h(z_{i_1}, \ldots, z_{i_m})\right] \\
&= \frac{1}{\binom{N}{m}} \binom{N}{m} \mathbb{E}[h(z_1, \ldots, z_m)] \\
&= \mathbb{E}[h(z_1, \ldots, z_m)]
\end{aligned}$$

**4. TSI and QSI as U-Statistics.** Consider a sample of $N$ joint pairs $z_i = (x_i, y_i)$. To demonstrate that TSI is a U-Statistic, we begin with its original definition as an average over the set of all ordered triplets $\mathcal{T}$, where $|\mathcal{T}| = (N)_3$:

$$\text{TSI}_{d_X,d_Y}(X,Y) = \frac{1}{(N)_3} \sum_{(i,j,k)\in\mathcal{T}} \mathbb{I}\big[O_{d_X}^\Delta(x_i,x_j,x_k) = O_{d_Y}^\Delta(y_i,y_j,y_k)\big]$$

Any ordered triplet $(i,j,k)$ can be formed by first selecting an unordered subset of 3 distinct indices, and then applying one of the $3! = 6$ possible permutations $\pi \in S_3$. We can rewrite the sum over ordered tuples as a double sum over unordered subsets and permutations:

$$\text{TSI}_{d_X,d_Y}(X,Y) = \frac{1}{\binom{N}{3}3!} \sum_{1\le i_1 < i_2 < i_3 \le N} \sum_{\pi\in S_3} \mathbb{I}\big[O_{d_X}^\Delta(x_{\pi(1)},x_{\pi(2)},x_{\pi(3)}) = O_{d_Y}^\Delta(y_{\pi(1)},y_{\pi(2)},y_{\pi(3)})\big]$$

$$= \frac{1}{\binom{N}{3}} \sum_{1\le i_1 < i_2 < i_3 \le N} \left(\frac{1}{6} \sum_{\pi\in S_3} \mathbb{I}\big[O_{d_X}^\Delta(x_{\pi(1)},x_{\pi(2)},x_{\pi(3)}) = O_{d_Y}^\Delta(y_{\pi(1)},y_{\pi(2)},y_{\pi(3)})\big]\right)$$

By defining the term in parentheses as the permutation-symmetric kernel $h_{\text{TSI}}(z_1, z_2, z_3)$, we perfectly recover the definition of a U-Statistic from Step 1. This demonstrates that TSI is strictly a U-Statistic of degree $m = 3$. By an identical construction using subsets of size 4 and permutations $\pi \in S_4$, QSI forms a symmetric kernel $h_{\text{QSI}}$, proving it is a U-Statistic of degree $m = 4$. Because both kernels are simple averages of indicator functions, their values are strictly bounded in $[0, 1]$.

**5. Obtaining the Population Equivalence and Concentration Bounds.** Using the property from Step 3, the expected value of our U-Statistics for any sample size $N \ge 3$ evaluates exactly to the expectation over a single kernel evaluation. Thus, we have:

$$\mathbb{E}_{(X,Y)\sim\mathcal{D}_{XY}^N}[\text{TSI}_{d_X,d_Y}(X,Y)] = \mathbb{E}_{(X,Y)\sim\mathcal{D}_{XY}^3}[h_{\text{TSI}}((x_1,y_1),(x_2,y_2),(x_3,y_3))]$$

Therefore, we can easily simplify the population similarity limit from Definition 4:

$$\text{TSI}_{d_X,d_Y}^*(\mathcal{D}_{XY}) = \lim_{N\to+\infty} \mathbb{E}_{(X,Y)\sim\mathcal{D}_{XY}^N}[\text{TSI}_{d_X,d_Y}(X,Y)]$$

$$= \lim_{N\to+\infty} \mathbb{E}_{(X,Y)\sim\mathcal{D}_{XY}^3}[h_{\text{TSI}}((x_1,y_1),(x_2,y_2),(x_3,y_3))]$$

$$= \mathbb{E}_{(X,Y)\sim\mathcal{D}_{XY}^3}[h_{\text{TSI}}((x_1,y_1),(x_2,y_2),(x_3,y_3))]$$

Finally, to compute the concentration bound, we apply the inequality from Step 2. We substitute $U(Z) = \text{TSI}_{d_X,d_Y}(X,Y)$, its expected value $\mathbb{E}[U(Z)] = \mathbb{E}_{(X,Y)\sim\mathcal{D}_{XY}^N}[\text{TSI}_{d_X,d_Y}(X,Y)]$, and the kernel degree $m = 3$:

$$\mathbb{P}\left[\left|\text{TSI}_{d_X,d_Y}(X,Y) - \mathbb{E}_{(X,Y)\sim\mathcal{D}_{XY}^N}[\text{TSI}_{d_X,d_Y}(X,Y)]\right| \ge \epsilon\right] \le 2\exp\left(-2\lfloor\tfrac{N}{3}\rfloor\epsilon^2\right) \iff$$

$$\mathbb{P}\left[\left|\text{TSI}_{d_X,d_Y}(X,Y) - \mathbb{E}_{(X,Y)\sim\mathcal{D}_{XY}^3}[h_{\text{TSI}}((x_1,y_1),(x_2,y_2),(x_3,y_3))]\right| \ge \epsilon\right] \le 2\exp\left(-2\lfloor\tfrac{N}{3}\rfloor\epsilon^2\right) \iff$$

$$\mathbb{P}\left[\left|\text{TSI}_{d_X,d_Y}(X,Y) - \text{TSI}_{d_X,d_Y}^*(\mathcal{D}_{XY})\right| \ge \epsilon\right] \le 2\exp\left(-2\lfloor\tfrac{N}{3}\rfloor\epsilon^2\right)$$

Applying the same sequence of substitutions with $m = 4$ for QSI yields its respective bound, completing the proof for Lemma 4. $\square$

### N.10. Computability of Global Lower Bounds for TSI and QSI in $\mathbb{R}$

This section provides the formal proof for Corollary 7. We demonstrate that the search for the global minimum similarity in $\mathbb{R}$ can be reduced from a continuous problem over $\mathbb{R}^N \times \mathbb{R}^N$ to a finite combinatorial search over a specialized representative set.

**Corollary 7** (Computability of Global Lower Bounds in $\mathbb{R}$). *For any number of considered representations $N$, the global lower bound of the ordinal similarity $S \in \{TSI, QSI\}$ over sets of $N$ distinct points in $\mathbb{R}$ is theoretically computable. Specifically, let $s_{\min} = \min_{X,Y\in\mathbb{R}^N \text{ with distinct entries}} S(X,Y)$. Then:*

$$s_{\min} = \min_{X,Y\in\mathcal{X}_{rep}} \left(\min_{\sigma\in Perm(N)} S(X,Y^\sigma)\right)$$

*where $\mathcal{X}_{rep}$ denotes a representative set as defined in Definition 7. Note that $\mathcal{X}_{rep}$ is finite and we can computationally obtain at least one such set (specifically a Monotonically Ordered Representative Set via Algorithm 6).*

*Proof.* The proof is structured as follows: we first discretize the representation space into finite equivalence classes (**Ordinal Profiles**); then, we show that the global search can be restricted to a finite **Representative Set**; finally, we leverage the geometric constraints of the real line to algorithmically construct this set (**MORS**).

**Equivalence Classes and Discretization**    To analyze the global minimum, we partition the continuous space $\mathbb{R}^N$ into finite regions of invariant ordinal behavior.

**Definition 5** (Ordinal Profile). To characterize the structure of $X \in \mathbb{R}^N$, we define the **Ordinal Profile** $\mathcal{P}_S(X)$ as an ordered set of outcomes for all underlying comparisons in the metric $S \in \{\text{TSI}, \text{QSI}\}$. For a dataset $X$, the profile is a vector in $\{-1, 0, 1\}^M$ (where $M = |\mathcal{T}|$ or $|\mathcal{Q}|$):

- **For TSI:** $\mathcal{P}_{\text{TSI}}(X) = \left(O_{d_X}^{\Delta}(x_i, x_j, x_k)\right)_{(i,j,k) \in \mathcal{T}}$

- **For QSI:** $\mathcal{P}_{\text{QSI}}(X) = \left(O_{d_X}(x_i, x_j, x_k, x_l)\right)_{(i,j,k,l) \in \mathcal{Q}}$

**Definition 6** (Equivalent Classes/Cells). We define the equivalence relation $\sim_S$ such that $X \sim_S Y \iff \mathcal{P}_S(X) = \mathcal{P}_S(Y)$. This relation partitions $\mathbb{R}^N$ with distinct entries into disjoint **equivalence classes** (or cells) $\{\mathcal{E}_k\}$, where each $\mathcal{E}_k = \{X \in \mathbb{R}^N \mid \mathcal{P}_S(X) = p_k\}$ for a unique profile $p_k$. By construction, for any $X, Y \in \mathcal{E}_k$, $S(X, Y) = 1$.

- **Conclusion 1 (Finiteness):** The number of cells $K$ is finite and bounded by $3^M$. Geometric constraints in $\mathbb{R}$ (e.g., $|x_i - x_k| = |x_i - x_j| + |x_j - x_k|$ for $i < j < k$) further restricts the number of realizable profiles.

- **Conclusion 2 (Class Score Invariance):** Since $S$ depends only on the profiles, the similarity between any two points from cells $\mathcal{E}_a$ and $\mathcal{E}_b$ is identical. Formally:

$$\forall X_1, X_2 \in \mathcal{E}_a, \forall Y_1, Y_2 \in \mathcal{E}_b : S(X_1, Y_1) = S(X_2, Y_2).$$

**Formal Definition of the Representative Set**    We define a permutation action on cells such that $\sigma(\mathcal{E}_a) = \{X^\sigma \mid X \in \mathcal{E}_a\}$. We call $\mathcal{E}_a$ and $\mathcal{E}_b$ $\sigma$-**equivalent** if there exists a permutation $\sigma \in \text{Perm}(N)$ such that $\sigma(\mathcal{E}_a) = \mathcal{E}_b$.

**Definition 7** (Representative Set). A **Representative Set** $\mathcal{X}_{rep} = \{X_1, \ldots, X_m\}$ is a set of configurations constructed such that:

1. **Distinct Orbits:** $\forall X_i, X_j \in \mathcal{X}_{rep}, i \neq j \implies \nexists \sigma \in \text{Perm}(N)$ s.t. $X_i \sim_S X_j^\sigma$.

2. **Full Coverage:** For every equivalence class $\mathcal{E}_k \subset \mathbb{R}^N$ with distinct entries, there exists a representative $X_i \in \mathcal{X}_{rep}$ and a permutation $\sigma \in \text{Perm}(N)$ such that $\mathcal{E}_k = \{X \mid X \sim_S X_i^\sigma\}$.

The definition of $\mathcal{X}_{rep}$ implies the **covering property**: $\forall X \in \mathbb{R}^N$ with distinct entries, $\exists X_i \in \mathcal{X}_{rep}, \sigma \in \text{Perm}(N)$ such that $X \sim_S X_i^\sigma$.

**Global Lower Bound Construction**    We now show that the search over $\mathcal{X}_{rep}$ and $\text{Perm}(N)$ is equivalent to the global minimum over $\mathbb{R}^N \times \mathbb{R}^N$ (with distinct entries). Let $X, Y \in \mathbb{R}^N$ with distinct entries. By the covering property:

$$X \sim_S X_i^{\sigma_1} \quad \text{and} \quad Y \sim_S X_j^{\sigma_2} \quad \text{for some } X_i, X_j \in \mathcal{X}_{rep} \text{ and } \sigma_1, \sigma_2 \in \text{Perm}(N).$$

By Class Score Invariance and Permutation Invariance ($S(A, B) = S(A^\sigma, B^\sigma)$):

$$\begin{aligned}
S(X, Y) &= S(X_i^{\sigma_1}, X_j^{\sigma_2}) \\
&= S(X_i^{\sigma_1 \sigma_1^{-1}}, X_j^{\sigma_2 \sigma_1^{-1}}) \\
&= S(X_i, X_j^{\sigma^*}) \quad \text{where } \sigma^* = \sigma_2 \sigma_1^{-1} \in \text{Perm}(N).
\end{aligned}$$

Therefore, the global minimum for distinct entries $s_{\min} = \min_{X,Y \in \mathbb{R}^N \text{ with distinct entries}} S(X,Y)$ reduces to:

$$s_{\min} = \min_{X_i, X_j \in \mathcal{X}_{\text{rep}}} \left( \min_{\sigma \in \text{Perm}(N)} S(X_i, X_j^\sigma) \right)$$

Finally, this procedure can be computed via Algorithm 5. However, this hinges on the ability to computationally obtain $\mathcal{X}_{\text{rep}}$, which we will cover next.

---

**Algorithm 5** Computation of Global Lower Bound for Ordinal Similarity in $\mathbb{R}$

---

**Require:** Number of points $N$; Similarity metric $S \in \{\text{TSI}, \text{QSI}\}$
1: $\mathcal{X}_{\text{rep}} \leftarrow \text{ObtainRepresentativeConfigs}(N, S)$
2: $s_{\min} \leftarrow 1.0$
3: $d(a, b) \leftarrow |a - b|$
4: **for all** $X \in \mathcal{X}_{\text{rep}}$ **do**
5:     **for all** $Y \in \mathcal{X}_{\text{rep}}$ **do**
6:         **for all** $\sigma \in \text{Permutations}(\{1, \ldots, N\})$ **do**
7:             $s \leftarrow S_{d,d}(X, Y^\sigma)$
8:             **if** $s < s_{\min}$ **then**
9:                 $s_{\min} \leftarrow s$
10:            **end if**
11:         **end for**
12:     **end for**
13: **end for**
14: **return** $s_{\min}$

---

**Efficient Computation of $\mathcal{X}_{\text{rep}}$**     While the number of ordinal profiles is finite, searching the entirety of $\mathbb{R}^N$ is still non-trivial. However, for 1D representations, the geometric structure of the real line allows for a significant simplification. Since any set of distinct points can be sorted, every ordinal profile in $\mathbb{R}^N$ is effectively a permuted version of a profile generated by a monotonically increasing sequence. We formalize this by restricting our search to the monotonic subset of the representation space.

Let $\mathcal{X}_< := \{X \in \mathbb{R}^N \mid x_1 < x_2 < \cdots < x_N\}$ denote the set of all strictly increasing configurations in $\mathbb{R}^N$. We define a specialized representative set for this domain:

**Definition 8** (Monotonically Ordered Representative Set). A **Monotonically Ordered Representative Set** (MORS), denoted $\mathcal{X}_{\text{rep}}^{MO}$, is a finite collection of configurations $\{X_1, \ldots, X_m\}$ such that:

1. **Monotonicity:** $\mathcal{X}_{\text{rep}}^{MO} \subset \mathcal{X}_<$.

2. **Distinct Cells:** $\forall X_i, X_j \in \mathcal{X}_{\text{rep}}, i \neq j \implies X_i \nsim_S X_j$ (they represent distinct ordinal profiles).

3. **Full Coverage of Ordered Cells:** For every equivalence class $\mathcal{E}_k$ such that $\mathcal{E}_k \cap \mathcal{X}_< \neq \emptyset$, there exists a representative $X_i \in \mathcal{X}_{\text{rep}}^{MO}$ such that $\mathcal{E}_k = \{X \mid X \in \mathbb{R}^N \text{ with distinct entries}, X \sim_S X_i\}$.

The following corollary establishes that identifying these ordered representatives is sufficient to characterize the similarity behavior of all possible configurations in $\mathbb{R}^N$.

**Corollary 13** (MORS are Representative Sets). *Let $\mathcal{X}_{rep}^{MO}$ be a **Monotonically Ordered Representative Set**. Then $\mathcal{X}_{rep}^{MO}$ satisfies the requirements of a **Representative Set (RS)** for $\mathbb{R}^N$ with distinct entries.*

*Proof.* To show that $\mathcal{X}_{\text{rep}}^{MO}$ is an RS, we must verify the **Distinct Orbits** and **Full Coverage** properties.

**1. Distinct Orbits:** Assume for contradiction that there exist $X_i, X_j \in \mathcal{X}_{\text{rep}}^{MO}$ and a permutation $\sigma \in \text{Perm}(N)$ such that $X_i \sim_S X_j^\sigma$. Since $X_i \in \mathcal{X}_<$, its profile $\mathcal{P}_S(X_i)$ satisfies specific monotonic distance constraints (e.g., $d(x_{i,1}, x_{i,2}) < d(x_{i,1}, x_{i,3})$). Because $X_j$ is also monotonically ordered, any non-identity permutation $\sigma$ would reorder the indices such that

the resulting distances $d(x_{j,\sigma(a)}, x_{j,\sigma(b)})$ would violate the unique ordering required to match $\mathcal{P}_S(X_i)$. For 1D Euclidean distances, $X_j^\sigma \sim_S X_i$ implies $\sigma$ must be the identity, which contradicts $i \neq j$ given the *Distinct Cells* property.

**2. Full Coverage:** Let $\mathcal{E}_k$ be an arbitrary equivalence class in $\mathbb{R}^N$ with distinct entries and let $Y \in \mathcal{E}_k$. Since the entries of $Y$ are distinct, there exists a unique permutation $\sigma \in \text{Perm}(N)$ such that $Y^\sigma \in \mathcal{X}_<$. By the *Full Coverage of Ordered Cells* property, there exists $X_i \in \mathcal{X}_{\text{rep}}^{MO}$ such that $Y^\sigma \sim_S X_i$. Applying the inverse permutation, we find $Y \sim_S X_i^{\sigma^{-1}}$. Thus, every equivalence class is represented by an element of $\mathcal{X}_{\text{rep}}^{MO}$ under some permutation. $\qquad\square$

**Computation of $\mathcal{X}_{\text{rep}}^{MO}$** To algorithmically construct $\mathcal{X}_{\text{rep}}^{MO}$, we observe that it is sufficient to identify a finite superset that spans all realizable ordinal profiles. The following properties guide this construction:

1. **Filtering Property:** If there exists a finite set $\mathcal{X}^* \subset \mathcal{X}_<$ such that $\mathcal{X}_{\text{rep}}^{MO} \subseteq \mathcal{X}^*$, then $\mathcal{X}_{\text{rep}}^{MO}$ can be obtained by iterating through $\mathcal{X}^*$ and retaining only one representative for each unique ordinal profile.

2. **Coverage Condition:** A set $\mathcal{X}^*$ contains a MORS if it is profile-exhaustive, i.e., $\{\mathcal{P}_S(X) \mid X \in \mathcal{X}^*\} = \{\mathcal{P}_S(X) \mid X \in \mathcal{X}_<\}$.

Based on these properties, the problem reduces to computing a finite set $\mathcal{X}^*$ that produces every possible ordinal profile achievable by strictly increasing configurations in $\mathbb{R}^N$. Rather than operating on coordinates $(x_1, \ldots, x_N)$, it is more efficient to parameterize the configurations using the vector of pairwise distances $\mathbf{d} = (d_{i,j})_{1 \leq i < j \leq N} \in \mathbb{R}^{\binom{N}{2}}$, where $d_{i,j} = x_j - x_i$.

For any $X \in \mathcal{X}_<$, we let $\mathbf{d} \in \mathbb{R}_+^M$ be the vector of its $M = \binom{N}{2}$ pairwise distances. We define the **total rank ordering** $\Omega(X)$ as the sequence of distance pairs sorted by magnitude, along with the set of transitional operators $\{<, =\}$ between adjacent elements in the sequence. For example, a possible rank ordering for $N = 3$ is $(d_{1,2} = d_{2,3} < d_{1,3})$. Since the ordinal profile $\mathcal{P}_S(X)$ (Definition 5) is determined solely by the relative signs of distance comparisons, it is a **deterministic function** of the total rank ordering: $\mathcal{P}_S(X) = \Phi(\Omega(X))$. It follows that by iterating through all geometrically feasible total rank orderings, we ensure that our exploration is **profile-exhaustive** over $\mathcal{X}_<$, as every realizable profile is captured by at least one such ordering.

Formally, obtaining a configuration $X$ that satisfies a candidate ordering $\Omega$ equates to finding a feasible solution $\mathbf{d}$ to a Linear Programming (LP) problem defined by the intersection of three constraint sets: $\mathcal{C}_\triangle$ (geometric triangle equalities), $\mathcal{C}_{\neq}$ (ensuring distinct points), and $\mathcal{C}_{\text{ord}}$ (the specific rank ordering and transitional operators defined by $\Omega$).

Crucially, we do not need to search the entire combinatorial space of all possible distance orderings. Because $X \in \mathcal{X}_<$, the constraints $\mathcal{C}_\triangle$ and $\mathcal{C}_{\neq}$ imply that for any $i < j < k$, $d_{i,k} = d_{i,j} + d_{j,k}$ with $d_{i,j}, d_{j,k} > 0$. This geometric necessity induces a **mandatory partial order** $\mathcal{C}_{\text{perm}}$ where $d_{i,k}$ must be strictly greater than its constituent segments $d_{i,j}$ and $d_{j,k}$. Consequently, any valid $\Omega$ must be consistent with $\mathcal{C}_{\text{perm}}$; specifically, the underlying permutation of distances must be a **topological sort** of the Directed Acyclic Graph (DAG) whose edges are defined by these transitional dependencies. Any ordering that violates $\mathcal{C}_{\text{perm}}$ is geometrically impossible in 1D space and can be pruned immediately.

Algorithm 6 formalizes this procedure. It iterates through the set of valid topological sorts $\mathcal{O}$ derived from the partial order $\mathcal{C}_{\text{perm}}$. For each sort, it considers all valid operator configurations $\mathbf{b} \in \{=, <\}^{M-1}$ to construct a candidate $\mathcal{C}_{\text{ord}}$. By solving the resulting LP for $\mathbf{d}$ under $\mathcal{C}_\triangle \wedge \mathcal{C}_{\text{ord}} \wedge \mathcal{C}_{\neq}$, the algorithm ensures that the resulting set $\mathcal{D}$ (and its coordinate counterpart $R_X$) is profile-exhaustive, allowing for the reliable extraction of the MORS (Definition 8).

---

**Algorithm 6** Obtain a $\mathcal{X}_{\text{rep}}^{MO}$

---

**Require:** Number of points $N$; Similarity metric $S \in \{\text{TSI}, \text{QSI}\}$; Precision $\Delta > 0$
1: $I \leftarrow \{(i,j) \mid 1 \leq i < j \leq N\}$
2: $\mathcal{C}_{\triangle} \leftarrow \{d_{i,j} + d_{j,k} = d_{i,k} \mid 1 \leq i < j < k \leq N\}$
3: $\mathcal{C}_{\neq} \leftarrow \{d_{i,j} \geq \Delta \mid 1 \leq i < j \leq N\}$
4: $\mathcal{C}_{\text{perm}} \leftarrow \{(d_{i,j}, d_{i,k}) \mid i < j < k\} \cup \{(d_{i,j}, d_{t,j}) \mid t < i < j\}$
5: $\mathcal{O} \leftarrow \text{TopologicalSorts}(\mathcal{C}_{\text{perm}})$
6: $\mathcal{D} \leftarrow \emptyset$
7: **for all** $\sigma \in \mathcal{O}$ **do**
8:     **for all** $\mathbf{b} \in \{=, <\}^{|I|-1}$ **do**
9:         $\mathcal{C}_{\text{ord}} \leftarrow \{d_{\sigma(k)} \, b_k \, d_{\sigma(k+1)} \mid k = 1, \ldots, |I| - 1\}$
10:         Solve LP: find $\mathbf{d}$ s.t. $\mathcal{C}_{\triangle} \wedge \mathcal{C}_{\text{ord}} \wedge \mathcal{C}_{\neq}$
11:         **if** LP is feasible **then**
12:             $\mathcal{D} \leftarrow \mathcal{D} \cup \{\mathbf{d}\}$
13:         **end if**
14:     **end for**
15: **end for**
16: $R_X \leftarrow [(0, d_{1,2}, d_{1,3}, \ldots, d_{1,N}) \mid \mathbf{d} \in \mathcal{D}]$
17: $\mathcal{X}_{\text{rep}}^{MO} \leftarrow \{R_X[i] \mid \nexists j > i : S_{d_X, d_X}(R_X[i], R_X[j]) = 1\}$
18: **return** $\mathcal{X}_{\text{rep}}^{MO}$

---

$\square$

