# OpenReview forum: "Scalable and Interpretable Representation Alignment with Ordinal Similarity"
_ICML.cc/2026/Conference — ICML 2026 regular_

### Official Review · Reviewer_2o5y · 2026-03-07

**Soundness:** 2
**Presentation:** 3
**Significance:** 2
**Originality:** 2
**Overall Recommendation:** 3
**Confidence:** 4

**Summary:**

This paper introduces two novel metrics for comparing neural representations: the Triplet Similarity Index (TSI) and the Quadruplet Similarity Index (QSI). These metrics are based on ordinal comparisons of distances between points in representation spaces. The authors provide a theoretical foundation for these metrics, including probabilistic interpretations, invariance properties, robustness guarantees, and connections to existing methods like Mutual Nearest Neighbors and Kendall’s Tau. They also propose efficient exact and approximate algorithms for computation and validate the utility of TSI and QSI in several experiments, including measuring representation convergence during training and multimodal alignment in CLIP models.

**Compliance With Llm Reviewing Policy:**

Affirmed.

**Key Questions For Authors:**

Your metrics rely on a distance function (e.g., Euclidean). How sensitive are TSI/QSI to this choice? Have you experimented with other distances (e.g., cosine, Manhattan) and observed significant differences?

In your approximation scheme, you sample triplets/quadruplets independently. How does the required number of samples scale with dataset size in practice?

**Limitations:**

--

**Strengths And Weaknesses:**

++The paper offers a fresh perspective on representation similarity by focusing on ordinal consistency rather than absolute distances.

++The authors provide a theoretical framework, including invariance proofs, robustness bounds, and connections to MutualNN and Kendall’s Tau.

++The approximation scheme based on U-statistics and Hoeffding bounds allows TSI and QSI to scale to large datasets with theoretical guarantees, addressing a major limitation of many existing metrics.


--Limited Empirical Validation: While the experiments are conceptually sound, they are limited in scope. The convergence analysis (Section 5.4) and CLIP alignment study (Section 5.5) are performed on small-scale settings (CIFAR-10, single model seeds). The paper would benefit from larger-scale evaluations on more diverse architectures and datasets (e.g., ImageNet, LLMs).

--Overclaiming of Interpretability: The authors claim that TSI/QSI are inherently interpretable, but in practice, interpreting a score of 0.6 vs 0.8 may still be difficult without further context. The baseline of 0.5 helps, but the meaning of intermediate values is not explored in depth.

--Weakness in Comparison to Baselines: In Figures 2 and 3, TSI/QSI appear stable and robust, but it is unclear whether this comes at the cost of sensitivity to meaningful differences. The paper does not include experiments showing that TSI/QSI can detect subtle but important representational changes that other metrics miss.

--Missing Discussion on Ties: The theoretical results often assume no ties, but in practice, ties can occur (especially in discrete or quantized representations). The paper does not thoroughly discuss how ties are handled or how they affect interpretability.

---

> ### Author Rebuttal · Authors · 2026-03-30
>
> We thank Reviewer 2o5y for their constructive review.  To aid the rebuttal, we provide a repository (https://anonymous.4open.science/r/icml2026-rebuttals-BB1D) containing plots, examples, bibliography, and proofs (referenced as Sup.{initials}). Shorthand **See tD1X-S1** kindly refers the reviewer to Section 1 of our response to Reviewer tD1X.
>
> ### 1. Larger-Scale Experiments
>
> > Limited Empirical Validation [...] benefit from larger-scale evaluations on more diverse architectures and datasets [...]
>
> * **Experiment - Larger-Scale Evaluations**
>     * **Setup** Following the experimental setup of Figure 3 in [3] (originally using MutualNN), we compute TSI and QSI between 11 LLMs and a ViT pre-trained on ImageNet21K [8] (4 distinct model sizes) on the WIT Dataset. We then plot these alignment scores against the LLM's downstream performance on OpenWebText [9] measured using 1-bit-per-byte.
>     * **Results** Similar to [3], across all ImageNet21K model sizes we observe a positive correlation between TSI/QSI scores and downstream LLM performance (see **Sup.EP.MLE**).
>     * **Insights** This confirms the practical utility of our metrics across diverse model architectures and datasets.
>
> > [...] The convergence analysis [...] CLIP alignment study [...] are performed on small-scale settings (CIFAR-10, single model seeds)
>
> * **Existing Experiments**
>     * **Convergence Analysis** Convergence analysis is an analysis averaged across 5 seeds as explained in Appendix J.
>     * **CLIP Alignment Study** CLIP alignment study is performed on the validation set of ImageNet containing 50,000 datapoints; we will include this missing detail in the final version.
>
> ### 2. Interpretability
>
> > [...] interpreting a score of 0.6 vs 0.8 may be difficult [...] the meaning of intermediate values is not explored in depth.
>
> * **Interpretability of Intermediate Scores** Unlike raw scores from other metrics, TSI and QSI have a direct probabilistic interpretation (Section 4.2). A TSI score of 0.6 translates to 60% of triplets agreeing on distance ordering.
>
> ### 3. Comparison to Baselines
>
> > [...] include experiments showing that TSI/QSI can detect [...] representational changes that other metrics miss
>
> *See **Sup.IEFD** and **Sup.IEGI.{LF, GF}** for formal descriptions and visualizations.*
>
> *   **Example - Local Failure Mode of CKA and MutualNN**
>     * **Setup** We use two clusters separated by $D \to \infty$. Intra-cluster permutation shuffles local points while preserving global assignments, introducing severe local deformation.
>     * **Scores** $\text{MutualNN}(X, X') = 1.0$; $\text{CKA}(X, X') = 1.0$; $\text{TSI}(X, X') < 1.0$; $\text{QSI}(X, X') < 1.0$.
>     * **Insights** MutualNN and CKA are insensitive to intra-cluster permutations, allowing local alignment to be destroyed without affecting the scores. Conversely, TSI and QSI successfully detect this misalignment.
>
> *   **Example - Global Failure Mode of CKNNA and MutualNN**
>     * **Description** We use three widely separated clusters $C_1, C_2, C_3$. In $X$, $C_3$ is closer to $C_1$; in $X'$, a global translation moves $C_3$ closer to $C_2$. This alters macroscopic layout while preserving local neighborhoods.
>     * **Similarity Scores** $\text{MutualNN}(X, X') = 1.0$; $\text{CKNNA}(X, X') = 1.0$; $\text{TSI}(X, X') < 1.0$; $\text{QSI}(X, X') < 1.0$.
>     * **Key Insights** $k$-NN-based metrics like MutualNN and CKNNA are blind to global transformations that preserve local neighborhoods. Conversely, TSI and QSI evaluate distances at all scales, successfully identifying macroscopic misalignment.
>
> ### 4. Handling of Ties
>
> > [...] explain how ties are handled or how they affect interpretability?
>
> *  **See R1U2-S5**
>
> ### 5. Sensitivity to Distance Function
>
> > [...] How sensitive are TSI/QSI to the distance function? [...] experiment with other distances (e.g., cosine, manhattan) [...]
>
> *   **Experiment - Sensitivity to Distance Function**
>     *   **Setup**: We interpolate between independent random datasets $X$ and $Y$ ($X_t = (1-t)X + tY$) and compute TSI/QSI between $X$ and $X_t$ across $t \in [0, 1]$ for Euclidean, Cosine, and Manhattan distances (averaged over 20 runs).
>     *   **Results**: TSI and QSI exhibit low sensitivity to the choice of distance function (see **Sup.EP.TQS**). While a consistent ranking persists (Cosine > Euclidean > Manhattan), the score differences are minimal across all values of $t$.
>     *   **Insights**: TSI and QSI behave similarly across these common distance metrics. However, since the optimal distance function depends on how semantic proximity is structured for each specific problem, these results are likely context-dependent.
>
> ### 6. Scaling with Size
>
> > [...] required number of triplets/quadruplets samples scale with dataset size in practice?
>
> * **See HCWG-S3**
>
> &nbsp;
>
> We thank Reviewer 2o5y for feedback that helped characterize distance selection and demonstrate TSI/QSI's superior detection of subtle structural changes. We look forward to the discussion.

---

> > ### Author Rebuttal · Reviewer_2o5y · 2026-04-08
> >
> > I still think this paper overclaim the interpretability. I can only maintain my score to still vote for a weak reject, but I do not have a strong disagreement if the AC or other reviewers want to accept it.

---

> > > ### Author Response · Authors · 2026-04-08
> > >
> > > We sincerely thank Reviewer 2o5y for their follow-up, for acknowledging that our initial response and additional experiments addressed the majority of their concerns, and for their openness to the paper's acceptance.
> > >
> > > > I still think this paper overclaim the interpretability. [...]
> > >
> > > Regarding interpretability, our claims are rooted in specific, mathematically grounded properties rather than subjective qualities. As detailed in the manuscript, the interpretability of TSI and QSI follows directly from two key components:
> > > 1. **Direct Probabilistic Meaning:** Unlike other metrics that yield abstract values, TSI and QSI are formulated as expectations of indicator functions (Section 4.2). An intermediate score of 0.8, for example, mathematically guarantees that exactly 80% of sampled ordinal relationships agree.
> > > 2. **Stable Baseline:** We theoretically prove (Lemma 1) and empirically validate (Section 5.1) that statistically independent representations yield a stable baseline of $\approx 0.5$. This provides a consistent, reliable reference point for dissimilarity across varying dimensions and sample sizes.
> > >
> > > Furthermore, as discussed in our follow-up response to Reviewer R1U2, we also established a sharp, quantitative connection between intermediate TSI scores and geometric mismatch. Specifically, Corollary 8 demonstrates that TSI is tightly upper-bounded by a convex combination of Mutual Nearest Neighbors (MNN) coefficients across all scales.
> > > $$
> > > \text{TSI}(X, Y) \leq \frac{1}{\binom{N-1}{2}} \sum_{k=1}^{N-2} k \cdot \text{MutualNN}_{k, d_X, d_Y}(X, Y)
> > > $$
> > > This means an intermediate score (e.g., a TSI of 0.8) doesn't just ambiguously imply similarity but places a hard, quantitative lower bound on how much the multi-scale neighborhood structure could have drifted before the TSI score is mathematically forced to decrease.
> > >
> > > We believe these properties: probabilistic clarity, consistent baselines, and sharp geometric bounds, collectively justify our claims regarding interpretability. We are grateful for the constructive discussion and the time the reviewer spent helping us refine this work.

---

### Official Review · Reviewer_HCWG · 2026-03-09

**Soundness:** 3
**Presentation:** 4
**Significance:** 2
**Originality:** 3
**Overall Recommendation:** 4
**Confidence:** 3

**Summary:**

This paper studies the problem of representation alignment, which aims to measure how similar two representation spaces are, a fundamental task in representation learning and model analysis. Existing similarity metrics often suffer from several issues, including poor interpretability, sensitivity to outliers, and limited scalability on large datasets.
To address these challenges, the authors propose an ordinal similarity framework that measures alignment by evaluating whether relative similarity relationships between samples are preserved across representation spaces. The framework is instantiated through two metrics: Triplet Similarity Index (TSI) and Quadruplet Similarity Index (QSI). These metrics compare ordinal relationships between samples (e.g., whether one sample is closer to another than to a third) instead of relying on absolute distance values.
The paper provides theoretical analysis showing that the proposed ordinal similarity formulation is inherently interpretable, robust to outliers, and computationally scalable. The authors also establish a connection between TSI-based alignment and local neighborhood alignment measured through Mutual Nearest Neighbors (MNN). Finally, extensive empirical experiments demonstrate that the proposed approach provides a practical and scalable way to evaluate representation alignment across different models and representation spaces.

**Compliance With Llm Reviewing Policy:**

Affirmed.

**Final Justification:**

The rebuttal resolved my main concerns on scalability and practical relevance, so I am raising my score from weak reject to weak accept.

**Key Questions For Authors:**

1.
The paper claims that the proposed Ordinal Similarity framework (TSI and QSI) is computationally scalable compared with existing representation similarity metrics. However, ordinal comparisons typically require evaluating a large number of pairwise or triplet relationships.
Could the authors provide a more explicit complexity analysis and empirical runtime comparison against commonly used metrics (e.g., CKA, RSA, SVCCA) on large-scale datasets (e.g., ≥1M samples)? In particular:
. What is the exact computational complexity of TSI and QSI?
. How does the method scale with dataset size and embedding dimensionality?
. Are sampling strategies required in practice?
A clear answer could significantly strengthen the claim that the method is truly scalable in realistic large-scale representation learning settings.

2.
The proposed metric is designed to measure representation alignment, but the paper focuses primarily on intrinsic evaluation.
Could the authors provide further evidence on how ordinal similarity correlates with downstream task performance, such as:
. transfer learning performance
. knowledge distillation effectiveness
. cross-model representation compatibility
Demonstrating such correlations would make the method significantly more compelling for practitioners.

**Limitations:**

The authors mention certain practical considerations, but the discussion of limitations and potential societal impact could be expanded to better contextualize the applicability and risks of the proposed method.
Although the method is described as scalable, ordinal similarity relies on relational comparisons that may grow rapidly with dataset size. The paper would benefit from briefly discussing the practical limits of the method and how sampling strategies might affect accuracy in very large datasets.
Since ordinal similarity produces a single interpretable score, practitioners might over-rely on it when evaluating models. The authors may consider clarifying that representation similarity does not necessarily imply functional equivalence or similar downstream performance.

**Strengths And Weaknesses:**

Soundness:
The paper is generally technically sound, with clear definitions of TSI and QSI and supporting theoretical analysis and empirical experiments. The connection to Mutual Nearest Neighbor (MNN) structures also provides useful theoretical insight. However, the scalability claims could be further strengthened with more analysis on computational cost and sampling strategies in very large datasets.

Presentation:
The paper is well organized, and the motivation for using ordinal relationships to measure representation similarity is clearly explained. Some theoretical sections are relatively dense, and additional intuitive explanations or examples could improve readability.

Significance:
Representation similarity is an important problem in modern machine learning, and the proposed framework provides a potentially interpretable and scalable tool for analyzing representation alignment. However, the practical impact could be clearer if the paper showed stronger connections between the metric and downstream tasks or model performance.

Originality:
The paper introduces an ordinal perspective on representation similarity through the TSI and QSI metrics. The contribution mainly lies in reformulating similarity through ordinal relations, and more discussion comparing with related relational similarity metrics would better highlight the novelty.

---

> ### Author Rebuttal · Authors · 2026-03-30
>
> We thank Reviewer HCWG for their feedback. To aid the rebuttal, we provide a repository (https://anonymous.4open.science/r/icml2026-rebuttals-BB1D) with plots, examples, bibliography, and proofs (**Sup.{initials}**). Shorthand **See tD1X-S1** kindly refers the reviewer to Section 1 of our response to Reviewer tD1X.
>
> ### 1. Comparison to other Similarity Metrics
>
> > [...] additional intuitive explanations or examples could improve readability [...] more discussion comparing with related relational similarity metrics would better highlight the novelty.
>
> * **Illustrative Examples** We provide three illustrative examples detailing structural differences between considered metrics. Detailed descriptions and illustrations are available in **Sup.IEFD** and **Sup.IEGI**, respectively.
>     *   **QSI captures global changes while TSI is locally invariant - See tD1X-S1**
>     *   **Global Failure Mode of CKNNA and MutualNN - See 2o5y-S3**
>     *   **Local Failure Mode of CKA and MutualNN - See 2o5y-S3**
> * **Connection to Mutual Nearest Neighbours** There are significant connections between TSI and Mutual Nearest Neighbors (MNN) structures.
>     * **Perfect TSI alignment is equivalent to identical Mutual Nearest Neighbors (MNN):** As proven in Corollary 1, full alignment of TSI is mathematically equivalent to the joint alignment of MNN across all possible scales $k$.
>     * **Higher TSI requires Shared MNN:** Expanding on this connection, Corollary 8 establishes that the TSI score is tightly upper-bounded by a convex combination of MNN coefficients, deeply intertwining our metric with geometric neighborhood preservation.
>
>
> ### 2. Correlation with Downstream Performance
>
> > [...] the practical impact could be clearer [...] with connections between the metric and downstream tasks [...]
>
> * **Experiment - Correlation with Downstream Performance**
>     * **Setup** Following the experimental setup of Figure 4 in [3] (originally using MutualNN), we compute TSI and QSI between 18 LLMs and DINOv2 [4] on the WIT dataset [5]. We then plot these alignment scores against the LLM's downstream performance on Hellaswag [6] and GSM8K [7].
>     * **Results** Similar to [3], we observe a positive correlation between TSI/QSI scores and downstream LLM performance (see **Sup.EP.QCD and Sup.EP.TCD**). Both metrics indicate low absolute similarity between LLM and DINOv2, consistent with the original study.
>     * **Insights** This confirms our metrics' utility for analyzing the link between model alignment and performance.
>
> ### 3. Scaling with Size
>
> Efficient computation is a key contribution, detailed in Section 4.5 of the manuscript.
>
> #### Exact Computation
>
> > [...] Could the authors provide a more explicit complexity analysis [...] comparison against commonly used metrics (e.g, CKA, RSA, SVCCA)? [...]
>
> * **Exact Computation:** TSI/QSI can be computed in $\mathcal{O}(N^2 \log N)$ time with $\mathcal{O}(N)$ and $\mathcal{O}(N^2)$ space complexity, respectively.
> * **Comparison:** While our exact complexity is comparable to CKA/RSA (at least $\mathcal{O}(N^2)$ to compute pairwise distances) and SVCCA ($\mathcal{O}(NM \min(M, N))$ for $M$-dimensional representations), our approximation scheme scales significantly better (see below).
>
> #### Approximate Computation
>
> > [...] How does the method scale with dataset size and embedding dimensionality? Are sampling strategies required in practice? [...] analysis on sampling strategies in very large datasets.
>
> * **Scaling with Dataset Size** Sampling $n = \lceil \frac{1}{2\epsilon^2}\log(\frac{2}{\delta}) \rceil$ tuples estimates TSI/QSI with additive error $\le \epsilon$ with probability $\ge 1-\delta$. Since $n$ is independent of $N$, we obtain an **equally good approximation regardless of dataset size**, a major scalability advantage.
> * **Embedding Dimensionality** Similar to other distance-based metrics, when equipped with euclidean distance, the complexity grows linearly with the embedding dimensionality $M$.
>
> ### 4. On Limitations and Societal Impact
>
> > [...] the discussion of limitations and potential societal impact could be expanded [...]
>
> * **Limitations:** As we show in **R1U2-S1**, we will clarify that empirical estimates require caution for small $N$.
> * **Societal Impact:** Given the potential applications (e.g., ML, neuroscience brain representations), we will include a discussion on the ethical implications of using these metrics on human-centric data.
>
> >[...] practitioners might overrely on single-similarity score [...] clarifying that representation similarity does not necessarily imply [...] similar downstream performance
>
> * **Performance vs. Alignment:** We will clarify that high similarity does not guarantee identical performance, distinguishing alignment from functional equivalence.
>
> &nbsp;
>
> We thank Reviewer HCWG for their suggestions. The resulting LLM experiments on downstream performance strengthen the practical utility of our metrics. We look forward to the discussion.

---

> > ### Author Rebuttal · Reviewer_HCWG · 2026-04-02
> >
> > Thank you for the rebuttal. My concerns have been adequately addressed. The rebuttal clarifies the scalability claims and provides additional evidence of practical relevance through correlation with downstream performance. Overall, this resolves my main concerns.

---

> > > ### Author Response · Authors · 2026-04-03
> > >
> > > We sincerely thank the reviewer for the positive follow-up and are very glad that our rebuttal adequately addressed your main concerns regarding scalability and practical relevance. Since all concerns have been successfully resolved, we would kindly ask if you might consider raising your score to reflect this outcome. We remain available should any further questions arise.

---

### Official Review · Reviewer_R1U2 · 2026-03-13

**Soundness:** 3
**Presentation:** 3
**Significance:** 3
**Originality:** 3
**Overall Recommendation:** 4
**Confidence:** 3

**Summary:**

This paper proposes ordinal similarity metrics for representation alignment: the Triplet Similarity Index (TSI) and Quadruplet Similarity Index (QSI). Instead of comparing raw distances or correlations, the method measures whether relative distance orderings are preserved across two representation spaces. The paper argues that this yields a notion of similarity that is interpretable, robust to outliers, and scalable than standard alternatives. It provides theoretical results on probabilistic interpretation, a link between perfect TSI and multiscale mutual-nearest-neighbor consistency, robustness under corruption, and efficient exact and approximate algorithms. Empirical results on representation spaces support these claims.

**Compliance With Llm Reviewing Policy:**

Affirmed.

**Final Justification:**

The follow-up rebuttal addresses several of my remaining questions. The authors now provide a quantitative bound relating $\text{TSI} < 1$ to MutualNN coefficients, strengthen the distance-sensitivity discussion with a controlled experiment across several distance functions, and give a concrete example where QSI captures a structural change that TSI misses. These clarifications strengthen the paper. My remaining reservation is that the practical interpretation of intermediate scores and the choice of distance function remain somewhat application-dependent. I therefore keep my initial weak accept.

**Key Questions For Authors:**

1) Can you provide a practical approximate counterpart? For example, if TSI is high but not 1, what quantitative statement can we make about average or worst-case neighborhood preservation?

2) How sensitive are TSI and QSI to the choice of distance function in realistic embedding spaces? The CLIP section uses both Euclidean and negative cosine, but the paper also notes that negative cosine may violate metric axioms and lose some guarantees. What should practitioners do by default?

3) How should ties be handled in practice for low-precision embeddings? Do your algorithms and theoretical guarantees degrade in such settings?

4) Can you construct examples where two representation spaces have high TSI/QSI but are badly misaligned in a way practitioners would still care about? This would help clarify the boundary of what ordinal similarity does and does not measure.

5) Why should practitioners use QSI in addition to TSI? Can you give a concrete setting where QSI reveals a failure mode that TSI misses?

**Limitations:**

1) The proposed methodology captures ordinal consistency only. It measures whether relative distance orderings are preserved, but does not account for absolute distance calibration, margins, or other geometric properties that may still matter in practice.

2) Some of the strongest interpretations rely on exact conditions, especially the TSI=1 result and the assumption that ties are negligible. As a result, intermediate scores are less directly interpretable, and the method may be less appealing in low-precision settings.

3) The method is sensitive to the choice of distance function, and some other relevant choices, such as negative cosine similarity, fall outside the metric setting assumed by parts of the theory.

4) Are the underlying comparisons combinatorial?

Read section above on *Weaknesses* herein repeated:

1) The notion of "alignment" may be too ordinal-only for some applications.

2) Tie handling feels underdeveloped.

3) Experiments limited.

**Strengths And Weaknesses:**

*Soundness*: The paper is technically sound: TSI/QSI are clearly defined, the theoretical claims are structured, and the results are supported by proofs and experiments. The $\approx 0.5$ independence baseline, robustness analysis, and scalability results support the method’s formulation. The issue is that the paper supports ordinal alignment specifically, and not representation similarity in a general geometric sense.

*Presentation*: The paper is clearly written and easy to follow. The motivation is strong, the definitions of TSI and QSI are intuitive, and the contribution list is well organized. The paper also does a good job of aligning the empirical sections with the claimed properties: interpretability, robustness, and scalability. The presentation would be even stronger with a more explicit discussion of limitations such as tie handling and the fact that ordinal agreement does not capture all aspects of geometric fidelity.

*Significance*: Representation similarity is important in modern ML, and the paper addresses limitations of existing metrics in interpretability, robustness, and scalability. The ordinal formulation is useful because it provides a stable baseline and a probabilistic interpretation. If generalized beyond the settings studied here, it could be useful for analyzing training dynamics, multimodal alignment, and representation convergence.

*Originality*: The paper’s main idea is to define representation alignment through preservation of ordinal relations. In this context, that is a novel formulation. The paper also provides theoretical interpretation, a link to multiscale neighborhood consistency, robustness guarantees, and scalable exact and approximate computation, which makes the contribution strong.

*Strengths*: The main idea is clear, and the score is easy to interpret, which is an advantage. The paper provides a baseline interpretation, a connection to neighborhood structure and multiscale nearest-neighbor consistency, and scalability guarantees. The experiments support the main claims.

*Weaknesses*: A strong limitation is that the method measures alignment only at the ordinal level. As a result, it does not capture absolute distance calibration, margin structure, or other geometric properties. The treatment of ties is limited, and the empirical evaluation is also limited to establish the method across many modalities and applications.

1) The notion of "alignment" may be too ordinal-only for some applications.

2) Tie handling feels underdeveloped.

3) Experiments limited.

---

> ### Author Rebuttal · Authors · 2026-03-30
>
> We thank Reviewer R1U2 for their valuable feedback. To aid the rebuttal, we provide a repository (https://anonymous.4open.science/r/icml2026-rebuttals-BB1D) containing plots, examples, bibliography, and proofs (referenced as Sup.{initials}). Shorthand **See tD1X-S1** kindly refers the reviewer to Section 1 of our response to Reviewer tD1X.
>
> ### 1. Ordinal vs Nominal based Representation Similarity
>
> >[...] measures alignment only at the ordinal level [...] does not capture absolute distance calibration [...] geometric properties [...]
>
> * **Ordinal Relationships are Strongly Restrictive as $N \to \infty$:** While discarding absolute distances may appear to inflate similarity, ordinal relationships impose a number of constraints that grows cubically (or faster) with the number of points $N$, sharply restricting admissible configurations [1]. Additionally, under suitable conditions and as $N \to \infty$, ordinal relationships uniquely determine a representation up to similarity-preserving transformations (e.g., isotropic scaling, orthogonal transforms) [2]. **This makes ordinal consistency as restrictive as nominal consistency when $N$ is large**.
> * **Connection to Geometric Properties:** There are significant connections between TSI and Mutual Nearest Neighbors (MNN) structures (**HCWG-S1**).
>
> > [..] construct examples where two representation spaces have high TSI/QSI but are badly misaligned [...]
>
> * **High TSI/QSI vs. Misalignment:** TSI=1 is equivalent to perfect Mutual Nearest Neighbor alignment at all scales (Section 4.3). Since perfect neighborhood alignment is itself a strong notion of alignment that practitioners care about, a construction with TSI=1 that is simultaneously considered misaligned would be contradictory. For values strictly below 1, the probabilistic interpretation provides a direct quantitative account of the degree of misalignment.
> * **Theory - Empirical vs. True Similarity:** A related concern is whether high *empirical* TSI/QSI on a finite sample set could be misleading when the true similarity  $\text{TSI}^\*$ of the joint distribution $\mathcal{D}_{XY}$ (i.e. the TSI value one would obtain with infinite data) is low. **Sup.P** shows that the empirical TSI/QSI concentrates tightly around this true value. In particular, for any $\epsilon > 0$, the probability that the empirical TSI (similarly for QSI) deviates from the true similarity is bounded by:
> $$
> \mathbb{P} [ | \text{TSI} - \text{TSI}^* | \geq \epsilon ] \leq 2 \exp(-2 \lfloor N/3 \rfloor \epsilon^2)
> $$
> * **Ordinal Consistency Measures Geometric Alignment:** Because ordinal constraints strongly dictate structure for large $N$, and our empirical metrics tightly estimate true similarity, TSI and QSI are reliable measures of geometric alignment. Their accuracy improves as dataset size grows, though caution is advised for very small $N$.
>
> ### 2. Multimodal Experiments
>
> > The empirical evaluation is also limited [...] many modalities and applications
>
> **See HCWG-S2 and 2o5y-S1**
>
> ### 3. Connection to Mutual Nearest Neighbours when TSI < 1
>
> > [...] if TSI is high but not 1, what quantitative statement can we make about average or worst-case neighborhood preservation? [...]
>
> * **Higher TSI requires Shared Mutual Nearest Neighbors (see HCWG-S1)**
>
> ### 4. Sensitivity to Distance Function
>
> > How sensitive are TSI and QSI to the choice of distance function in realistic embedding spaces? [...] What should practitioners select by default? [...]
>
> *  **Sensitivity Analysis (see 2o5y-S5)**
> *  **Distance Function Selection** Preferably the selected distance functions satisfy the metric axioms. However, in general, practitioners should select the distances that best represent similarity for their concrete use-case.
>
> ### 5. Handling of Ties
>
> > [...] How should ties be handled in practice for low-precision embeddings? [...] the assumption that ties are negligible [...] intermediate scores are less directly interpretable [...]
>
> * **Impact of Ties on Baseline Dissimilarity:** The baseline dissimilarity (Lemma 1) is a function of tie probability. Practitioners can estimate tie probabilities via Monte-Carlo to determine the expected baseline for any dataset. Thus, a **theoretically defined baseline exists for any tie probability**, although it may deviate from 1/2 when ties are present.
> * **Consistency under Ties:** Theoretical connections to Mutual Nearest Neighbors (MNN) assume no ties as MNN is ill-defined with them. The tie assumption does not degrade TSI nor QSI.
> * **Interpretability of Intermediate Scores:** See **2o5y-S2**.
>
> ### 6. TSI/QSI Differences
>
> > [...] Can you give a concrete setting where QSI reveals a failure mode that TSI misses?
>
> *  **See tD1X-S1**
>
> ### 7. Scaling with Size
>
> > Are the underlying comparisons combinatorial?
>
> *  **See HCWG-S3**
>
> &nbsp;
>
> We thank Reviewer R1U2 for feedback that led us to prove empirical ordinal similarity is bounded and faithful to nominal consistency for large $N$. We look forward to the discussion.

---

> > ### Author Rebuttal · Reviewer_R1U2 · 2026-04-03
> >
> > The rebuttal partially addresses my questions.
> >
> > On the first and fourth points, the authors strengthen their case that ordinal similarity captures meaningful geometric structure: they argue that for large $N$, ordinal constraints are highly restrictive, relate TSI more directly to MNN structure, and add a concentration argument showing empirical TSI/QSI tracks the population quantity. However, the response still does not provide a sharp practical approximate counterpart for TSI $<1$ beyond the qualitative statement that higher TSI implies more shared mutual nearest neighbors, and it does not fully characterize the boundary between high ordinal similarity and other forms of geometric mismatch.
> >
> > On the distance-function question, the answer is mostly pragmatic -prefer metric distances, but use what fits the application- so principled default guidance remains limited.
> >
> > On ties, the rebuttal is more satisfactory: it explains that the baseline can be adjusted using tie probabilities, though the clean MNN-based interpretation still relies on the no-ties setting.
> >
> > Finally, the response does not clearly answer when QSI should be preferred over TSI or provide a concrete failure mode that QSI captures and TSI misses.
> >
> > *Conclusion: Overall, the rebuttal improves the paper, but my main concerns are only partially resolved, so I keep my initial weak accept.*

---

> > > ### Author Response · Authors · 2026-04-03
> > >
> > > We thank Reviewer R1U2 for the follow-up and for maintaining the positive recommendation. We provide a brief clarification on the three points mentioned as partially resolved:
> > >
> > > ### 1. A Sharp, Quantitative Counterpart for $\text{TSI} < 1$
> > >
> > > > [...] does not provide a sharp practical approximate counterpart for $\text{TSI} < 1$ [...] and it does not fully characterize the boundary between high ordinal similarity and other forms of geometric mismatch.
> > >
> > > * **A Sharp, Quantitative Counterpart for $\text{TSI} < 1$:** The connection between TSI and shared mutual nearest neighbors is a strict quantitative bound rather than a qualitative statement.
> > >     * **Theory - Upper Bound:** In Corollary 8 (Appendix E of the original manuscript), we show that TSI is tightly upper-bounded by a convex combination of Mutual Nearest Neighbors coefficients:
> > >     $$
> > >     \text{TSI}(X, Y) \leq \frac{1}{\binom{N-1}{2}} \sum_{k=1}^{N-2} k \cdot \text{MutualNN}_{k, d_X, d_Y}(X, Y)
> > >     $$
> > >     * **Key Insight:** This directly provides the requested sharp practical counterpart and characterizes the boundary of geometric mismatch: a high TSI score mathematically necessitates a specific minimum preservation of shared mutual nearest neighbors across scales. For practitioners, a TSI of 0.8 doesn't just "imply" similarity; it places a hard, quantitative bound on how much the neighborhood structure could have drifted before the TSI score is forced to decrease.
> > >
> > > ### 2. Hardness of Principled Default Guidance
> > >
> > > > On the distance-function question, the answer is mostly pragmatic -prefer metric distances, but use what fits the application- so principled default guidance remains limited.
> > >
> > > * **Hardness of Principled Default Guidance:** Providing a principled default distance is challenging because the appropriate metric fundamentally depends on the specific problem. For example, if representations were trained to optimize a distance distillation loss, Euclidean distance is a natural choice. Conversely, if evaluating raw biological data like protein sequences, an evolutionary distance is more appropriate. Ultimately, the optimal distance function is deeply tied to the practitioner's specific use-case and research questions.
> > >
> > > In our response to Reviewer 2o5y, we explore the sensitivity of TSI and QSI to the choice of
> > > distance function in a controlled experiment.
> > >
> > > *   **Experiment - Sensitivity to Distance Function**
> > >     *   **Setup**: We interpolate between independent random datasets $X$ and $Y$ ($X_t = (1-t)X + tY$) and compute TSI/QSI between $X$ and $X_t$ across $t \in [0, 1]$ for Euclidean, Cosine, and Manhattan distances (averaged over 20 runs).
> > >     *   **Results**: TSI and QSI exhibit low sensitivity to the choice of distance function (see plot here **[Sup.EP.TQS](https://anonymous.4open.science/r/icml2026-rebuttals-BB1D/EP%20-%20experiment-plots/TQS%20-%20tsi-qsi-sensitivity-to-distance-selection.png)**). While a consistent ranking persists (Cosine > Euclidean > Manhattan), the score differences are minimal across all values of $t$.
> > >     *   **Insights**: TSI and QSI behave similarly across these common distance metrics. However, since the optimal distance function depends on how semantic proximity is structured for each specific problem, these results are likely context-dependent.
> > >
> > > ### 3. Concrete Failure Mode (QSI vs. TSI)
> > >
> > > > Finally, the response does not clearly answer when QSI should be preferred over TSI or provide a concrete failure mode that QSI captures and TSI misses.
> > >
> > > In our response to Reviewer tD1X, we provide an example where QSI captures global changes that leave local nearest neighbor structures intact.
> > >
> > > *   **Example - QSI captures global changes that leave local nearest neighbor structures intact**
> > >     * **Setup** We use a 1D dataset $X = \{0, 1, D, D + 2\}$ and its transformed version $X' = \{0, 1, D, D + 1\}$, with $D \gg 1$. This represents two distant clusters where one undergoes a local contraction preserving distance rankings for each anchor. See **[Sup.IEGI.DTQ](https://anonymous.4open.science/r/icml2026-rebuttals-BB1D/IEGI%20-%20illustrative-examples-graphical-images/DTQ%20-%20distinguishing-tsi-qsi.png)** for a visual representation of the example and **[Sup.IEFD](https://anonymous.4open.science/r/icml2026-rebuttals-BB1D/IEFD%20-%20illustrative-examples-formal-description.md)** for formal details.
> > >     * **Scores** $\text{TSI}(X, X') = 1$; $\text{QSI}(X, X') = 1/3$.
> > >     * **Insights** TSI is invariant to transformations preserving the nearest-neighbor structure of each individual point. Conversely, QSI measures consistency across all pairs, providing a stricter global alignment metric sensitive to structural changes that leave local neighborhoods intact.
> > >
> > > &nbsp;
> > >
> > > We will incorporate these specific insights and the concentration arguments into the final text to provide the guidance the reviewer suggested. Given these clarifications and the newly added multimodal experiments, we hope to have addressed the reviewer's concerns.

---

### Official Review · Reviewer_tD1X · 2026-03-13

**Soundness:** 4
**Presentation:** 4
**Significance:** 3
**Originality:** 2
**Overall Recommendation:** 5
**Confidence:** 5

**Summary:**

The paper introduces two metrics for measuring the similarity of two representations- triple similarity index (TSI) and quadruplet similarity index (QSI). Rather than relying on concrete distances, these metrics use the rank ordering of distances between triplets and quadruplets of points. These similarity metrics have useful properties for practitioners, including a probabilistic interpretation, an interpretable baseline for independent representations, robustness to outliers, and efficient computation.

**Compliance With Llm Reviewing Policy:**

Affirmed.

**Final Justification:**

The author rebuttal was compelling and I think the work overall is well-executed and clearly communicated. I recommend acceptance.

**Key Questions For Authors:**

None

**Limitations:**

Yes

**Strengths And Weaknesses:**

Strengths:
* The paper is clearly written and provides intuitive explanations and useful resources for practitioners.
* The authors provide useful implementational details, drawing on connections to well-known quantities like Kendall’s tau, that significantly improve computational load compared to brute force. This is both an interesting conceptual connection as well as a useful implementation.
* The synthetic data results are compelling.

Weaknesses:
* The paper presents two metrics but does not carefully discuss their differences &mdash; for example, are there cases where one metric is superior to the other? Are they more or less sensitive to different aspects of the data?
* Novelty is somewhat limited, at least in the computational neuroscience and neuroimaging literature &mdash; in RSA it is quite common to take the Spearman correlation between the intra-system representational distances (see e.g. [Kriegeskorte et al. 2008, Neuron](https://www.cell.com/neuron/fulltext/S0896-6273(08)00943-4)).
* The real-world data cases are more subtle than the synthetic data. For example, CKNNA and MutualNN capture the same effect as these models but are "intrinsically limited." But these limitations do not appear to affect the results so an example where the reliance on local structure is truly detrimental would make the point stronger.
* As noted in Appendix I.2, the MutualNN and CKNNA baselines behave strangely because "standard implementations... typically utilize the negative dot product as a proxy for distance" which does not satisfy the triangle inequality. It seems like using a proper metric in these baselines would be an easy fix, even if it is not "standard" in the literature. So the baselines are potentially weaker than necessary.
* The overall similarity score provided by this method does not provide a proper metric between neural representations, which some have argued is important ([Williams et al. 2021](https://proceedings.neurips.cc/paper/2021/hash/252a3dbaeb32e7690242ad3b556e626b-Abstract.html)).
* It would be nice if the ordinal similarity measures could be rigorously related to something about the topology or the geometry of the neural representations or (even better) the decodable information within the representations. Better establishing the functional importance of this statistic would be helpful.

---

> ### Author Rebuttal · Authors · 2026-03-30
>
> We thank Reviewer tD1X for their valuable feedback and thoughtful questions. To aid the rebuttal, we provide a repository https://anonymous.4open.science/r/icml2026-rebuttals-BB1D containing plots, examples, bibliography, and proofs (referenced as **Sup.{initials}**). Shorthand **See R1U2-S1** kindly refers the reviewer to Section 1 of our response to Reviewer R1U2.
>
> ### 1. TSI/QSI Differences
>
> > The paper presents two metrics [...] are there cases where one metric is superior to the other? Are they more or less sensitive to different aspects of the data?
>
> *   **Example - QSI captures global changes that leave local nearest neighbor structures intact**
>     * **Setup** We use a 1D dataset $X = \{0, 1, D, D + 2\}$ and its transformed version $X' = \{0, 1, D, D + 1\}$, with $D \gg 1$. This represents two distant clusters where one undergoes a local contraction preserving distance rankings for each anchor. See **Sup.IEFD** and **Sup.IEGI.DTQ** for formal details and visualization.
>     * **Scores** $\text{TSI}(X, X') = 1$; $\text{QSI}(X, X') = 1/3$.
>     * **Insights** TSI is invariant to transformations preserving the nearest-neighbor structure of each individual point. Conversely, QSI measures consistency across all pairs, providing a stricter global alignment metric sensitive to structural changes that leave local neighborhoods intact.
>
> ### 2. Comparison to Baselines
>
> > Novelty is [...] limited, [...] in RSA it is [...] common to take the Spearman correlation [...]
>
> * **Connection to Rank Correlations (RSA)** Appendix F shows that TSI and QSI are specific aggregations of the Kendall rank correlation (Corollaries 9 and 10); we will add a brief pointer to this in the main text as well.
> * **Theoretical Novelty** Our key contribution is less about the ordinal metric itself and more about establishing its rigorous theoretical foundations. We derive properties previously unexplored in this context, such as interpretable scores (Section 4.2), connection to neighborhood consistency (Section 4.3), outlier robustness (Section 4.4), and scalable approximations (Section 4.5), all of which are highly relevant for practitioners.
>
>
> > [...] CKNNA and MutualNN [...] are "intrinsically limited." [...] an example where the reliance on local structure is truly detrimental would make the point stronger.
>
> *  **See "Global Failure [...]" in 2o5y-S3**
>
> > [...] the MutualNN and CKNNA baselines behave strangely because [...] they utilize the negative dot product [...] which does not satisfy the triangle inequality [...] using a proper metric in these baselines would be an easy fix [...]
>
> * **Replacing Negative Dot Product in MutualNN/CKNNA:**
>     * **No Theoretical Guarantees of Robustness** While replacing the negative dot product addresses the issue in certain practical settings, there are still scenarios where these metrics remain dramatically affected by outliers.
>     * **Structural Limitations** Even with proper metrics, MutualNN and CKNNA suffer from fundamental limitations, to name a few illustrated by the examples above: they are blind to the relative ordering of neighbors within the $k$-neighborhood, they are inherently limited to a single neighborhood scale ($k$).
> * **TSI/QSI are Theoretically Robust** Unlike local metrics, the theoretical robustness bounds of TSI and QSI presented in Corollary 2 are applicable to all distance functions, even including the negative dot product.
>
> ### 3. Do TSI and QSI satisfy the Metric Axioms?
>
> > The overall similarity score provided by this method does not provide a proper metric between neural representations [...]
>
> *   **Similarity and Distance are inversely related concepts** Lower distance implies higher similarity and vice-versa. Therefore, it is natural that our metrics, being similarity indices, are not proper distance metrics in their original form.
> * **Theory - TSI/QSI complements are Proper Metrics:** In **Sup.P**, we prove that $1-\text{TSI}$ and $1-\text{QSI}$ satisfy all metric axioms, including the triangle inequality. This follows from showing that these distances are equivalent to a normalized Hamming distance over ordinal relationship arrays, which naturally satisfies the axioms.
>
> ### 4. Connection to Topology and Geometry
>
> > [...] ordinal similarity measures could be rigorously related to [...] topology, the geometry or decodable information of the neural representations [...]
>
> * **Topology and Geometry:** These properties are intrinsically linked to nearest neighbor structure and its preservation across spaces.
>     *  **See "Connection to Mutual Nearest Neighbours" in HCWG-S1**
> * **Connection to Decodable Information:** While linking ordinal similarity to downstream decodability is complex, depending on decoder choice, we agree this represents an exciting avenue for future research.
>
> &nbsp;
>
> We thank Reviewer tD1X for feedback that led to proving $1-\text{TSI}$ and $1-\text{QSI}$ as proper metrics. We look forward to the discussion.

---

> > ### Author Rebuttal · Reviewer_tD1X · 2026-04-01
> >
> > Thanks. I raised my score to accept &mdash; I particularly like the fact that it can be shown that $1-\text{TSI}$ and $1-\text{QSI}$ are distance metrics.

---

> > > ### Author Response · Authors · 2026-04-03
> > >
> > > We thank Reviewer tD1X for the thoughtful feedback and for raising the score to Accept.

---

### Decision · Program_Chairs · 2026-04-30

**Decision:**

Accept (regular)

**Comment:**

Reviewers were largely positive on the paper, finding that it was exhaustive in investigating properties of the proposed ordinal similarity metric, even if the idea of an ordinal similarity metric itself is not novel. One reviewer (HCWG) maintained concerns about the *ad hoc* nature of the choice of distance function, but I assess that this can be incorporated as a limitation of the present work, with the work still standing enough on its own. Further, I assess that the authors have been sufficiently precise in defining interpretability here to address a concern about the subjective nature of interpretability elsewhere (2o5y).